# Inductive Bias of Gradient Descent for Exponentially Weight Normalized Smooth Homogeneous Neural Nets

## Abstract

We analyze the inductive bias of gradient descent for weight normalized smooth homogeneous neural nets, when trained on exponential or cross-entropy loss. Our analysis focuses on exponential weight normalization (EWN), which encourages weight updates along the radial direction. This paper shows that the gradient flow path with EWN is equivalent to gradient flow on standard networks with an adaptive learning rate, and hence causes the weights to be updated in a way that prefers asymptotic relative sparsity. These results can be extended to hold for gradient descent via an appropriate adaptive learning rate. The asymptotic convergence rate of the loss in this setting is given by $\Theta(\frac{1}{t(\log t)^2})$, and is independent of the depth of the network. We contrast these results with the inductive bias of standard weight normalization (SWN) and unnormalized architectures, and demonstrate their implications on synthetic data sets. Experimental results on simple data sets and architectures support our claim on sparse EWN solutions, even with SGD. This demonstrates its potential applications in learning prunable neural networks.

## 1 Introduction

The prevailing hypothesis for explaining the generalization ability of deep neural nets, despite their ability to fit even random labels (Zhang et al., 2017), is that the optimisation/training algorithms such as gradient descent have a 'bias' towards 'simple' solutions. This property is often called inductive bias, and has been an active research area over the past few years.

It has been shown that gradient descent does indeed seem to prefer 'simpler' solutions over more 'complex' solutions, where the notion of complexity is often problem/architecture specific. The predominant line of work typically shows that gradient descent prefers a least norm solution in some variant of the $L_2$-norm. This is satisfying, as gradient descent over the parameters abides by the rules of $L_2$ geometry, i.e. the weight vector moves along direction of steepest descent, with length measured using the Euclidean norm. However, there is nothing special about the Euclidean norm in the parameter space, and hence several other notions of 'length' and 'steepness' are equally valid. In recent years, several alternative parameterizations of the weight vector, such as Batch normalization and Weight normalization, have seen immense success and these do not seem to respect $L_2$ geometry in the 'weight space'. We pose the question of inductive bias of gradient descent for some of these parameterizations, and demonstrate interesting inductive biases. In particular, it can still be argued that gradient descent with these reparameterizations prefers simpler solutions, but the notion of complexity is different.

### 1.1 Contributions

The three main contributions of the paper are as follows.

- We establish that the gradient flow path with exponential weight normalization is equal to the gradient flow path of an unnormalized network using an adaptive neuron dependent learning rate. This provides a crisp description of the difference between exponential weight normalized networks and unnormalized networks.

- We establish the inductive bias of gradient descent on standard weight normalized and exponentially weight normalized networks and show that exponential weight normalization is likely to lead to asymptotic sparsity in weights.

- We provide tight asymptotic convergence rates for exponentially weight normalized networks.

## 2  RELATED WORK

### 2.1  INDUCTIVE BIAS

Soudry et al. (2018) showed that gradient descent(GD) on the logistic loss with linearly separable data converges to the $L_2$ maximum margin solution for almost all datasets. These results were extended to loss functions with super-polynomial tails in Nacson et al. (2019b). Nacson et al. (2019c) extended these results to hold for stochastic gradient descent(SGD) and Gunasekar et al. (2018a) extended the results for other optimization geometries. Ji & Telgarsky (2019b) provided tight convergence bounds in terms of dataset size as well as training time. Ji & Telgarsky (2019a) provide similar results when the data is not linearly separable.

Ji & Telgarsky (2019c) showed that for deep linear nets, under certain conditions on the initialization, for almost all linearly separable datasets, the network, in function space, converges to the maximum margin solution. Gunasekar et al. (2018b) established that for linear convolutional nets, under certain assumptions regarding convergence of gradients etc, the function converges to a KKT point of the maximum margin problem in fourier space. Nacson et al. (2019a) shows that for smooth homogeneous nets, the network converges to a KKT point of the maximum margin problem in parameter space. Lyu & Li (2020) established these results with weaker assumptions and also provide asymptotic convergence rates for the loss. Chizat & Bach (2020) explore the inductive bias for a 2-layer infinitely wide ReLU neural net in function space and show that the function learnt is a max-margin classifier for variation norm.

### 2.2  NORMALIZATION

Salimans & Kingma (2016) introduced weight normalization and demonstrated that it replicates the convergence speedup of BatchNorm. Similarly, other normalization techniques have been proposed as well(Ba et al., 2016)(Qiao et al., 2020)(Li et al., 2019), but only a few have been theoretically explored. Santurkar et al. (2018) demonstrated that batch normalization makes the loss surface smoother and $L_2$ normalization in batchnorm can even be replaced by $L_1$ and $L_\infty$ normalizations. Kohler et al. (2019) showed that for GD, batchnorm speeds up convergence in the case of GLM by splitting the optimization problem into learning the direction and the norm. Cai et al. (2019) analyzed GD on BN for squared loss and showed that it converges for a wide range of lr. Bjorck et al. (2018) showed that the primary reason BN allows networks to achieve higher accuracy is by enabling higher learning rates. Arora et al. (2019) showed that in case of GD or SGD with batchnorm, lr for scale-invariant parameters does not affect the convergence rate towards stationary points. Du et al. (2018) showed that for GD over one-hidden-layer weight normalized CNN, with a constant probability over initialization, iterates converge to global minima. Qiao et al. (2019) compared different normalization techniques from the perspective of whether they lead to points, where neurons are consistently deactivated. Wu et al. (2019) established the inductive bias of gradient flow with weight normalization for overparameterized least squares and showed that for a wider range of initializations as compared to normal parameterization, it converges to the minimum $L_2$ norm solution. Dukler et al. (2020) analyzed weight normalization for multilayer ReLU net in the infinite width regime and showed that it may speedup convergence. Some other papers(Luo et al., 2019; Roburin et al., 2020) also provide other perspectives to think about normalization techniques.

## 3  PROBLEM SETUP

We use a standard view of neural networks as a collection of nodes/neurons grouped by layers. Each node $u$ is associated with a weight vector $\mathbf{w}_u$, that represents the incoming weight vector for that node. In case of CNNs, weights can be shared across different nodes. $\mathbf{w}$ represents all

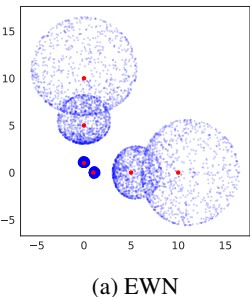
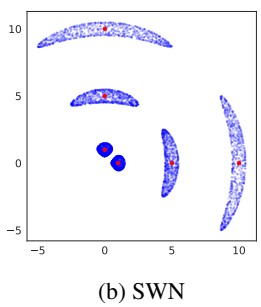
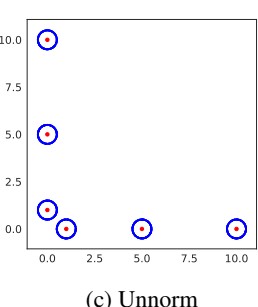

(a) EWN         (b) SWN         (c) Unnorm

Figure 1: $L_2$ neighborhoods with $\epsilon = 0.5$ radius in parameter space for different parameterizations. For EWN in the left (resp. SWN in the middle) the parameter $[\gamma, \mathbf{v}]$ (the parameter $[\alpha, \mathbf{v}]$ resp.) is restricted to a 3-d ball of radius $\epsilon$ and the values that the 2-d weight vector $\mathbf{w}$ takes is illustrated for 6 different centers.

the parameters of the network arranged in form of a vector. The dataset is represented in terms of $(\mathbf{x}_i, y_i)$ pairs and $m$ represents the number of points in the dataset. The function represented by the neural network is denoted by $\Phi(\mathbf{w}, .)$. The loss for a single data point $\mathbf{x}_i$ is given by $\ell(y_i, \Phi(\mathbf{w}, \mathbf{x}_i))$ and the loss vector is represented by $\boldsymbol{\ell}$. The overall loss is represented by $\mathcal{L}(\mathbf{w})$ and is given by $\mathcal{L}(\mathbf{w}) = \sum_{i=1}^{m} \ell(y_i, \Phi(\mathbf{w}, \mathbf{x}_i))$. We sometimes abbreviate $\mathcal{L}(\mathbf{w}(t))$ as $\mathcal{L}$ when the context is clear.

In standard weight normalisation (SWN), each weight vector $\mathbf{w}_u$ is reparameterized as $\gamma_u \frac{\mathbf{v}_u}{\|\mathbf{v}_u\|}$. This was proposed by Salimans & Kingma (2016), as a substitute for Batch Normalization and has been practically used in multiple papers such as Sokolic et al. (2017), Dauphin et al. (2017), Kim et al. (2018) and Hieber et al. (2018). The corresponding update equations for gradient descent are given by

$$\gamma_u(t+1) = \gamma_u(t) - \eta(t) \frac{\mathbf{v}_u(t)^\top \nabla_{\mathbf{w}_u} \mathcal{L}}{\|\mathbf{v}_u(t)\|} \tag{1}$$

$$\mathbf{v}_u(t+1) = \mathbf{v}_u(t) - \eta(t) \frac{\gamma_u(t)}{\|\mathbf{v}_u(t)\|} \left( I - \frac{\mathbf{v}_u(t)\mathbf{v}_u(t)^\top}{\|\mathbf{v}_u(t)\|^2} \right) \nabla_{\mathbf{w}_u} \mathcal{L} \tag{2}$$

In exponential weight normalisation (EWN), each weight vector $\mathbf{w}_u$ is reparameterized as $e^{\alpha_u} \frac{\mathbf{v}_u}{\|\mathbf{v}_u\|}$. This was mentioned in Salimans & Kingma (2016), but to the best of our knowledge, has not been widely used. The corresponding update equations for gradient descent with learning rate $\eta(t)$ are given by

$$\alpha_u(t+1) = \alpha_u(t) - \eta(t) e^{\alpha_u(t)} \frac{\mathbf{v}_u(t)^\top \nabla_{\mathbf{w}_u} \mathcal{L}}{\|\mathbf{v}_u(t)\|} \tag{3}$$

$$\mathbf{v}_u(t+1) = \mathbf{v}_u(t) - \eta(t) \frac{e^{\alpha_u(t)}}{\|\mathbf{v}_u(t)\|} \left( I - \frac{\mathbf{v}_u(t)\mathbf{v}_u(t)^\top}{\|\mathbf{v}_u(t)\|^2} \right) \nabla_{\mathbf{w}_u} \mathcal{L} \tag{4}$$

The update equations for gradient flow are the continuous counterparts for the same. In case of gradient flow, for both SWN and EWN, we assume $\|\mathbf{v}_u(0)\| = 1$, to simplify the update equations.

## 4    INDUCTIVE BIAS OF WEIGHT NORMALIZATION

In this section, we state our main results for weight normalized smooth homogeneous models on exponential loss($\ell(y_i, \Phi(\mathbf{w}, \mathbf{x}_i) = e^{-y_i \Phi(\mathbf{w}, \mathbf{x}_i)})$. The results for cross-entropy loss and proofs have been deferred to the appendix due to space constraints. First, we state the main proposition that helps in establishing these results for EWN.

**Theorem 1.** *The gradient flow path with learning rate $\eta(t)$ for EWN and SWN are given as follows:*

$$EWN: \frac{d\mathbf{w}_u(t)}{dt} = -\eta(t)\|\mathbf{w}_u(t)\|^2 \nabla_{\mathbf{w}_u} \mathcal{L} \tag{5}$$

$$SWN: \frac{d\mathbf{w}_u(t)}{dt} = -\eta(t)(\|\mathbf{w}_u(t)\|^2 \nabla_{\mathbf{w}_u} \mathcal{L} + \left( \frac{1 - \|\mathbf{w}_u(t)\|^2}{\|\mathbf{w}_u(t)\|^2} \right) (\mathbf{w}_u(t)^\top \nabla_{\mathbf{w}_u} \mathcal{L}) \mathbf{w}_u(t)) \tag{6}$$

Thus, the gradient flow path of EWN can be replicated by an adaptive learning rate given by $\eta(t)\|\mathbf{w}_u(t)\|^2$ on unnormalized network(Unnorm). These parameterizations also induce different neighborhoods in the parameter space, that have been shown in Figure 1.

## 4.1 Assumptions

The assumptions in the paper can be broadly divided into loss function/architecture based assumptions and trajectory based assumptions. The loss functions/architecture based assumptions are shared across both gradient flow and gradient descent.

**Loss function/Architecture based assumptions**

1. $\ell(y_i, \Phi(\mathbf{w}, \mathbf{x}_i)) = e^{-y_i\Phi(\mathbf{w}, \mathbf{x}_i)}$
2. $\Phi(., \mathbf{x})$ is a $\mathcal{C}^2$ function, for a fixed $\mathbf{x}$
3. $\Phi(\lambda\mathbf{w}, \mathbf{x}) = \lambda^L\Phi(\mathbf{w}, \mathbf{x})$, for some $\lambda > 0$ and $L > 0$

**Gradient flow**. For gradient flow, we make the following trajectory based assumptions

(A1) $\lim_{t\to\infty} \mathcal{L}(\mathbf{w}(t)) = 0$

(A2) $\lim_{t\to\infty} \frac{\mathbf{w}(t)}{\|\mathbf{w}(t)\|} := \widetilde{\mathbf{w}}$

(A3) $\lim_{t\to\infty} \frac{\boldsymbol{\ell}(\mathbf{w}(t))}{\|\boldsymbol{\ell}(\mathbf{w}(t))\|} := \widetilde{\boldsymbol{\ell}}$

(A4) Let $\rho = \min_i y_i\Phi(\widetilde{\mathbf{w}}, \mathbf{x}_i)$. Then $\rho > 0$.

The first assumption is typically satisfied in scenarios where a positively homogeneous network achieves 100% training accuracy. This is not a completely unreasonable assumption, given recent papers demonstrating neural networks with sufficient overparameterization can fit even random labels(Zhang et al. (2017), Jacot et al. (2018)), and is a standard assumption made when the purpose is to find the inductive bias. The second assumption states that the network converges in direction and this has been recently shown in Ji & Telgarsky (2020) to hold for gradient flow on homogeneous neural nets without normalization under some regularity assumptions. The third and fourth assumptions are required to show convergence of the gradients in direction. The fourth assumption is indeed true for SWN as shown in Lyu & Li (2020) [1].

**Gradient Descent**. For gradient descent, we also require the learning rate $\eta(t)$ to not grow too fast.

(A5) $\lim_{t\to\infty} \eta(t)\|\mathbf{w}_u(t)\|\|\nabla_{\mathbf{w}_u}\mathcal{L}(\mathbf{w}(t))\| = 0$ for all $u$ in the network

**Proposition 1.** *Under assumptions (A1)-(A4),* $\lim_{t\to\infty} \eta(t)\|\mathbf{w}_u(t)\|\|\nabla_{\mathbf{w}_u}\mathcal{L}(\mathbf{w}(t))\| = 0$ *holds for every $u$ in the network with $\eta(t) = O(\frac{1}{\mathcal{L}^c})$, where $c < 1$.*

This proposition establishes that the assumption (A5) is mild and holds for constant $\eta(t)$, that is generally used in practice.

While some of these assumptions are non-standard we believe they do generally hold, and demonstrate the viability of these assumptions in a toy experiment which we call `Lin-Sep`. In this experiment a 2-layered EWN neural network, with 8 neurons in the hidden layer and a ReLU-squared activation function, is trained on a linearly separable dataset. The learning rate schedule used was $O(\frac{1}{\mathcal{L}^{0.97}})$ and the network was trained till a loss of $e^{-300}$. The corresponding graphs for EWN are shown in Figure 2. Similar results for SWN have been deferred to Figure 8 in the appendix.

## 4.2 Effect of Normalisation on Weight and Gradient Norms

This section contains the main theorems and the difference between EWN and SWN that makes EWN asymptotically relatively sparse as compared to SWN. First, we will state a common proposition for both SWN and EWN.

**Proposition 2.** *Under assumptions (A1)-(A4) for gradient flow and (A1)-(A5) for gradient descent, for both SWN and EWN, the following hold:*

(i) $\lim_{t\to\infty} \frac{-\nabla_{\mathbf{w}}\mathcal{L}(\mathbf{w}(t))}{\|\nabla_{\mathbf{w}}\mathcal{L}(\mathbf{w}(t))\|} = \mu\sum_{i=1}^m \widetilde{\ell}_i y_i \nabla_{\mathbf{w}}\Phi(\widetilde{\mathbf{w}}, \mathbf{x}_i) = \widetilde{\mathbf{g}}$, *where $\mu > 0$.*

---

[1] Homogeneous networks in the $\mathbf{w}$ space are also homogeneous in the $\gamma, \mathbf{v}$ space. Therefore results regarding convergence rates and monotonic margin hold from Lyu & Li (2020). However, the results for convergence to a KKT point of the max margin problem do not hold. For details, refer Appendix K.

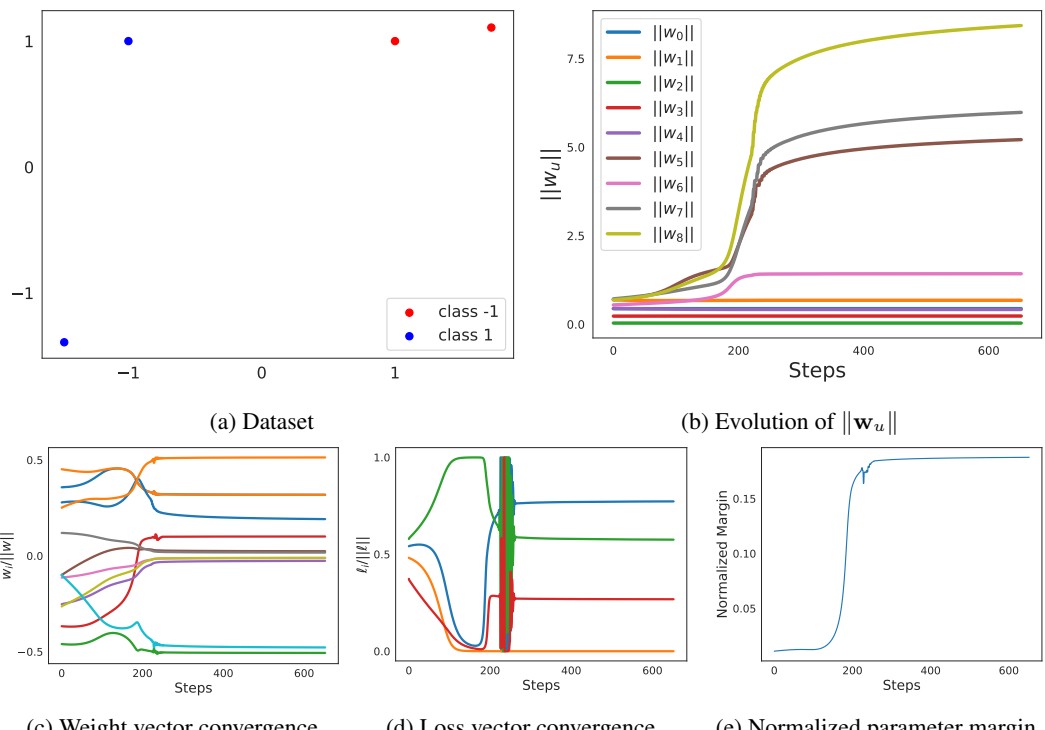

(a) Dataset          (b) Evolution of $\|\mathbf{w}_u\|$

(c) Weight vector convergence     (d) Loss vector convergence     (e) Normalized parameter margin

Figure 2: **Verification of assumptions for EWN in `Lin-Sep` experiment:** (a) shows the dataset. In (b), it can be seen that only weights 5,7 and 8 keep on growing in norm. So, only for these, $\|\widetilde{\mathbf{w}}_u\| > 0$. (c) shows the components of the unit vector $\frac{\mathbf{w}}{\|\mathbf{w}\|}$, only for the weights 5, 7 and 8 as they keep evolving with time. Eventually their contribution to the unit vector become constant. (d) shows the components of the loss vector and they also become constant eventually. (e) shows the normalized parameter margin converging to a value greater than 0.

> (ii) Let $\widetilde{\mathbf{w}}_u = \lim_{t\to\infty} \frac{\mathbf{w}_u(t)}{\|\mathbf{w}(t)\|}$ and $\widetilde{\mathbf{g}}_u = \lim_{t\to\infty} \frac{-\nabla_{\mathbf{w}_u}\mathcal{L}(\mathbf{w}(t))}{\|\nabla_{\mathbf{w}}\mathcal{L}(\mathbf{w}(t))\|}$. Then, $\widetilde{\mathbf{w}}_u = \lambda\widetilde{\mathbf{g}}_u$ for some $\lambda \geq 0$

The first and second part state that under the given assumptions, for both SWN and EWN, gradients converge in direction and the weights that contribute to the final direction of $\mathbf{w}$, converge in opposite direction of the gradients. Now, we provide the main theorem that distinguishes SWN and EWN.

**Theorem 2.** *Under assumptions (A1)-(A4) for gradient flow and (A1)-(A5) for gradient descent, the following hold*

> (i) *for EWN,* $\|\widetilde{\mathbf{w}}_u\| > 0, \|\widetilde{\mathbf{w}}_v\| > 0 \implies \lim_{t\to\infty} \frac{\|\mathbf{w}_u(t)\|\|\nabla_{\mathbf{w}_u}\mathcal{L}(\mathbf{w}(t))\|}{\|\mathbf{w}_v(t)\|\|\nabla_{\mathbf{w}_v}\mathcal{L}(\mathbf{w}(t))\|} = \frac{\|\widetilde{\mathbf{w}}_u\|\|\widetilde{\mathbf{g}}_u\|}{\|\widetilde{\mathbf{w}}_v\|\|\widetilde{\mathbf{g}}_v\|} = 1$

> (ii) *for SWN,* $\|\widetilde{\mathbf{w}}_u\| > 0, \|\widetilde{\mathbf{w}}_v\| > 0 \implies \lim_{t\to\infty} \frac{\|\mathbf{w}_u(t)\|\|\nabla_{\mathbf{w}_v}\mathcal{L}(\mathbf{w}(t))\|}{\|\mathbf{w}_v(t)\|\|\nabla_{\mathbf{w}_u}\mathcal{L}(\mathbf{w}(t))\|} = \frac{\|\widetilde{\mathbf{w}}_u\|\|\widetilde{\mathbf{g}}_v\|}{\|\widetilde{\mathbf{w}}_v\|\|\widetilde{\mathbf{g}}_u\|} = 1$

Thus, asymptotically, for EWN, $\|\mathbf{w}_u(t)\| = \frac{k_1(t)}{\|\nabla_{\mathbf{w}_u}\mathcal{L}(\mathbf{w}(t))\|}$ while for SWN, $\|\mathbf{w}_u(t)\| = k_2(t)\|\nabla_{\mathbf{w}_u}\mathcal{L}(\mathbf{w}(t))\|$, where $k_1(t)$ and $k_2(t)$ are independent of the neuron $u$. We demonstrate this property of EWN on the `Lin-Sep` experiment in Figure 3. The results for SWN have been deferred to Figure 9 in the Appendix.

Now, we provide a corollary for the case of multilayer linear nets.

**Corollary 1.** *Consider a weight normalized(SWN or EWN) multilayer linear net, represented by $y = W_n W_{n-1}...W_1 x$. Under assumptions (A1)-(A4) for gradient flow and (A1)-(A5) for gradient descent, if the dataset is sampled from a continuous distribution w.r.t $\mathbb{R}^d$, then, with probability 1, $\boldsymbol{\theta} = W_1^\top W_2^\top....W_n^\top$ converges in direction to the maximum margin separator for all linearly separable datasets.*

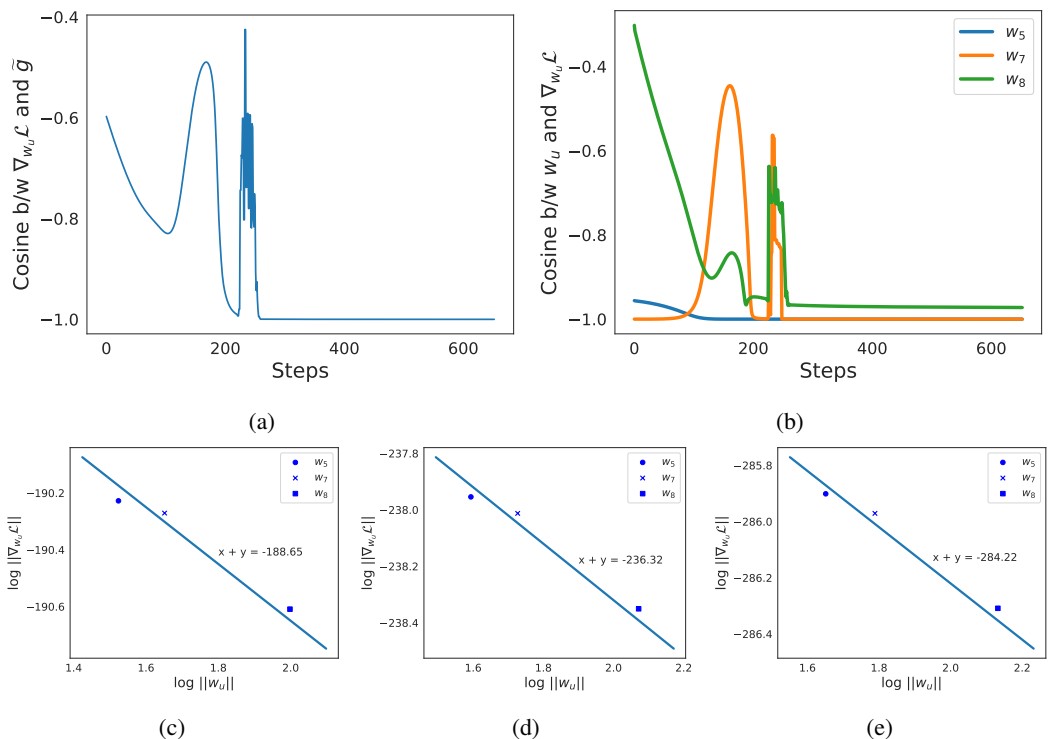

Figure 3: **Demonstration of Results for EWN in `Lin-Sep` experiment:** (a) demonstrates part 1 of Proposition 2, where $\widetilde{\mathbf{g}}$ is approximated by using $\mathbf{w}$ from the last point of the trajectory. Clearly, $\nabla_{\mathbf{w}_u}\mathcal{L}$ stops oscillating and converges to $\widetilde{\mathbf{g}}$. (b) demonstrates part 2 of Proposition 2 and shows that for weight vectors 5,7 and 8, $\mathbf{w}_u(t)$ converges in opposite direction of $\nabla_{\mathbf{w}_u}\mathcal{L}(\mathbf{w}(t))$. (c), (d) and (e) demonstrate Theorem 2 for EWN, where for weight vectors 5,7 and 8. The three graphs are plotted at loss values of $e^{-200}, e^{-250}$ and $e^{-300}$ respectively. At each loss value, for the 3 weights, $\log\|\mathbf{w}_u\| + \log\|\nabla_{\mathbf{w}_u}\mathcal{L}\|$ is approximately same.

## 4.3 SPARSITY INDUCTIVE BIAS FOR EXPONENTIAL WEIGHT NORMALISATION

The inverse relation between $\|\mathbf{w}_u(t)\|$ and $\|\nabla_{\mathbf{w}_u}\mathcal{L}(\mathbf{w}(t))\|$ in the EWN trajectory results in an interesting inductive bias that favours movement along sparse directions.

**Proposition 3.** *Let assumptions (A1)-(A5) be satisfied. Consider two nodes $u$ and $v$ in the network such that $\|\widetilde{\mathbf{g}}_v\| \geq \|\widetilde{\mathbf{g}}_u\| > 0$ and $\|\mathbf{w}_u(t)\|, \|\mathbf{w}_v(t)\| \to \infty$. Let $\frac{\|\widetilde{\mathbf{g}}_u\|}{\|\widetilde{\mathbf{g}}_v\|}$ be denoted by $c$. Let $\epsilon, \delta$ be such that $0 < \epsilon < c$ and $0 < \delta < 2\pi$. Then, the following holds:*

1. *There exists a time $t_1$, such that for all $t > t_1$ both SWN and EWN trajectories have the following properties:*

    (a) $\frac{\|\nabla_{\mathbf{w}_u}\mathcal{L}(\mathbf{w}(t))\|}{\|\nabla_{\mathbf{w}_v}\mathcal{L}(\mathbf{w}(t))\|} \in [c - \epsilon, c + \epsilon]$

    (b) $\left(\frac{\mathbf{w}_u(t)}{\|\mathbf{w}_u(t)\|}\right)^{\top}\left(\frac{-\nabla_{\mathbf{w}_u}\mathcal{L}(\mathbf{w}(t))}{\|\nabla_{\mathbf{w}_u}\mathcal{L}(\mathbf{w}(t))\|}\right) \geq \cos(\delta)$

    (c) $\left(\frac{\mathbf{w}_v(t)}{\|\mathbf{w}_v(t)\|}\right)^{\top}\left(\frac{-\nabla_{\mathbf{w}_v}\mathcal{L}(\mathbf{w}(t))}{\|\nabla_{\mathbf{w}_v}\mathcal{L}(\mathbf{w}(t))\|}\right) \geq \cos(\delta).$

2. *for SWN, $\lim_{t\to\infty}\frac{\|\mathbf{w}_u(t)\|}{\|\mathbf{w}_v(t)\|} = c$*

3. *for EWN, if at some time $t_2 > t_1$,*

    (a) $\frac{\|\mathbf{w}_u(t_2)\|}{\|\mathbf{w}_v(t_2)\|} > \frac{1}{(c-\epsilon)\cos(\delta)} \implies \lim_{t\to\infty}\frac{\|\mathbf{w}_u(t)\|}{\|\mathbf{w}_v(t)\|} = \infty$

    (b) $\frac{\|\mathbf{w}_u(t_2)\|}{\|\mathbf{w}_v(t_2)\|} < \frac{\cos(\delta)}{c+\epsilon} \implies \lim_{t\to\infty}\frac{\|\mathbf{w}_u(t)\|}{\|\mathbf{w}_v(t)\|} = 0$

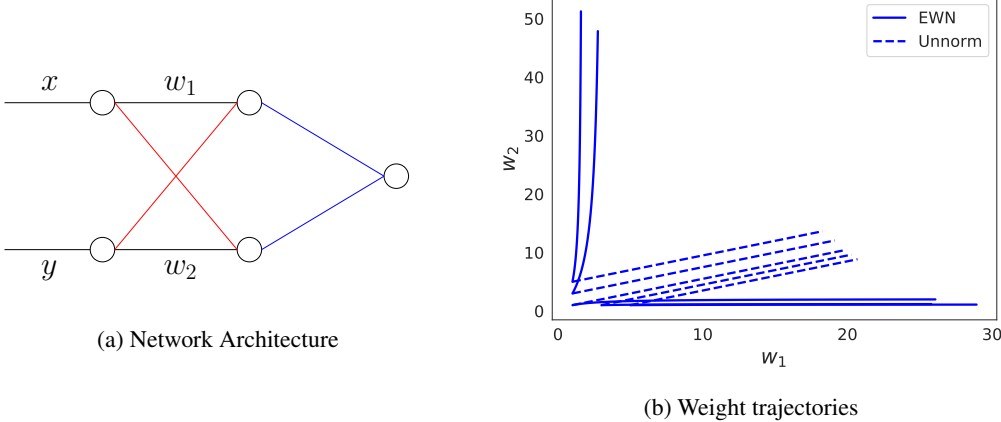

(a) Network Architecture

(b) Weight trajectories

Figure 4: (a) Network architecture for the `Simple-Traj` experiment . (b) Trajectories of the two weights for EWN and Unnorm, starting from 5 different initialization points.

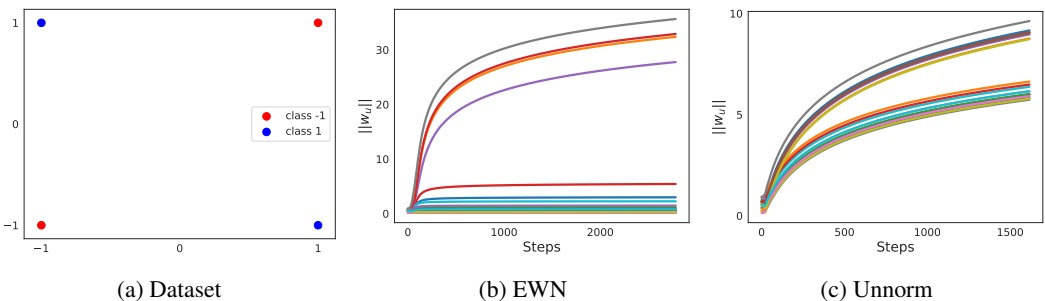

(a) Dataset            (b) EWN            (c) Unnorm

Figure 5: (a) Training data for the `XOR` experiment. (b, c) Norm of the incoming neuron weights for the EWN and unnormalized architectures.

The above proposition shows that the limit property of the weights in Theorem 2, makes non-sparse $\mathbf{w}$ an unstable convergent direction for EWN. But that is not the case for SWN. We demonstrate the relative sparsity between EWN, SWN and Unnorm through two toy experiments – `Simple-Traj` and `XOR`.

In the `Simple-Traj` experiment, we have a single data point at $(2, 1)$, that is labelled positive and train a network with linear activations. The architecture is shown in Figure 4a, where weights in blue and red are frozen to values $1$ and $0$ respectively. Thus, there are effectively only two scalar parameters- $w_1$ and $w_2$. The network is trained till a loss value of $e^{-50}$ starting from 5 different initialization points. The weight trajectories in Figure 4b shows that EWN prefers to converge either along the x or y axis, and hence has an asymptotic relative sparsity property.

In the `XOR` experiment, we train a 2-layer ReLU network, with 20 hidden neurons on XOR dataset(shown in Figure 5a). The second layer is fixed to the values 1 or -1 randomly. For attaining 100% accuracy on this dataset with this architecture, at least 4 hidden units are needed. As can be seen in Figure 5, EWN asymptotically uses exactly 4 neurons out of 20, while Unnorm uses all the 20 neurons. The results for SWN have been deferred to Figure 10 in the appendix.

## 5   CONVERGENCE RATES

Under Assumption (A2), $\mathbf{w}$ can be represented as $\mathbf{w} = g(t)\widetilde{\mathbf{w}} + \mathbf{r}(t)$, where $\lim_{t \to \infty} \frac{\|\mathbf{r}(t)\|}{g(t)} = 0$. Let $d : \mathbb{N} \to \mathbb{R}$, given by $d(t) = \sum_{\tau=0}^{t} \eta(\tau)$ denote total step size.

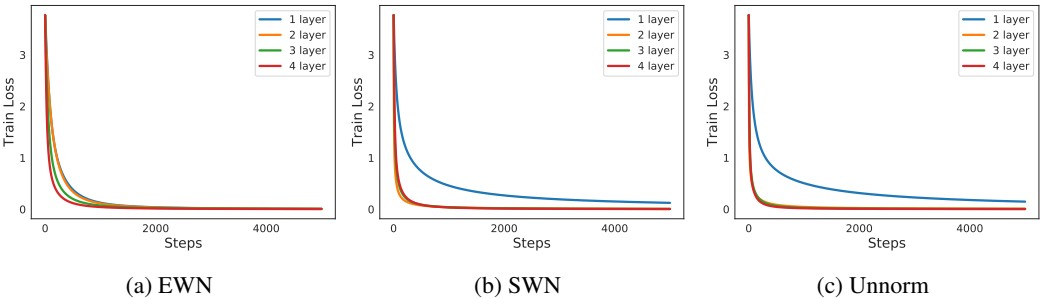

(a) EWN            (b) SWN            (c) Unnorm

Figure 6: Variation of convergence rate of train loss with number of layers for multilayer linear nets

The asymptotic convergence rate of loss for SWN and Unnorm have already been established in Lyu & Li (2020) as $\Theta\left(\frac{1}{d(t)(\log d(t))^{2-\frac{2}{L}}}\right)$. For EWN, the corresponding theorem is provided below

**Theorem 3.** *For EWN, under Assumptions (A1)-(A5) and $\lim_{t\to\infty} \frac{\|\mathbf{r}(t+1)-\mathbf{r}(t)\|}{g(t+1)-g(t)} = 0$, the following hold*

1. *$\|\mathbf{w}(t)\|$ asymptotically grows at $\Theta\left((\log(d(t))^{\frac{1}{L}}\right)$*

2. *$\mathcal{L}(\mathbf{w}(t))$ asymptotically goes down at the rate of $\Theta\left(\frac{1}{d(t)(\log d(t))^2}\right)$.*

For multilayer linear nets, the variation of convergence rate with number of layers for a linearly separable dataset is illustrated in Figure 6. All of these networks were explicitly initialized to represent the same point in function space. It can be seen that EWN, SWN and unnormalized networks all converge faster with more layers, but the effect is much less pronounced for EWN.

## 6 MNIST PRUNING EXPERIMENTS

As EWN leads to asymptotically sparse solutions, it is likely that a sufficiently trained EWN network would be comparatively robust to pruning. In this section, we compare the pruning efficacy of EWN, SWN and Unnorm on a 2-layer ReLU network trained on the MNIST dataset. In case of EWN and SWN, only the first layer is weight normalized as only this layer needs to be pruned. The pruning criterion used is the difference between the initial and final weight norm, i.e, the weights that grow the least in norm are pruned first. The corresponding pruning graphs at different loss values are shown in Figure 7. It can be seen that when the loss levels are sufficiently low, the EWN network becomes better adapted for pruning, significantly outperforming SWN and the unnormalized network in terms of test accuracy for a given level of pruning. The variation of norm of the weight vectors with gradient descent steps for neurons in the first layer has been deferred to Figure 11 in the appendix.

## 7 CONCLUSION

In this paper, we analyze the inductive bias of weight normalization for smooth homogeneous neural nets and show that exponential weight normalization is likely to lead to asymptotically sparse solutions and has a faster convergence rate than unnormalized or standard weight normalized networks.

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

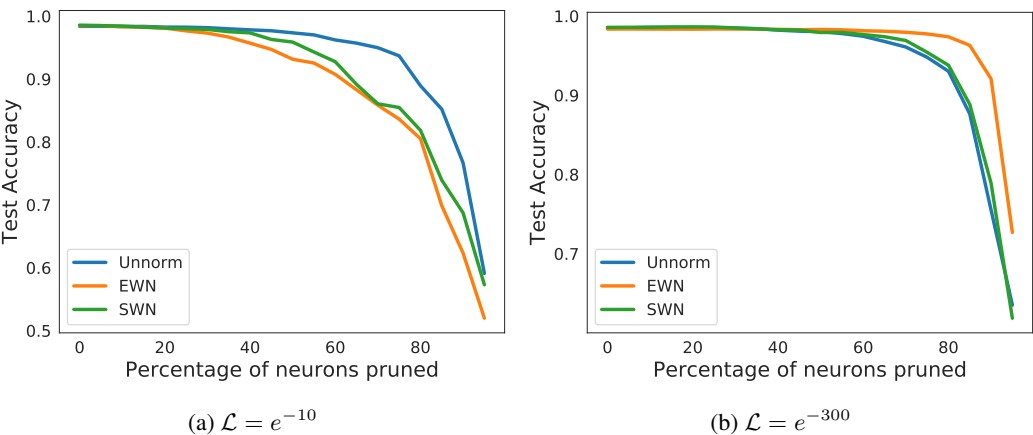

(a) $\mathcal{L} = e^{-10}$          (b) $\mathcal{L} = e^{-300}$

Figure 7: Variation of test accuracy vs percentage of neurons pruned in first layer at different loss values for MNIST experiment

Nils Bjorck, Carla P Gomes, Bart Selman, and Kilian Q Weinberger. Understanding batch normalization. In S. Bengio, H. Wallach, H. Larochelle, K. Grauman, N. Cesa-Bianchi, and R. Garnett (eds.), *Advances in Neural Information Processing Systems 31*, pp. 7694–7705. Curran Associates, Inc., 2018. URL http://papers.nips.cc/paper/7996-understanding-batch-normalization.pdf.

Yongqiang Cai, Qianxiao Li, and Zuowei Shen. A quantitative analysis of the effect of batch normalization on gradient descent. volume 97 of *Proceedings of Machine Learning Research*, pp. 882–890, Long Beach, California, USA, 09–15 Jun 2019. PMLR. URL http://proceedings.mlr.press/v97/cai19a.html.

Lénaïc Chizat and Francis Bach. Implicit bias of gradient descent for wide two-layer neural networks trained with the logistic loss. volume 125 of *Proceedings of Machine Learning Research*, pp. 1305–1338. PMLR, 09–12 Jul 2020. URL http://proceedings.mlr.press/v125/chizat20a.html.

Yann N. Dauphin, Angela Fan, Michael Auli, and David Grangier. Language modeling with gated convolutional networks. volume 70 of *Proceedings of Machine Learning Research*, pp. 933–941, International Convention Centre, Sydney, Australia, 06–11 Aug 2017. PMLR. URL http://proceedings.mlr.press/v70/dauphin17a.html.

Simon Du, Jason Lee, Yuandong Tian, Aarti Singh, and Barnabas Poczos. Gradient descent learns one-hidden-layer CNN: Don't be afraid of spurious local minima. volume 80 of *Proceedings of Machine Learning Research*, pp. 1339–1348, Stockholmsmässan, Stockholm Sweden, 10–15 Jul 2018. PMLR. URL http://proceedings.mlr.press/v80/du18b.html.

Yonatan Dukler, Quanquan Gu, and Guido Montúfar. Optimization theory for relu neural networks trained with normalization layers. In *International Conference on Machine Learning*, 2020.

Suriya Gunasekar, Jason Lee, Daniel Soudry, and Nathan Srebro. Characterizing implicit bias in terms of optimization geometry. volume 80 of *Proceedings of Machine Learning Research*, pp. 1832–1841, Stockholmsmässan, Stockholm Sweden, 10–15 Jul 2018a. PMLR. URL http://proceedings.mlr.press/v80/gunasekar18a.html.

Suriya Gunasekar, Jason D Lee, Daniel Soudry, and Nati Srebro. Implicit bias of gradient descent on linear convolutional networks. In S. Bengio, H. Wallach, H. Larochelle, K. Grauman, N. Cesa-Bianchi, and R. Garnett (eds.), *Advances in Neural Information Processing Systems 31*, pp. 9461–9471. Curran Associates, Inc., 2018b. URL http://papers.nips.cc/paper/8156-implicit-bias-of-gradient-descent-on-linear-convolutional-networks.pdf.

Felix Hieber, Tobias Domhan, Michael Denkowski, David Vilar, Artem Sokolov, Ann Clifton, and Matt Post. Sockeye: A toolkit for neural machine translation, 2018.

Arthur Jacot, Franck Gabriel, and Clement Hongler. Neural tangent kernel: Convergence and generalization in neural networks. In S. Bengio, H. Wallach, H. Larochelle, K. Grauman, N. Cesa-Bianchi, and R. Garnett (eds.), *Advances in Neural Information Processing Systems 31*, pp. 8571–8580. Curran Associates, Inc., 2018. URL http://papers.nips.cc/paper/ 8076-neural-tangent-kernel-convergence-and-generalization-in-neural-networks. pdf.

Ziwei Ji and Matus Telgarsky. The implicit bias of gradient descent on nonseparable data. volume 99 of *Proceedings of Machine Learning Research*, pp. 1772–1798, Phoenix, USA, 25–28 Jun 2019a. PMLR. URL http://proceedings.mlr.press/v99/ji19a.html.

Ziwei Ji and Matus Telgarsky. A refined primal-dual analysis of the implicit bias, 2019b.

Ziwei Ji and Matus Telgarsky. Gradient descent aligns the layers of deep linear networks. In *International Conference on Learning Representations*, 2019c. URL https://openreview. net/forum?id=HJflg30qKX.

Ziwei Ji and Matus Telgarsky. Directional convergence and alignment in deep learning. *arXiv preprint arXiv:2006.06657*, 2020.

Jin-Hwa Kim, Jaehyun Jun, and Byoung-Tak Zhang. Bilinear attention networks. In S. Bengio, H. Wallach, H. Larochelle, K. Grauman, N. Cesa-Bianchi, and R. Garnett (eds.), *Advances in Neural Information Processing Systems*, volume 31, pp. 1564–1574. Curran Associates, Inc., 2018. URL https://proceedings.neurips.cc/paper/2018/file/ 96ea64f3a1aa2fd00c72faacf0cb8ac9-Paper.pdf.

Jonas Kohler, Hadi Daneshmand, Aurelien Lucchi, Thomas Hofmann, Ming Zhou, and Klaus Neymeyr. Exponential convergence rates for batch normalization: The power of length-direction decoupling in non-convex optimization. volume 89 of *Proceedings of Machine Learning Research*, pp. 806–815. PMLR, 16–18 Apr 2019. URL http://proceedings.mlr.press/ v89/kohler19a.html.

Boyi Li, Felix Wu, Kilian Q Weinberger, and Serge Belongie. Positional normalization. In H. Wallach, H. Larochelle, A. Beygelzimer, F. d'Alché Buc, E. Fox, and R. Garnett (eds.), *Advances in Neural Information Processing Systems 32*, pp. 1622–1634. Curran Associates, Inc., 2019. URL http://papers.nips.cc/paper/8440-positional-normalization.pdf.

Ping Luo, Xinjiang Wang, Wenqi Shao, and Zhanglin Peng. Towards understanding regularization in batch normalization. In *International Conference on Learning Representations*, 2019. URL https://openreview.net/forum?id=HJlLKjR9FQ.

Kaifeng Lyu and Jian Li. Gradient descent maximizes the margin of homogeneous neural networks. In *International Conference on Learning Representations*, 2020. URL https:// openreview.net/forum?id=SJeLIgBKPS.

M. Muresan. *A Concrete Approach to Classical Analysis*. CMS Books in Mathematics. Springer New York, 2015. ISBN 9780387789330. URL https://books.google.co.in/books? id=N8rBgtIu_qgC.

Mor Shpigel Nacson, Suriya Gunasekar, Jason Lee, Nathan Srebro, and Daniel Soudry. Lexicographic and depth-sensitive margins in homogeneous and non-homogeneous deep models. volume 97 of *Proceedings of Machine Learning Research*, pp. 4683–4692, Long Beach, California, USA, 09–15 Jun 2019a. PMLR. URL http://proceedings.mlr.press/v97/ nacson19a.html.

Mor Shpigel Nacson, Jason Lee, Suriya Gunasekar, Pedro Henrique Pamplona Savarese, Nathan Srebro, and Daniel Soudry. Convergence of gradient descent on separable data. volume 89 of *Proceedings of Machine Learning Research*, pp. 3420–3428. PMLR, 16–18 Apr 2019b. URL http://proceedings.mlr.press/v89/nacson19b.html.

Mor Shpigel Nacson, Nathan Srebro, and Daniel Soudry. Stochastic gradient descent on separable data: Exact convergence with a fixed learning rate. volume 89 of *Proceedings of Machine Learning Research*, pp. 3051–3059. PMLR, 16–18 Apr 2019c. URL http://proceedings.mlr.press/v89/nacson19a.html.

Siyuan Qiao, Huiyu Wang, Chenxi Liu, Wei Shen, and Alan Yuille. Rethinking normalization and elimination singularity in neural networks. *arXiv preprint arXiv:1911.09738*, 2019.

Siyuan Qiao, Huiyu Wang, Chenxi Liu, Wei Shen, and Alan Yuille. Micro-batch training with batch-channel normalization and weight standardization, 2020.

Simon Roburin, Yann de Mont-Marin, Andrei Bursuc, Renaud Marlet, Patrick Pérez, and Mathieu Aubry. Spherical perspective on learning with batch norm. *arXiv preprint arXiv:2006.13382*, 2020.

Tim Salimans and Durk P Kingma. Weight normalization: A simple reparameterization to accelerate training of deep neural networks. In D. D. Lee, M. Sugiyama, U. V. Luxburg, I. Guyon, and R. Garnett (eds.), *Advances in Neural Information Processing Systems 29*, pp. 901–909. Curran Associates, Inc., 2016. URL http://papers.nips.cc/paper/6114-weight-normalization-a-simple-reparameterization-to-accelerate-training-of-de pdf.

Shibani Santurkar, Dimitris Tsipras, Andrew Ilyas, and Aleksander Madry. How does batch normalization help optimization? In S. Bengio, H. Wallach, H. Larochelle, K. Grauman, N. Cesa-Bianchi, and R. Garnett (eds.), *Advances in Neural Information Processing Systems 31*, pp. 2483–2493. Curran Associates, Inc., 2018. URL http://papers.nips.cc/paper/7515-how-does-batch-normalization-help-optimization.pdf.

Jure Sokolic, Raja Giryes, Guillermo Sapiro, and Miguel R. D. Rodrigues. Robust large margin deep neural networks. *IEEE Transactions on Signal Processing*, 65(16):4265–4280, Aug 2017. ISSN 1941-0476. doi: 10.1109/tsp.2017.2708039. URL http://dx.doi.org/10.1109/TSP.2017.2708039.

Daniel Soudry, Elad Hoffer, and Nathan Srebro. The implicit bias of gradient descent on separable data. In *International Conference on Learning Representations*, 2018. URL https://openreview.net/forum?id=r1q7n9gAb.

Xiaoxia Wu, Edgar Dobriban, Tongzheng Ren, Shanshan Wu, Zhiyuan Li, Suriya Gunasekar, Rachel Ward, and Qiang Liu. Implicit regularization of normalization methods. *arXiv preprint arXiv:1911.07956*, 2019.

Chiyuan Zhang, Samy Bengio, Moritz Hardt, Benjamin Recht, and Oriol Vinyals. Understanding deep learning requires rethinking generalization. In *International Conference on Learning Representations*, 2017. URL https://openreview.net/forum?id=Sy8gdB9xx.

## A  PROOF OF THEOREM 1

**Theorem.** *The gradient flow path with learning rate $\eta(t)$ for EWN and SWN are given as follows:*

$$EWN: \frac{d\mathbf{w}_u(t)}{dt} = -\eta(t)\|\mathbf{w}_u(t)\|^2 \nabla_{\mathbf{w}_u}\mathcal{L}$$

$$SWN: \frac{d\mathbf{w}_u(t)}{dt} = -\eta(t)(\|\mathbf{w}_u(t)\|^2 \nabla_{\mathbf{w}_u}\mathcal{L} + \left(\frac{1-\|\mathbf{w}_u(t)\|^2}{\|\mathbf{w}_u(t)\|^2}\right)(\mathbf{w}_u(t)^\top \nabla_{\mathbf{w}_u}\mathcal{L})\mathbf{w}_u(t))$$

The proof for the two parts will be provided in different subsections, where the corresponding part will be restated for ease of the reader.

### A.1 EXPONENTIAL WEIGHT NORMALIZATION

**Theorem.** *The gradient flow path with learning rate $\eta(t)$ for EWN is given by:*

$$\frac{d\mathbf{w}_u(t)}{dt} = -\eta(t)\|\mathbf{w}_u(t)\|^2 \nabla_{\mathbf{w}_u}\mathcal{L}$$

*Proof.* In case of EWN, weights are reparameterized as $\mathbf{w}_u = e^{\alpha_u}\frac{\mathbf{v}_u}{\|\mathbf{v}_u\|}$. Then

$$\nabla_{\alpha_u}\mathcal{L} = e^{\alpha_u}\frac{\mathbf{v}_u^\top \nabla_{\mathbf{w}_u}\mathcal{L}}{\|\mathbf{v}_u\|}$$

$$\nabla_{\mathbf{v}_u}\mathcal{L} = \frac{e^{\alpha_u}}{\|\mathbf{v}_u\|}(I - \frac{\mathbf{v}_u\mathbf{v}_u^\top}{\|\mathbf{v}_u\|^2})\nabla_{\mathbf{w}_u}\mathcal{L}$$

Now, in case of gradient flow with learning rate $\eta(t)$, we can say

$$\frac{d\alpha_u(t)}{dt} = -\eta(t)\nabla_{\alpha_u}\mathcal{L} = -\eta(t)e^{\alpha_u(t)}\frac{\mathbf{v}_u(t)^\top \nabla_{\mathbf{w}_u}\mathcal{L}}{\|\mathbf{v}_u(t)\|}$$

$$\frac{d\mathbf{v}_u(t)}{dt} = -\eta(t)\nabla_{\mathbf{v}_u}\mathcal{L} = -\eta(t)\frac{e^{\alpha_u(t)}}{\|\mathbf{v}_u(t)\|}(I - \frac{\mathbf{v}_u(t)\mathbf{v}_u(t)^\top}{\|\mathbf{v}_u(t)\|^2})\nabla_{\mathbf{w}_u}\mathcal{L}$$

Now, using these equations, we can say

$$\frac{d\|\mathbf{v}_u(t)\|^2}{dt} = 2\mathbf{v}_u(t)^\top\left(\frac{d\mathbf{v}_u(t)}{dt}\right) = 0$$

Thus, $\|\mathbf{v}_u(t)\|$ does not change with time. As we assumed $\|\mathbf{v}_u(0)\|$ to be 1, therefore for any t, $\|\mathbf{v}_u(t)\| = 1$. Using this simplification, we can write

$$\frac{d\mathbf{w}_u(t)}{dt} = \frac{d(e^{\alpha_u(t)}\mathbf{v}_u(t))}{dt}$$
$$= e^{\alpha_u(t)}(-\eta(t)e^{\alpha_u(t)}(I - \mathbf{v}_u(t)\mathbf{v}_u(t)^\top)\nabla_{\mathbf{w}_u}\mathcal{L}) - \eta(t)e^{2\alpha_u(t)}(\mathbf{v}_u(t)^\top\nabla_{\mathbf{w}_u}\mathcal{L})\mathbf{v}_u(t)$$
$$= -\eta(t)e^{2\alpha_u(t)}\nabla_{\mathbf{w}_u}\mathcal{L}$$

Thus, the gradient flow path with exponential weight normalization can be replicated by an adaptive learning rate given by $\eta(t)\|\mathbf{w}_u(t)\|^2$. $\square$

### A.2 STANDARD WEIGHT NORMALIZATION

**Theorem.** *The gradient flow path with learning rate $\eta(t)$ for SWN is given by:*

$$\frac{d\mathbf{w}_u(t)}{dt} = -\eta(t)(\|\mathbf{w}_u(t)\|^2\nabla_{\mathbf{w}_u}\mathcal{L} + \left(\frac{1 - \|\mathbf{w}_u(t)\|^2}{\|\mathbf{w}_u(t)\|^2}\right)(\mathbf{w}_u(t)^\top\nabla_{\mathbf{w}_u}\mathcal{L})\mathbf{w}_u(t))$$

*Proof.* In case of SWN, weights are reparameterized as $\mathbf{w}_u = \gamma_u\frac{\mathbf{v}_u}{\|\mathbf{v}_u\|}$. Then

$$\nabla_{\gamma_u}\mathcal{L} = \frac{\mathbf{v}_u^\top\nabla_{\mathbf{w}_u}\mathcal{L}}{\|\mathbf{v}_u\|}$$

$$\nabla_{\mathbf{v}_u}\mathcal{L} = \frac{\gamma_u}{\|\mathbf{v}_u\|}(I - \frac{\mathbf{v}_u\mathbf{v}_u^\top}{\|\mathbf{v}_u\|^2})\nabla_{\mathbf{w}_u}\mathcal{L}$$

Now, in case of gradient flow with learning rate $\eta(t)$, we can say

$$\frac{d\gamma_u(t)}{dt} = -\eta(t)\nabla_{\alpha_u}\mathcal{L} = -\eta(t)\frac{\mathbf{v}_u(t)^\top\nabla_{\mathbf{w}_u}\mathcal{L}}{\|\mathbf{v}_u(t)\|}$$

$$\frac{d\mathbf{v}_u(t)}{dt} = -\eta(t)\nabla_{\mathbf{v}_u}\mathcal{L} = -\eta(t)\frac{\gamma_u(t)}{\|\mathbf{v}_u(t)\|}(I - \frac{\mathbf{v}_u(t)\mathbf{v}_u(t)^\top}{\|\mathbf{v}_u(t)\|^2})\nabla_{\mathbf{w}_u}\mathcal{L}$$

Now, similar to EWN, $\|\mathbf{v}_u(t)\|$ does not change with time. Using the fact that $\|\mathbf{v}_u(t)\| = 1$ for all $t$, we can say

$$
\begin{aligned}
\frac{d\mathbf{w}_u(t)}{dt} &= \frac{d(\gamma_u(t)\mathbf{v}_u(t))}{dt} \\
&= \gamma_u(t)(-\eta(t)\gamma_u(t)(I - \mathbf{v}_u(t)\mathbf{v}_u(t)^\top)\nabla_{\mathbf{w}_u}\mathcal{L}) - \eta(t)(\mathbf{v}_u(t)^\top\nabla_{\mathbf{w}_u}\mathcal{L})\mathbf{v}_u(t) \\
&= -\eta(t)(\gamma_u(t)^2\nabla_{\mathbf{w}_u}\mathcal{L} + (1 - \gamma_u(t)^2)(\mathbf{v}_u(t)^\top\nabla_{\mathbf{w}_u}\mathcal{L})\mathbf{v}_u(t) \\
&= -\eta(t)(\gamma_u(t)^2\nabla_{\mathbf{w}_u}\mathcal{L} + \left(\frac{1 - \gamma_u(t)^2}{\gamma_u(t)^2}\right)(\mathbf{w}_u(t)^\top\nabla_{\mathbf{w}_u}\mathcal{L})\mathbf{w}_u(t)
\end{aligned}
$$

Replacing $\gamma_u(t)$ by $\|\mathbf{w}_u(t)\|$ gives the required expression. $\qquad\square$

## B  PROOF OF PROPOSITION 1

**Proposition.** *Under assumptions (A1)-(A4),* $\lim_{t\to\infty}\eta(t)\|\mathbf{w}_u(t)\|\|\nabla_{\mathbf{w}_u}\mathcal{L}(\mathbf{w}(t))\| = 0$ *holds for every $u$ in the network with $\eta(t) = O(\frac{1}{\mathcal{L}^c})$, where $c < 1$.*

*Proof.* Under assumption (A1) and (A2), $\mathbf{w}$ can be represented as $\mathbf{w} = g(t)\widetilde{\mathbf{w}} + \mathbf{r}(t)$, where $\lim_{t\to\infty}\frac{\|\mathbf{r}(t)\|}{g(t)} = 0$. Now, for exponential loss,

$$
-\nabla_{\mathbf{w}}\mathcal{L}(\mathbf{w}(t)) = \sum_{i=1}^m \ell_i(\mathbf{w}(t))(y_i\nabla_{\mathbf{w}}\Phi(\mathbf{w}(t), x_i))
$$

$$
\ell_i = e^{-y_i\Phi(\mathbf{w}(t), x_i)} = e^{-g(t)^L y_i\Phi(\widetilde{\mathbf{w}} + \frac{\mathbf{r}(t)}{g(t)}, x_i)}
$$

$$
\nabla_{\mathbf{w}}\Phi(\mathbf{w}(t), x_i)) = g(t)^{L-1}\nabla_{\mathbf{w}}\Phi(\widetilde{\mathbf{w}} + \frac{\mathbf{r}(t)}{g(t)}, x_i))
$$

From assumption (A4), we know $\Phi(\widetilde{\mathbf{w}}, \mathbf{x}_i) \geq \rho$ for all $i$. Now, using Euler's homogeneous theorem, we can say

$$
\widetilde{\mathbf{w}}^\top\nabla_{\mathbf{w}}\Phi(\widetilde{\mathbf{w}}, \mathbf{x}_i) = L\Phi(\widetilde{\mathbf{w}}, \mathbf{x}_i)
$$

Thus, $\|\nabla_{\mathbf{w}}\Phi(\widetilde{\mathbf{w}}, \mathbf{x}_i)\| > 0$ for all $i$. Now, using the equations above and assumption (A3), we can say

$$
\lim_{t\to\infty}\frac{\|\nabla_{\mathbf{w}}\mathcal{L}(\mathbf{w}(t))\|}{e^{-\rho g(t)^L}g(t)^{L-1}\|\sum_{i=1}^m\widetilde{\ell}_i y_i\nabla_{\mathbf{w}}\Phi(\widetilde{\mathbf{w}}, \mathbf{x}_i)\|} = k_1
$$

where $k_1$ is some constant. Now, if $\eta(t) = O(\frac{1}{\mathcal{L}(w(t))^c})$, where $c < 1$, then using the fact that $\mathcal{L}$ goes down at the rate of $e^{-\rho g(t)^L}$ and $\|\mathbf{w}\|$ goes up at the rate of $g(t)$, we can say

$$
\begin{aligned}
\lim_{t\to\infty}\eta(t)\|\mathbf{w}(t)\|\|\nabla_{\mathbf{w}}\mathcal{L}(\mathbf{w}(t))\| &\leq \lim_{t\to\infty}\frac{k_2\|\mathbf{w}(t)\|\|\nabla_{\mathbf{w}}\mathcal{L}(\mathbf{w}(t))\|}{\mathcal{L}(w(t))^c} \\
&= \lim_{t\to\infty}\frac{k_1 k_2 e^{-\rho g(t)^L}g(t)^{L-1}\|\sum_{i=1}^m\widetilde{\ell}_i y_i\nabla_{\mathbf{w}}\Phi(\widetilde{\mathbf{w}}, \mathbf{x}_i)\|\|\mathbf{w}(t)\|}{\mathcal{L}(\mathbf{w}(t))^c} \\
&= 0
\end{aligned}
$$

$\qquad\square$

## C  PROOF OF PROPOSITION 2

**Proposition.** *Under assumptions (A1)-(A4) for gradient flow and (A1)-(A5) for gradient descent, for both SWN and EWN, the following hold:*

*(i)* $\lim_{t\to\infty}\frac{-\nabla_{\mathbf{w}}\mathcal{L}(\mathbf{w}(t))}{\|\nabla_{\mathbf{w}}\mathcal{L}(\mathbf{w}(t))\|} = \mu\sum_{i=1}^m\widetilde{\ell}_i y_i\nabla_{\mathbf{w}}\Phi(\widetilde{\mathbf{w}}, \mathbf{x}_i) = \widetilde{\mathbf{g}}$, *where $\mu > 0$.*

*(ii) Let $\widetilde{\mathbf{w}}_u = \lim_{t\to\infty}\frac{\mathbf{w}_u(t)}{\|\mathbf{w}(t)\|}$ and $\widetilde{\mathbf{g}}_u = \lim_{t\to\infty}\frac{-\nabla_{\mathbf{w}_u}\mathcal{L}(\mathbf{w}(t))}{\|\nabla_{\mathbf{w}}\mathcal{L}(\mathbf{w}(t))\|}$. Then, $\widetilde{\mathbf{w}}_u = \lambda\widetilde{\mathbf{g}}_u$ for some $\lambda \geq 0$*

The proof for different cases will be split into different subsections and corresponding proposition will be stated there for ease of the reader. The proof will depend on the Stolz Cesaro theorem(stated in Appendix J),Integral Form of Stolz-Cesaro Theorem(stated and proved in Appendix J) and following lemmas that have been proved in Appendix I.

**Lemma 1.** *Under assumptions (A1)-(A4) for gradient flow and (A1)-(A5) for gradient descent, for both SWN and EWN, $\widetilde{\mathbf{w}}_u^\top \widetilde{\mathbf{g}}_u \geq 0$ for all nodes $u$ in the network.*

**Lemma 2.** *Under assumptions (A1)-(A4) for gradient flow and (A1)-(A5) for gradient descent, for both SWN and EWN, there exists atleast one node $u$ in the network satisfying $\|\widetilde{\mathbf{w}}_u\| > 0$ and $\|\widetilde{\mathbf{g}}_u\| > 0$.*

**Lemma 3.** *Consider two unit vectors $\mathbf{a}$ and $\mathbf{b}$ satisfying $\mathbf{a}^\top \mathbf{b} \geq 0$ and $\mathbf{a}^\top \mathbf{b} < 1$. Then, there exists a small enough $\epsilon > 0$, such that for any unit vector $\mathbf{c}$ satisfying $\mathbf{c}^\top \mathbf{a} \geq \cos(\epsilon)$ and any unit vector $\mathbf{d}$ satisfying $\mathbf{d}^\top \mathbf{b} \geq \cos(\epsilon)$, $\mathbf{b}^\top (I - \mathbf{c}\mathbf{c}^\top)\mathbf{d} \geq \epsilon$.*

**Lemma 4.** *Consider sequence $a$ satisfying the following properties*

1. *$a_k > 0$*

2. *$\sum_{k=0}^{\infty} a_k = \infty$*

3. *$\lim_{k \to \infty} a_k = 0$*

*Then $\sum_{k=0}^{\infty} \frac{a_k}{\sqrt{\sum_{j=0}^{k} a_j^2}} = \infty$*

**Lemma 5.** *Consider two sequences $a$ and $b$ satisfying the following properties*

1. *$a_k > 0, \sum_{k=0}^{\infty} a_k = \infty$ and $\lim_{k \to \infty} a_k = 0$*

2. *$b_0 > 0$, $b$ is increasing and $b_{k+1}^2 \leq b_k^2 + (\frac{a_k}{b_k})^2$*

*Then $\sum_{k=0}^{\infty} \frac{a_k}{b_k} = \infty$.*

**Lemma 6.** *Consider two sequences $a$ and $b$ satisfying the following properties*

1. *$a_k > 0$ and $\sum_{k=0}^{\infty} a_k = \infty$*

2. *$b_k > 0$ and $\sum_{k=0}^{\infty} b_k = \infty$*

3. *$\sum_{k=0}^{\infty}(a_k - b_k)$ converges to a finite value*

4. *$\lim_{k \to \infty} \frac{a_k}{b_k}$ exists*

*Then $\lim_{k \to \infty} \frac{a_k}{b_k} = 1$.*

## C.1 EXPONENTIAL WEIGHT NORMALIZATION

In this section, we will use $e^{\alpha_u(t)}$ and $\|\mathbf{w}_u(t)\|$ interchangeably.

### C.1.1 GRADIENT FLOW

**Proposition.** *Under assumptions (A1)-(A4) for gradient flow, for EWN, the following hold:*

(i) *$\lim_{t \to \infty} \frac{-\nabla_\mathbf{w} \mathcal{L}(\mathbf{w}(t))}{\|\nabla_\mathbf{w} \mathcal{L}(\mathbf{w}(t))\|} = \mu \sum_{i=1}^{m} \widetilde{\ell}_i y_i \nabla_\mathbf{w} \Phi(\widetilde{\mathbf{w}}, \mathbf{x}_i) = \widetilde{\mathbf{g}}$, where $\mu > 0$.*

(ii) *Let $\widetilde{\mathbf{w}}_u = \lim_{t \to \infty} \frac{\mathbf{w}_u(t)}{\|\mathbf{w}(t)\|}$ and $\widetilde{\mathbf{g}}_u = \lim_{t \to \infty} \frac{-\nabla_{\mathbf{w}_u} \mathcal{L}(\mathbf{w}(t))}{\|\nabla_\mathbf{w} \mathcal{L}(\mathbf{w}(t))\|}$. Then, $\widetilde{\mathbf{w}}_u = \lambda \widetilde{\mathbf{g}}_u$ for some $\lambda \geq 0$*

**Update Equations:**

$$\frac{d\alpha_u(t)}{dt} = -\eta(t)e^{\alpha_u(t)}(\mathbf{v}_u(t)^\top \nabla_{\mathbf{w}_u} \mathcal{L}(\mathbf{w}(t))) \tag{7}$$

$$\frac{d\mathbf{v}_u(t)}{dt} = -\eta(t)e^{\alpha_u(t)}(I - \mathbf{v}_u(t)\mathbf{v}_u(t)^\top)\nabla_{\mathbf{w}_u}\mathcal{L}(\mathbf{w}(t)) \tag{8}$$

**(i)** $\lim_{t\to\infty} \frac{-\nabla_{\mathbf{w}}\mathcal{L}(\mathbf{w}(t))}{\|\nabla_{\mathbf{w}}\mathcal{L}(\mathbf{w}(t))\|} = \mu \sum_{i=1}^m \widetilde{\ell}_i y_i \nabla_{\mathbf{w}}\Phi(\widetilde{\mathbf{w}}, \mathbf{x}_i) = \widetilde{\mathbf{g}}$, where $\mu > 0$

*Proof.* Using the fact that $\rho > 0$ and Euler's homogeneous theorem, we can say

$$\widetilde{\mathbf{w}}^\top \nabla_{\mathbf{w}}\Phi(\widetilde{\mathbf{w}}, x_i) = L\Phi(\widetilde{\mathbf{w}}, x_i) > 0$$

Thus, $\|\nabla_{\mathbf{w}}\Phi(\widetilde{\mathbf{w}}, x_i)\| > 0$ for all $i$.

Let $\mathbf{w}(t) = g(t)\widetilde{\mathbf{w}} + \mathbf{r}(t)$, where $\lim_{t\to\infty} \frac{\|\mathbf{r}(t)\|}{g(t)} = 0$. Now, by Taylor's Theorem, we can say

$$\nabla_{\mathbf{w}}\Phi(\mathbf{w}(t), x_i) = \nabla_{\mathbf{w}}\Phi(g(t)\widetilde{\mathbf{w}}, x_i) + \int_{k=0}^{k=1} \nabla^2\Phi(g(t)\widetilde{\mathbf{w}} + k\mathbf{r}(t), x_i)\mathbf{r}(t)dk$$

$$= g(t)^{L-1}\nabla_{\mathbf{w}}\Phi(\widetilde{\mathbf{w}}, x_i) + g(t)^{L-1}\int_{k=0}^{k=1} \nabla^2\Phi(\widetilde{\mathbf{w}} + k\frac{\mathbf{r}(t)}{g(t)}, x_i)\frac{\mathbf{r}(t)}{g(t)}dk$$

Now, $\nabla^2\Phi(\widetilde{\mathbf{w}} + k\frac{\mathbf{r}(t)}{g(t)}, x_i)$ can be bounded by a constant and $\lim_{t\to\infty} \frac{\|\mathbf{r}(t)\|}{g(t)} = 0$. Thus, $\lim_{t\to\infty}\int_{k=0}^{k=1} \nabla^2\Phi(\widetilde{\mathbf{w}} + k\frac{\mathbf{r}(t)}{g(t)}, x_i)\frac{\mathbf{r}(t)}{g(t)}dk = 0$. Thus, we can say, if $\|\nabla_{\mathbf{w}}\Phi(\widetilde{\mathbf{w}}, x_i)\| > 0$ for some $i$, then

$$\lim_{t\to\infty} \frac{\nabla_{\mathbf{w}}\Phi(\mathbf{w}(t), x_i)}{g(t)^{L-1}} = \nabla_{\mathbf{w}}\Phi(\widetilde{\mathbf{w}}, x_i) \tag{9}$$

Now,

$$-\nabla_{\mathbf{w}}\mathcal{L}(\mathbf{w}(t)) = \sum_{i=1}^m \ell_i(\mathbf{w}(t))(y_i\nabla_{\mathbf{w}}\Phi(\mathbf{w}(t), x_i))$$

Now, Let $S = \{i : y_i\Phi(\widetilde{\mathbf{w}}, x_i) = \min_j y_j\Phi(\widetilde{\mathbf{w}}, x_j)\}$. Let $\epsilon$ denote $\min_{j\notin S} y_j\Phi(\widetilde{\mathbf{w}}, x_j) - \rho$. Consider $a \in S$ and $b \notin S$, then

$$\lim_{t\to\infty} \frac{\ell_b(\mathbf{w}(t))}{\ell_a(\mathbf{w}(t)} = \lim_{t\to\infty} e^{-g(t)^L(\Phi(\widetilde{\mathbf{w}} + \frac{\mathbf{r}(t)}{g(t)}, x_a) - \Phi(\widetilde{\mathbf{w}} + \frac{\mathbf{r}(t)}{g(t)}, x_b))}$$

Now, as the minimum difference is $\epsilon$, therefore $\lim_{t\to\infty} \frac{\ell_b(\mathbf{w}(t))}{\ell_a(\mathbf{w}(t)} = 0$. Thus, $\forall j \notin S, \widetilde{\ell}_j = 0$. Now, using Equation (9) and the expression for $-\nabla_{\mathbf{w}}\mathcal{L}(\mathbf{w}(t))$ from before, we can say

$$\lim_{t\to\infty} \frac{-\nabla_{\mathbf{w}}\mathcal{L}(\mathbf{w}(t))}{\|\nabla_{\mathbf{w}}\mathcal{L}(\mathbf{w}(t))\|} = \mu \sum_{i\in S} \widetilde{\ell}_i y_i \nabla_{\mathbf{w}}\Phi(\widetilde{\mathbf{w}}, x_i)$$

where $\mu = \frac{1}{\|\sum_{i\in S} \widetilde{\ell}_i(y_i\Phi(\widetilde{\mathbf{w}}, x_i))\|}$. $\qquad\square$

**(ii)** $\|\widetilde{\mathbf{w}}_u\| > 0 \implies \widetilde{\mathbf{w}}_u = \lambda\widetilde{\mathbf{g}}_u$ for some $\lambda > 0$

*Proof.* Consider a node $u$ having $\|\widetilde{\mathbf{w}}_u\| > 0$. The proof will be split into two parts depending on $\|\widetilde{\mathbf{g}}_u\| > 0$ or $\|\widetilde{\mathbf{g}}_u\| = 0$.

**Case 1:** $\|\widetilde{\mathbf{g}}_u\| > 0$

Let the angle between $\widetilde{\mathbf{w}}_u$ and $\widetilde{\mathbf{g}}_u$ be denoted by $\Delta$. Using Lemma 1, we can say $\Delta \leq \frac{\pi}{2}$. We will prove the statement by contradiction, so let's assume $\Delta > 0$.

Now, we know, $-\frac{\nabla_{\mathbf{w}_u}\mathcal{L}(\mathbf{w}(t))}{\|\nabla_{\mathbf{w}_u}\mathcal{L}(\mathbf{w}(t))\|}$ converges to $\widetilde{\mathbf{g}}_u$ and $\mathbf{v}_u(t)$ converges in direction of $\widetilde{\mathbf{w}}_u$. Taking dot product with $\frac{\widetilde{\mathbf{g}}_u}{\|\widetilde{\mathbf{g}}_u\|}$ on both sides of Equation (8) and using Lemma 3, we can say there exists a time $t_1$ and a small enough $\epsilon$, such that for any $t > t_1$,

$$\left(\frac{\widetilde{\mathbf{g}}_u}{\|\widetilde{\mathbf{g}}_u\|}\right)^\top \frac{d\mathbf{v}_u(t)}{dt} \geq \eta(t)e^{\alpha_u(t)}\|\nabla_{\mathbf{w}_u}\mathcal{L}(\mathbf{w}(t))\|\epsilon \tag{10}$$

Now, using the fact that $\alpha_u \to \infty$ and Equation (7), we can say

$$\int_{t_1}^{\infty} \eta(t) e^{\alpha_u(t)} \|\nabla_{\mathbf{w}_u} \mathcal{L}(\mathbf{w}(t))\| dt = \infty$$

Integrating Equation (10) on both the sides from $t_1$ to $\infty$, we get

$$\left( \frac{\widetilde{\mathbf{g}}_u}{\|\widetilde{\mathbf{g}}_u\|} \right)^{\top} \left( \frac{\widetilde{\mathbf{w}}_u}{\|\widetilde{\mathbf{w}}_u\|} - \mathbf{v}_u(t_1) \right) \geq \infty$$

This is not possible as vectors on LHS have bounded norm. This contradicts. Hence $\Delta = 0$.

**Case 2:** $\|\widetilde{\mathbf{g}}_u\| = 0$

We are going to show that it is not possible to have $\|\widetilde{\mathbf{w}}_u\| > 0$ and $\|\widetilde{\mathbf{g}}_u\| = 0$. Using Lemma 2, we can say there exists atleast one node $v$ satisfying $\|\widetilde{\mathbf{w}}_v\| > 0$ and $\|\widetilde{\mathbf{g}}_v\| > 0$. Now, using Equation (7), we can say

$$\|\mathbf{w}_u(t)\| \leq \int_{k=0}^{t} \eta(k) \|\mathbf{w}_u(k)\|^2 \|\nabla_{\mathbf{w}_u} \mathcal{L}(\mathbf{w}(k))\| dk$$

From **Case 1**, we know, for any $\epsilon > 0$, that there exists a time $t_1$, such that for $t > t_1$, $\left( \frac{\mathbf{w}_v(t)}{\|\mathbf{w}_v(t)\|} \right)^{\top} \left( \frac{-\nabla_{\mathbf{w}_v} \mathcal{L}(\mathbf{w}(t))}{\|\nabla_{\mathbf{w}_v} \mathcal{L}(\mathbf{w}(t))\|} \right) \geq \cos(\epsilon)$. Now, using Equation (7), we can say

$$\|\mathbf{w}_v(t)\| \geq \|\mathbf{w}_v(t_1)\| + \cos(\epsilon) \int_{k=t_1}^{t} \eta(k) \|\mathbf{w}_v(k)\|^2 \|\nabla_{\mathbf{w}_v} \mathcal{L}(\mathbf{w}(k))\| dk$$

Thus, we can say, for $t > t_1$,

$$\frac{\|\mathbf{w}_u(t)\|}{\|\mathbf{w}_v(t)\|} \leq \frac{\int_{k=0}^{t} \eta(k) \|\mathbf{w}_u(k)\|^2 \|\nabla_{\mathbf{w}_u} \mathcal{L}(\mathbf{w}(k))\| dk}{\|\mathbf{w}_v(t_1)\| + \cos(\epsilon) \int_{k=t_1}^{t} \eta(k) \|\mathbf{w}_v(k)\|^2 \|\nabla_{\mathbf{w}_v} \mathcal{L}(\mathbf{w}(k))\| dk}$$

Now, as $\|\widetilde{\mathbf{w}}_u\| > 0$ and $\|\widetilde{\mathbf{w}}_v\| > 0$, therefore both the integrals diverge. Also, the integrands converge in ratio to 0 as $\|\widetilde{\mathbf{g}}_u\| = 0$ and $\|\widetilde{\mathbf{g}}_v\| > 0$. Thus, taking limit $t \to \infty$ on both the sides and using the Integral form of Stolz-Cesaro theorem, we can say

$$\lim_{t \to \infty} \frac{\|\mathbf{w}_u(t)\|}{\|\mathbf{w}_v(t)\|} \leq 0$$

However, this is not possible as $\|\widetilde{\mathbf{w}}_u\| > 0$ and $\|\widetilde{\mathbf{w}}_v\| > 0$. This contradicts. Therefore, such a case is not possible. $\qquad\square$

### C.1.2 GRADIENT DESCENT

**Proposition.** *Under assumptions (A1)-(A5) for gradient descent, for EWN, the following hold:*

(i) $\lim_{t \to \infty} \frac{-\nabla_{\mathbf{w}} \mathcal{L}(\mathbf{w}(t))}{\|\nabla_{\mathbf{w}} \mathcal{L}(\mathbf{w}(t))\|} = \mu \sum_{i=1}^{m} \widetilde{\ell}_i y_i \nabla_{\mathbf{w}} \Phi(\widetilde{\mathbf{w}}, \mathbf{x}_i) = \widetilde{\mathbf{g}}$, *where* $\mu > 0$.

(ii) *Let* $\widetilde{\mathbf{w}}_u = \lim_{t \to \infty} \frac{\mathbf{w}_u(t)}{\|\mathbf{w}(t)\|}$ *and* $\widetilde{\mathbf{g}}_u = \lim_{t \to \infty} \frac{-\nabla_{\mathbf{w}_u} \mathcal{L}(\mathbf{w}(t))}{\|\nabla_{\mathbf{w}} \mathcal{L}(\mathbf{w}(t))\|}$. *Then,* $\widetilde{\mathbf{w}}_u = \lambda \widetilde{\mathbf{g}}_u$ *for some* $\lambda \geq 0$

**Update Equations:**

$$\alpha_u(t+1) = \alpha_u(t) - \eta(t) e^{\alpha_u(t)} \frac{\mathbf{v}_u(t)^{\top} \nabla_{\mathbf{w}_u} \mathcal{L}(\mathbf{w}(t))}{\|\mathbf{v}_u(t)\|} \tag{11}$$

$$\mathbf{v}_u(t+1) = \mathbf{v}_u(t) - \eta(t) \frac{e^{\alpha_u(t)}}{\|\mathbf{v}_u(t)\|} \left( I - \frac{\mathbf{v}_u(t) \mathbf{v}_u(t)^{\top}}{\|\mathbf{v}_u(t)\|^2} \right) \nabla_{\mathbf{w}_u} \mathcal{L}(\mathbf{w}(t)) \tag{12}$$

**(i)** $\lim_{t \to \infty} \frac{-\nabla_{\mathbf{w}} \mathcal{L}(\mathbf{w}(t))}{\|\nabla_{\mathbf{w}} \mathcal{L}(\mathbf{w}(t))\|} = \mu \sum_{i=1}^{m} \widetilde{\ell}_i y_i \nabla_{\mathbf{w}} \Phi(\widetilde{\mathbf{w}}, \mathbf{x}_i) = \widetilde{\mathbf{g}}$, where $\mu > 0$

*Proof.* Follows exactly as shown for gradient flow in Appendix C.1.1. $\qquad\square$

**(ii)** $\|\widetilde{\mathbf{w}}_u\| > 0 \implies \widetilde{\mathbf{w}}_u = \lambda \widetilde{\mathbf{g}}_u$ for some $\lambda > 0$

*Proof.* Consider a node $u$ having $\|\widetilde{\mathbf{w}}_u\| > 0$. The proof will be split into two parts depending on $\|\widetilde{\mathbf{g}}_u\| > 0$ or $\|\widetilde{\mathbf{g}}_u\| = 0$.

**Case 1:** $\|\widetilde{\mathbf{g}}_u\| > 0$.

Let the angle between $\widetilde{\mathbf{w}}_u$ and $\widetilde{\mathbf{g}}_u$ be denoted by $\Delta$. Using Lemma 1, we can say $\Delta \leq \frac{\pi}{2}$. We will prove the statement by contradiction, so let's assume $\Delta > 0$.

Now, we know, $-\frac{\nabla_{\mathbf{w}_u}\mathcal{L}(\mathbf{w}(t))}{\|\nabla_{\mathbf{w}_u}\mathcal{L}(\mathbf{w}(t))\|}$ converges to $\widetilde{\mathbf{g}}_u$ and $\mathbf{v}_u(t)$ converges in direction of $\widetilde{\mathbf{w}}_u$. Taking dot product with $\frac{\widetilde{\mathbf{g}}_u}{\|\widetilde{\mathbf{g}}_u\|}$ on both sides of Equation (12) and using Lemma 3, we can say there exists a time $t_1$ and a small enough $\epsilon$, such that for any $t > t_1$,

$$\frac{\mathbf{v}_u(t+1)^\top \widetilde{\mathbf{g}}_u}{\|\widetilde{\mathbf{g}}_u\|} \geq \frac{\mathbf{v}_u(t)^\top \widetilde{\mathbf{g}}_u}{\|\widetilde{\mathbf{g}}_u\|} + \epsilon(\eta(t)\frac{e^{\alpha_u(t)}}{\|\mathbf{v}_u(t)\|}\|\nabla_{\mathbf{w}_u}\mathcal{L}(\mathbf{w}(t))\|)$$

However, in this case, $\|\mathbf{v}_u(t)\|$ doesn't stay constant and thus increase in dot product doesn't directly correspond to an increase in angle. Now, using Equation (12), we can say

$$\|\mathbf{v}_u(t+1)\|^2 \leq \|\mathbf{v}_u(t)\|^2 + (\eta(t)\frac{e^{\alpha_u(t)}}{\|\mathbf{v}_u(t)\|}\|\nabla_{\mathbf{w}_u}\mathcal{L}(\mathbf{w}(t))\|)^2 \tag{13}$$

Using the above two equations, we can say, for time $t > t_1$,

$$\frac{\mathbf{v}_u(t+1)^\top \widetilde{\mathbf{g}}_u}{\|\mathbf{v}_u(t+1)\|\|\widetilde{\mathbf{g}}_u\|} \geq \frac{\frac{\mathbf{v}_u(t)^\top \widetilde{\mathbf{g}}_u}{\|\widetilde{\mathbf{g}}_u\|} + \epsilon(\eta(t)\frac{e^{\alpha_u(t)}}{\|\mathbf{v}_u(t)\|}\|\nabla_{\mathbf{w}_u}\mathcal{L}(\mathbf{w}(t))\|)}{\sqrt{\|\mathbf{v}_u(t)\|^2 + (\eta(t)\frac{e^{\alpha_u(t)}}{\|\mathbf{v}_u(t)\|}\|\nabla_{\mathbf{w}_u}\mathcal{L}(\mathbf{w}(t))\|)^2}}$$

Unrolling the equation above, we get

$$\frac{\mathbf{v}_u(t+1)^\top \widetilde{\mathbf{g}}_u}{\|\mathbf{v}_u(t+1)\|\|\widetilde{\mathbf{g}}_u\|} \geq \frac{\frac{\mathbf{v}_u(t_1)^\top \widetilde{\mathbf{g}}_u}{\|\widetilde{\mathbf{g}}_u\|} + \sum_{k=t_1}^{k=t} \epsilon(\eta(k)\frac{e^{\alpha_u(k)}}{\|\mathbf{v}_u(k)\|}\|\nabla_{\mathbf{w}_u}\mathcal{L}(\mathbf{w}(k))\|)}{\sqrt{\|\mathbf{v}_u(t_1)\|^2 + \sum_{k=t_1}^{k=t}(\eta(k)\frac{e^{\alpha_u(k)}}{\|\mathbf{v}_u(k)\|}\|\nabla_{\mathbf{w}_u}\mathcal{L}(\mathbf{w}(k))\|)^2}} \tag{14}$$

Now, as $\alpha_u(t) \to \infty$, therefore, using Equation (11), we can say

$$\sum_{k=t_1}^{k=\infty} \eta(k)e^{\alpha_u(k)}\|\nabla_{\mathbf{w}_u}\mathcal{L}(\mathbf{w}(k))\| = \infty$$

Now, using this identity, along with the Assumption (A5), Equation (13) and Lemma 5, we can say

$$\sum_{k=t_1}^{\infty} \eta(k)\frac{e^{\alpha_u(k)}}{\|\mathbf{v}_u(k)\|}\|\nabla_{\mathbf{w}_u}\mathcal{L}(\mathbf{w}(k))\| = \infty$$

Using this along with Equation (14) and Lemma 4, we can say

$$\lim_{t\to\infty} \frac{\mathbf{v}_u(t+1)^\top \widetilde{\mathbf{g}}_u}{\|\mathbf{v}_u(t+1)\|\|\widetilde{\mathbf{g}}_u\|} \geq \infty$$

However, this is not possible as the vectors on LHS have bounded norm. This contradicts. Thus $\Delta = 0$.

**Case 2:** $\|\widetilde{\mathbf{g}}_u\| = 0$

We are going to show that its not possible to have $\|\widetilde{\mathbf{w}}_u\| > 0$ and $\|\widetilde{\mathbf{g}}_u\| = 0$. By Lemma 3, we know there exists atleast one node $s$ satisfying $\|\widetilde{\mathbf{w}}_s\| > 0$ and $\|\widetilde{\mathbf{g}}_s\| > 0$. Now from Equation (11), we can say

$$\alpha_u(t) = \alpha_u(0) - \sum_{k=0}^{k=t-1} \eta(k)e^{\alpha_u(k)}\frac{\mathbf{v}_u(k)^\top \nabla_{\mathbf{w}_u}\mathcal{L}(\mathbf{w}(k))}{\|\mathbf{v}_u(k)\|}$$

$$\alpha_s(t) = \alpha_s(0) - \sum_{k=0}^{k=t-1} \eta(k)e^{\alpha_s(k)}\frac{\mathbf{v}_s(k)^\top \nabla_{\mathbf{w}_s}\mathcal{L}(\mathbf{w}(k))}{\|\mathbf{v}_s(k)\|}$$

Thus,

$$\alpha_u(t) - \alpha_s(t) = (\alpha_u(0) - \alpha_s(0)) + \sum_{k=0}^{k=t-1} (\eta(k) e^{\alpha_s(k)} \frac{\mathbf{v}_s(k)^\top \nabla_{\mathbf{w}_s(k)} \mathcal{L}(\mathbf{w}(k))}{\|\mathbf{v}_s(k)\|} - \eta(k) e^{\alpha_u(k)} \frac{\mathbf{v}_u(k)^\top \nabla_{\mathbf{w}_u} \mathcal{L}(\mathbf{w}(k))}{\|\mathbf{v}_u(k)\|})$$

(15)

Now, we know, $\alpha_u(t)$ and $\alpha_s(t) \to \infty$. Also, as $\|\widetilde{\mathbf{w}}_u\| > 0$ and $\|\widetilde{\mathbf{w}}_s\| > 0$, therefore $\alpha_u - \alpha_s$ converges. Therefore the RHS of Equation (15) converges as well. However, RHS is the difference of two diverging series. Also, as $\mathbf{v}_s(t)$ and $\nabla_{\mathbf{w}_s} \mathcal{L}(\mathbf{w}(t))$ eventually get aligned and $\lim_{t\to\infty} \frac{\|\nabla_{\mathbf{w}_u} \mathcal{L}(\mathbf{w}(t))\|}{\|\nabla_{\mathbf{w}_s} \mathcal{L}(\mathbf{w}(t))\|} = 0$, so, we can say

$$\lim_{t\to\infty} \frac{e^{\alpha_u(t)} \frac{\mathbf{v}_u(t)^\top \nabla_{\mathbf{w}_u} \mathcal{L}(\mathbf{w}(t))}{\|\mathbf{v}_u(t)\|}}{e^{\alpha_s(t)} \frac{\mathbf{v}_s(t)^\top \nabla_{\mathbf{w}_s} \mathcal{L}(\mathbf{w}(t))}{\|\mathbf{v}_s(t)\|}} = 0$$

However, this contradicts Lemma 6, as the ratio must be converging to 1 if the limit exists. Therefore this case is not possible. □

## C.2 STANDARD WEIGHT NORMALIZATION

### C.2.1 GRADIENT FLOW

**Proposition.** *Under assumptions (A1)-(A4) for gradient flow, for SWN, the following hold:*

*(i)* $\lim_{t\to\infty} \frac{-\nabla_{\mathbf{w}} \mathcal{L}(\mathbf{w}(t))}{\|\nabla_{\mathbf{w}} \mathcal{L}(\mathbf{w}(t))\|} = \mu \sum_{i=1}^m \widetilde{\ell}_i y_i \nabla_{\mathbf{w}} \Phi(\widetilde{\mathbf{w}}, \mathbf{x}_i) = \widetilde{\mathbf{g}}$, *where* $\mu > 0$.

*(ii) Let* $\widetilde{\mathbf{w}}_u = \lim_{t\to\infty} \frac{\mathbf{w}_u(t)}{\|\mathbf{w}(t)\|}$ *and* $\widetilde{\mathbf{g}}_u = \lim_{t\to\infty} \frac{-\nabla_{\mathbf{w}_u} \mathcal{L}(\mathbf{w}(t))}{\|\nabla_{\mathbf{w}} \mathcal{L}(\mathbf{w}(t))\|}$. *Then,* $\widetilde{\mathbf{w}}_u = \lambda \widetilde{\mathbf{g}}_u$ *for some* $\lambda \geq 0$

**Update Equations:**

$$\frac{d\gamma_u(t)}{dt} = -\eta(t) \frac{\mathbf{v}_u(t)^\top \nabla_{\mathbf{w}_u} \mathcal{L}(\mathbf{w}(t))}{\|\mathbf{v}_u(t)\|}$$

(16)

$$\frac{d\mathbf{v}_u(t)}{dt} = -\eta(t) \frac{\gamma_u(t)}{\|\mathbf{v}_u(t)\|} (I - \frac{\mathbf{v}_u(t)\mathbf{v}_u(t)^\top}{\|\mathbf{v}_u(t)\|^2}) \nabla_{\mathbf{w}_u} \mathcal{L}(\mathbf{w}(t))$$

(17)

*(i)* $\lim_{t\to\infty} \frac{-\nabla_{\mathbf{w}} \mathcal{L}(\mathbf{w}(t))}{\|\nabla_{\mathbf{w}} \mathcal{L}(\mathbf{w}(t))\|} = \mu \sum_{i=1}^m \widetilde{\ell}_i y_i \nabla_{\mathbf{w}} \Phi(\widetilde{\mathbf{w}}, \mathbf{x}_i) = \widetilde{\mathbf{g}}$, *where* $\mu > 0$

*Proof.* Follows exactly as shown for gradient flow in Appendix C.1.1. □

*(ii)* $\|\widetilde{\mathbf{w}}_u\| > 0 \implies \widetilde{\mathbf{w}}_u = \lambda \widetilde{\mathbf{g}}_u$ for some $\lambda > 0$

*Proof.* Consider a node $u$ having $\|\widetilde{\mathbf{w}}_u\| > 0$. In this case, $\gamma_u(t)$ can either tend to $\infty$ or $-\infty$. We will consider the case $\gamma_u(t) \to \infty$. The other case can be handled similarly. The proof will be split into two parts depending on $\|\widetilde{\mathbf{g}}_u\| > 0$ or $\|\widetilde{\mathbf{g}}_u\| = 0$.

**Case 1:** $\|\widetilde{\mathbf{g}}_u\| > 0$

Let the angle between $\widetilde{\mathbf{w}}_u$ and $\widetilde{\mathbf{g}}_u$ be denoted by $\Delta$. Using Lemma 1, we can say $\Delta \leq \frac{\pi}{2}$. We will prove the statement by contradiction, so let's assume $\Delta > 0$.

Now, we know, $-\frac{\nabla_{\mathbf{w}_u} \mathcal{L}(\mathbf{w}(t))}{\|\nabla_{\mathbf{w}_u} \mathcal{L}(\mathbf{w}(t))\|}$ converges to $\widetilde{\mathbf{g}}_u$ and $\mathbf{v}_u(t)$ converges in direction of $\widetilde{\mathbf{w}}_u$. Taking dot product with $\frac{\widetilde{\mathbf{g}}_u}{\|\widetilde{\mathbf{g}}_u\|}$ on both sides of Equation (17) and using Lemma 3, we can say there exists a time $t_1$ and a small enough $\epsilon$, such that for any $t > t_1$,

$$\left(\frac{\widetilde{\mathbf{g}}_u}{\|\widetilde{\mathbf{g}}_u\|}\right)^\top \frac{d\mathbf{v}_u(t)}{dt} \geq \eta(t) \gamma_u(t) \|\nabla_{\mathbf{w}_u} \mathcal{L}(\mathbf{w}(t))\| \epsilon$$

(18)

Now, using the fact that $\gamma_u \to \infty$ and Equation (16), we can say

$$\int_{t_1}^{\infty} \eta(t)\|\nabla_{\mathbf{w}_u}\mathcal{L}(\mathbf{w}(t))\|dt = \infty$$

Integrating Equation (18) on both the sides from $t_1$ to $\infty$, we get

$$\left(\frac{\widetilde{\mathbf{g}}_u}{\|\widetilde{\mathbf{g}}_u\|}\right)^{\top}\left(\frac{\widetilde{\mathbf{w}}_u}{\|\widetilde{\mathbf{w}}_u\|} - \mathbf{v}_u(t_1)\right) \geq \infty$$

This is not possible as vectors on LHS have bounded norm. This contradicts. Hence $\Delta = 0$.

**Case 2:** $\|\widetilde{\mathbf{g}}_u\| = 0$

We are going to show that it is not possible to have $\|\widetilde{\mathbf{w}}_u\| > 0$ and $\|\widetilde{\mathbf{g}}_u\| = 0$. Using Lemma 2, we can say there exists atleast one node $s$ satisfying $\|\widetilde{\mathbf{w}}_s\| > 0$ and $\|\widetilde{\mathbf{g}}_s\| > 0$. Now, using Equation (16), we can say

$$\|\mathbf{w}_u(t)\| \leq \int_{k=0}^{t} \eta(k)\|\nabla_{\mathbf{w}_u}\mathcal{L}(\mathbf{w}(k))\|dk$$

From **Case 1**, we know, for any $\epsilon > 0$, that there exists a time $t_1$, such that for $t > t_1$, $\left(\frac{\mathbf{w}_s(t)}{\|\mathbf{w}_s(t)\|}\right)^{\top}\left(\frac{-\nabla_{\mathbf{w}_s}\mathcal{L}(\mathbf{w}(t))}{\|\nabla_{\mathbf{w}_s}\mathcal{L}(\mathbf{w}(t))\|}\right) \geq \cos(\epsilon)$. Now, using Equation (16), we can say

$$\|\mathbf{w}_s(t)\| \geq \|\mathbf{w}_s(t_1)\| + \cos(\epsilon)\int_{k=t_1}^{t} \eta(k)\|\nabla_{\mathbf{w}_s}\mathcal{L}(\mathbf{w}(k))\|dk$$

Thus, we can say, for $t > t_1$,

$$\frac{\|\mathbf{w}_u(t)\|}{\|\mathbf{w}_s(t)\|} \leq \frac{\int_{k=0}^{t} \eta(k)\|\nabla_{\mathbf{w}_u}\mathcal{L}(\mathbf{w}(k))\|dk}{\|\mathbf{w}_v(t_1)\| + \cos(\epsilon)\int_{k=t_1}^{t} \eta(k)\|\nabla_{\mathbf{w}_s}\mathcal{L}(\mathbf{w}(k))\|dk}$$

Now, as $\|\widetilde{\mathbf{w}}_u\| > 0$ and $\|\widetilde{\mathbf{w}}_s\| > 0$, therefore both the integrals diverge. Also, the integrands converge in ratio to 0 as $\|\widetilde{\mathbf{g}}_u\| = 0$ and $\|\widetilde{\mathbf{g}}_v\| > 0$. Thus, taking limit $t \to \infty$ on both the sides and using the Integral form of Stolz-Cesaro theorem, we can say

$$\lim_{t \to \infty} \frac{\|\mathbf{w}_u(t)\|}{\|\mathbf{w}_s(t)\|} \leq 0$$

However, this is not possible as $\|\widetilde{\mathbf{w}}_u\| > 0$ and $\|\widetilde{\mathbf{w}}_s\| > 0$. This contradicts. Therefore, such a case is not possible. $\qquad\square$

### C.2.2 GRADIENT DESCENT

**Proposition.** *Under assumptions (A1)-(A5) for gradient descent, for SWN, the following hold:*

*(i)* $\lim_{t \to \infty} \frac{-\nabla_{\mathbf{w}}\mathcal{L}(\mathbf{w}(t))}{\|\nabla_{\mathbf{w}}\mathcal{L}(\mathbf{w}(t))\|} = \mu \sum_{i=1}^{m} \widetilde{\ell}_i y_i \nabla_{\mathbf{w}}\Phi(\widetilde{\mathbf{w}}, \mathbf{x}_i) = \widetilde{\mathbf{g}}$, *where $\mu > 0$.*

*(ii) Let $\widetilde{\mathbf{w}}_u = \lim_{t \to \infty} \frac{\mathbf{w}_u(t)}{\|\mathbf{w}(t)\|}$ and $\widetilde{\mathbf{g}}_u = \lim_{t \to \infty} \frac{-\nabla_{\mathbf{w}_u}\mathcal{L}(\mathbf{w}(t))}{\|\nabla_{\mathbf{w}}\mathcal{L}(\mathbf{w}(t))\|}$. Then, $\widetilde{\mathbf{w}}_u = \lambda\widetilde{\mathbf{g}}_u$ for some $\lambda \geq 0$*

**Update Equations:**

$$\gamma_u(t+1) = \gamma_u(t) - \eta(t)\frac{\mathbf{v}_u(t)^{\top}\nabla_{\mathbf{w}_u}\mathcal{L}(\mathbf{w}(t))}{\|\mathbf{v}_u(t)\|} \tag{19}$$

$$\mathbf{v}_u(t+1) = \mathbf{v}_u(t) - \eta(t)\frac{\gamma_u(t)}{\|\mathbf{v}_u(t)\|}\left(I - \frac{\mathbf{v}_u(t)\mathbf{v}_u(t)^{\top}}{\|\mathbf{v}_u(t)\|^2}\right)\nabla_{\mathbf{w}_u}\mathcal{L}(\mathbf{w}(t)) \tag{20}$$

*(i)* $\lim_{t \to \infty} \frac{-\nabla_{\mathbf{w}}\mathcal{L}(\mathbf{w}(t))}{\|\nabla_{\mathbf{w}}\mathcal{L}(\mathbf{w}(t))\|} = \mu \sum_{i=1}^{m} \widetilde{\ell}_i y_i \nabla_{\mathbf{w}}\Phi(\widetilde{\mathbf{w}}, \mathbf{x}_i) = \widetilde{\mathbf{g}}$, where $\mu > 0$

*Proof.* Follows exactly as shown for gradient flow in Appendix C.1.1. $\qquad\square$

**(ii)** $\|\widetilde{\mathbf{w}}_u\| > 0 \implies \widetilde{\mathbf{w}}_u = \lambda \widetilde{\mathbf{g}}_u$ for some $\lambda > 0$

*Proof.* Consider a node $u$ having $\|\widetilde{\mathbf{w}}_u\| > 0$. In this case, $\gamma_u(t)$ can either tend to $\infty$ or $-\infty$. We will consider the case $\gamma_u(t) \to \infty$. The other case can be handled similarly. The proof will be split into two parts depending on $\|\widetilde{\mathbf{g}}_u\| > 0$ or $\|\widetilde{\mathbf{g}}_u\| = 0$.

**Case 1:** $\|\widetilde{\mathbf{g}}_u\| > 0$

Let the angle between $\widetilde{\mathbf{w}}_u$ and $\widetilde{\mathbf{g}}_u$ be denoted by $\Delta$. Using Lemma 1, we can say $\Delta \leq \frac{\pi}{2}$. We will prove the statement by contradiction, so let's assume $\Delta > 0$.

Now, we know, $-\frac{\nabla_{\mathbf{w}_u}\mathcal{L}(\mathbf{w}(t))}{\|\nabla_{\mathbf{w}_u}\mathcal{L}(\mathbf{w}(t))\|}$ converges to $\widetilde{\mathbf{g}}_u$ and $\mathbf{v}_u(t)$ converges in direction of $\widetilde{\mathbf{w}}_u$. Taking dot product with $\frac{\widetilde{\mathbf{g}}_u}{\|\widetilde{\mathbf{g}}_u\|}$ on both sides of Equation (20) and using Lemma 3, we can say there exists a time $t_1$ and a small enough $\epsilon$, such that for any $t > t_1$,

$$\frac{\mathbf{v}_u(t+1)^\top \widetilde{\mathbf{g}}_u}{\|\widetilde{\mathbf{g}}_u\|} \geq \frac{\mathbf{v}_u(t)^\top \widetilde{\mathbf{g}}_u}{\|\widetilde{\mathbf{g}}_u\|} + \epsilon(\eta(t)\frac{\gamma_u(t)}{\|\mathbf{v}_u(t)\|}\|\nabla_{\mathbf{w}_u}\mathcal{L}(\mathbf{w}(t))\|)$$

However, in this case, $\|\mathbf{v}_u(t)\|$ doesn't stay constant and thus increase in dot product doesn't directly correspond to an increase in angle. Now, using Equation (20), we can say

$$\|\mathbf{v}_u(t+1)\|^2 \leq \|\mathbf{v}_u(t)\|^2 + (\eta(t)\frac{\gamma_u(t)}{\|\mathbf{v}_u(t)\|}\|\nabla_{\mathbf{w}_u}\mathcal{L}(\mathbf{w}(t))\|)^2 \tag{21}$$

Using the above two equations, we can say, for time $t > t_1$,

$$\frac{\mathbf{v}_u(t+1)^\top \widetilde{\mathbf{g}}_u}{\|\mathbf{v}_u(t+1)\|\|\widetilde{\mathbf{g}}_u\|} \geq \frac{\frac{\mathbf{v}_u(t)^\top \widetilde{\mathbf{g}}_u}{\|\widetilde{\mathbf{g}}_u\|} + \epsilon(\eta(t)\frac{\gamma_u(t)}{\|\mathbf{v}_u(t)\|}\|\nabla_{\mathbf{w}_u}\mathcal{L}(\mathbf{w}(t))\|)}{\sqrt{\|\mathbf{v}_u(t)\|^2 + (\eta(t)\frac{\gamma_u(t)}{\|\mathbf{v}_u(t)\|}\|\nabla_{\mathbf{w}_u}\mathcal{L}(\mathbf{w}(t))\|)^2}}$$

Unrolling the equation above, we get

$$\frac{\mathbf{v}_u(t+1)^\top \widetilde{\mathbf{g}}_u}{\|\mathbf{v}_u(t+1)\|\|\widetilde{\mathbf{g}}_u\|} \geq \frac{\frac{\mathbf{v}_u(t_1)^\top \widetilde{\mathbf{g}}_u}{\|\widetilde{\mathbf{g}}_u\|} + \sum_{k=t_1}^{k=t} \epsilon(\eta(k)\frac{\gamma_u(k)}{\|\mathbf{v}_u(k)\|}\|\nabla_{\mathbf{w}_u}\mathcal{L}(\mathbf{w}(k))\|)}{\sqrt{\|\mathbf{v}_u(t_1)\|^2 + \sum_{k=t_1}^{k=t}(\eta(k)\frac{\gamma_u(k)}{\|\mathbf{v}_u(k)\|}\|\nabla_{\mathbf{w}_u}\mathcal{L}(\mathbf{w}(k))\|)^2}} \tag{22}$$

Now, as $\gamma_u(t) \to \infty$, therefore, using Equation (20), we can say

$$\sum_{k=t_1}^{k=\infty} \eta(k)\|\nabla_{\mathbf{w}_u}\mathcal{L}(\mathbf{w}(k))\| = \infty$$

Now, using this identity, along with the assumption (A5), Equation (21) and Lemma 5, we can say

$$\sum_{k=t_1}^{\infty} \eta(k)\frac{\gamma_u(k)}{\|\mathbf{v}_u(k)\|}\|\nabla_{\mathbf{w}_u}\mathcal{L}(\mathbf{w}(k))\| = \infty$$

Using this along with Equation (22) and Lemma 4, we can say

$$\lim_{t\to\infty} \frac{\mathbf{v}_u(t+1)^\top \widetilde{\mathbf{g}}_u}{\|\mathbf{v}_u(t+1)\|\|\widetilde{\mathbf{g}}_u\|} \geq \infty$$

However, this is not possible as the vectors on LHS have bounded norm. This contradicts. Thus $\Delta = 0$.

**Case 2:** $\|\widetilde{\mathbf{g}}_u\| = 0$

We are going to show that its not possible to have $\|\widetilde{\mathbf{w}}_u\| > 0$ and $\|\widetilde{\mathbf{g}}_u\| = 0$. By Lemma 3, we know there exists atleast one node $s$ satisfying $\|\widetilde{\mathbf{w}}_s\| > 0$ and $\|\widetilde{\mathbf{g}}_s\| > 0$. Now from Equation (19), we can say

$$\gamma_u(t) = \gamma_u(0) - \sum_{k=0}^{k=t-1} \eta(k)\frac{\mathbf{v}_u(k)^\top \nabla_{\mathbf{w}_u}\mathcal{L}(\mathbf{w}(k))}{\|\mathbf{v}_u(k)\|}$$

$$\gamma_s(t) = \gamma_s(0) - \sum_{k=0}^{k=t-1} \eta(k) \frac{\mathbf{v}_s(k)^\top \nabla_{\mathbf{w}_s} \mathcal{L}(\mathbf{w}(k))}{\|\mathbf{v}_s(k)\|}$$

Now, $\gamma_s(t)$ either diverges to $\infty$ or $-\infty$. In both the cases, it is a strictly monotonic sequence for large enough $t$. Also

$$\lim_{t\to\infty} \frac{\gamma_u(t+1) - \gamma_u(t)}{\gamma_s(t+1) - \gamma_s(t)} = 0$$

as $\|\widetilde{\mathbf{g}}_u\| = 0, \|\widetilde{\mathbf{g}}_s\| > 0$ and from **Case 1**, $\mathbf{w}_s(t)$ and $-\nabla_{\mathbf{w}_s}\mathcal{L}(\mathbf{w}(t))$ eventually get aligned. Thus, using Stolz-Cesaro theorem, we can say

$$\lim_{t\to\infty} \frac{\gamma_u(t)}{\gamma_s(t)} = 0$$

However, this is not possible as $\|\widetilde{\mathbf{w}}_u\| > 0$ and $\|\widetilde{\mathbf{w}}_s\| > 0$. This contradicts. Therefore, this is not possible. $\qquad\square$

## D  PROOF OF THEOREM 2

**Theorem.** *Under assumptions (A1)-(A4) for gradient flow and (A1)-(A5) for gradient descent, the following hold*

*(i) for EWN,* $\|\widetilde{\mathbf{w}}_u\| > 0, \|\widetilde{\mathbf{w}}_v\| > 0 \implies \lim_{t\to\infty} \frac{\|\mathbf{w}_u(t)\|\|\nabla_{\mathbf{w}_u}\mathcal{L}(\mathbf{w}(t))\|}{\|\mathbf{w}_v(t)\|\|\nabla_{\mathbf{w}_v}\mathcal{L}(\mathbf{w}(t))\|} = 1$

*(ii) for SWN,* $\|\widetilde{\mathbf{w}}_u\| > 0, \|\widetilde{\mathbf{w}}_v\| > 0 \implies \lim_{t\to\infty} \frac{\|\mathbf{w}_u(t)\|\|\nabla_{\mathbf{w}_v}\mathcal{L}(\mathbf{w}(t))\|}{\|\mathbf{w}_v(t)\|\|\nabla_{\mathbf{w}_u}\mathcal{L}(\mathbf{w}(t))\|} = 1$

Proof for different cases will be split into different subsections and the corresponding case will be restated there for ease of the reader.

### D.1  EXPONENTIAL WEIGHT NORMALIZATION

#### D.1.1  GRADIENT FLOW

**Theorem.** *Under assumptions (A1)-(A4) for gradient flow, the following holds for EWN:*

*(i)* $\|\widetilde{\mathbf{w}}_u\| > 0, \|\widetilde{\mathbf{w}}_v\| > 0 \implies \lim_{t\to\infty} \frac{\|\mathbf{w}_u(t)\|\|\nabla_{\mathbf{w}_u}\mathcal{L}(\mathbf{w}(t))\|}{\|\mathbf{w}_v(t)\|\|\nabla_{\mathbf{w}_v}\mathcal{L}(\mathbf{w}(t))\|} = 1$

*Proof.* Consider $u$ and $v$ such that $\|\widetilde{\mathbf{w}}_u\| > 0$ and $\|\widetilde{\mathbf{w}}_v\| > 0$. Using Proposition 2, we can say $\|\widetilde{\mathbf{g}}_u\| > 0$ and $\|\widetilde{\mathbf{g}}_v\| > 0$. Also, from Proposition 2, we can say, for both $u$ and $v$, weights and gradients converge in opposite directions. Hence there exists a time $t_2$, such that for any $t > t_2$,

- $-\nabla_{\mathbf{w}_u}\mathcal{L}(\mathbf{w}(t))$ and $\mathbf{w}_u(t)$ make an angle $\epsilon$ or lesser with each other

- $-\nabla_{\mathbf{w}_v}\mathcal{L}(\mathbf{w}(t))$ and $\mathbf{w}_v(t)$ make an angle $\epsilon$ or lesser with each other

Then, using Equation (7), we can say for any time $t > t_2$,

$$\|\mathbf{w}_u(t)\| \geq \|\mathbf{w}_u(t_2)\| + \cos(\epsilon) \int_{k=t_2}^t \eta(k)\|w_u(k)\|^2 \|\nabla_{\mathbf{w}_u}\mathcal{L}(\mathbf{w}(k))\| dk$$

$$\|\mathbf{w}_u(t)\| \leq \|\mathbf{w}_u(t_2)\| + \int_{k=t_2}^t \eta(k)\|w_u(k)\|^2 \|\nabla_{\mathbf{w}_u}\mathcal{L}(\mathbf{w}(k))\| dk$$

$$\|\mathbf{w}_v(t)\| \geq \|\mathbf{w}_v(t_2)\| + \cos(\epsilon) \int_{k=t_2}^t \eta(k)\|w_v(k)\|^2 \|\nabla_{\mathbf{w}_v}\mathcal{L}(\mathbf{w}(k))\| dk$$

$$\|\mathbf{w}_v(t)\| \leq \|\mathbf{w}_v(t_2)\| + \int_{k=t_2}^t \eta(k)\|w_v(k)\|^2 \|\nabla_{\mathbf{w}_v}\mathcal{L}(\mathbf{w}(k))\| dk$$

Using the above equations, we can say, for time $t > t_2$,

$$\frac{\|\mathbf{w}_u(t)\|}{\|\mathbf{w}_v(t)\|} \geq \frac{\|\mathbf{w}_u(t_2)\| + \cos(\epsilon) \int_{k=t_2}^{t} \eta(k)\|w_u(k)\|^2 \|\nabla_{\mathbf{w}_u} \mathcal{L}(\mathbf{w}(k))\| dk}{\|\mathbf{w}_v(t_2)\| + \int_{k=t_2}^{t} \eta(k)\|w_v(k)\|^2 \|\nabla_{\mathbf{w}_v} \mathcal{L}(\mathbf{w}(k))\| dk}$$

$$\frac{\|\mathbf{w}_u(t)\|}{\|\mathbf{w}_v(t)\|} \leq \frac{\|\mathbf{w}_u(t_2)\| + \int_{k=t_2}^{t} \eta(k)\|w_u(k)\|^2 \|\nabla_{\mathbf{w}_u} \mathcal{L}(\mathbf{w}(k))\| dk}{\|\mathbf{w}_v(t_2)\| + \cos(\epsilon) \int_{k=t_2}^{t} \eta(k)\|w_v(k)\|^2 \|\nabla_{\mathbf{w}_v} \mathcal{L}(\mathbf{w}(k))\| dk}$$

We know that both integrals diverge as $\|\mathbf{w}_u(t)\|$ and $\|\mathbf{w}_v(t)\| \to \infty$, $\lim_{t \to \infty} \frac{\|w_u(t)\|}{\|w_v(t)\|}$ and $\lim_{t \to \infty} \frac{\|\nabla_{w_u(t)} \mathcal{L}\|}{\|\nabla_{w_v(t)} \mathcal{L}\|}$ exist. Taking limit $t \to \infty$ on both the equations and using the Integral form of Stolz-Cesaro theorem, we get

$$\lim_{t \to \infty} \frac{\cos(\epsilon)\|\mathbf{w}_u(t)\|^2 \|\nabla_{\mathbf{w}_u(t)} \mathcal{L}(\mathbf{w}(t))\|}{\|\mathbf{w}_v(t)\|^2 \|\nabla_{\mathbf{w}_v(t)} \mathcal{L}(\mathbf{w}(t))\|} \leq \lim_{t \to \infty} \frac{\|\mathbf{w}_u(t)\|}{\|\mathbf{w}_v(t)\|} \leq \lim_{t \to \infty} \frac{\|\mathbf{w}_u(t)\|^2 \|\nabla_{\mathbf{w}_u(t)} \mathcal{L}(\mathbf{w}(t))\|}{\cos(\epsilon)\|\mathbf{w}_v(t)\|^2 \|\nabla_{\mathbf{w}_v(t)} \mathcal{L}(\mathbf{w}(t))\|}$$

As we know these limits exist and this holds for any $\epsilon > 0$, therefore

$$\lim_{t \to \infty} \frac{\|\mathbf{w}_u(t)\|}{\|\mathbf{w}_v(t)\|} = \lim_{t \to \infty} \frac{\|\mathbf{w}_u(t)\|^2 \|\nabla_{\mathbf{w}_u(t)} \mathcal{L}(\mathbf{w}(t))\|}{\|\mathbf{w}_v(t)\|^2 \|\nabla_{\mathbf{w}_v(t)} \mathcal{L}(\mathbf{w}(t))\|}$$

Simplifying it further, we get

$$\lim_{t \to \infty} \frac{\|\mathbf{w}_u(t)\| \|\nabla_{\mathbf{w}_u(t)} \mathcal{L}\|}{\|\mathbf{w}_v(t)\| \|\nabla_{\mathbf{w}_v(t)} \mathcal{L}\|} = 1$$

□

### D.1.2 GRADIENT DESCENT

**Theorem.** *Under assumptions (A1)-(A5) for gradient descent, the following holds for EWN:*

*(i)* $\|\widetilde{\mathbf{w}}_u\| > 0, \|\widetilde{\mathbf{w}}_v\| > 0 \implies \lim_{t \to \infty} \frac{\|\mathbf{w}_u(t)\| \|\nabla_{\mathbf{w}_u} \mathcal{L}(\mathbf{w}(t))\|}{\|\mathbf{w}_v(t)\| \|\nabla_{\mathbf{w}_v} \mathcal{L}(\mathbf{w}(t))\|} = 1$

*Proof.* Consider two nodes $u$ and $s$ such that $\|\widetilde{\mathbf{w}}_u\| > 0$ and $\|\widetilde{\mathbf{w}}_s\| > 0$. From Proposition2, we know $\|\widetilde{\mathbf{g}}_u\| > 0, \|\widetilde{\mathbf{g}}_s\| > 0$ and for both $u$ and $s$, weights and gradients eventually get aligned opposite to each other. Using Equation (11), we can say

$$\alpha_u(t) = \alpha_u(0) - \sum_{k=1}^{t-1} \eta(k) e^{\alpha_u(k)} \frac{\mathbf{v}_u(k)^\top \nabla_{\mathbf{w}_u} \mathcal{L}(\mathbf{w}(k))}{\|\mathbf{v}_u(k)\|}$$

$$\alpha_s(t) = \alpha_s(0) - \sum_{k=1}^{t-1} \eta(k) e^{\alpha_s(k)} \frac{\mathbf{v}_s(k)^\top \nabla_{\mathbf{w}_s} \mathcal{L}(\mathbf{w}(k))}{\|\mathbf{v}_s(k)\|}$$

Thus,

$$\alpha_u(t) - \alpha_s(t) = (\alpha_u(0) - \alpha_s(0)) + \sum_{k=0}^{k=t-1} \left( \eta(k) e^{\alpha_s(k)} \frac{\mathbf{v}_s(k)^\top \nabla_{\mathbf{w}_s(k)} \mathcal{L}(\mathbf{w}(k))}{\|\mathbf{v}_s(k)\|} - \eta(k) e^{\alpha_u(k)} \frac{\mathbf{v}_u(k)^\top \nabla_{\mathbf{w}_u} \mathcal{L}(\mathbf{w}(k))}{\|\mathbf{v}_u(k)\|} \right)$$

$$(23)$$

Now, we know, $\alpha_u(t)$ and $\alpha_s(t) \to \infty$. Also, as $\|\widetilde{\mathbf{w}}_u\| > 0$ and $\|\widetilde{\mathbf{w}}_s\| > 0$, therefore $\alpha_u(t) - \alpha_s(t)$ converges. Therefore the RHS of Equation (23) converges as well. However, RHS is the difference of two diverging series. Also, $\lim_{t \to \infty} \frac{e^{\alpha_u(t)} \frac{\mathbf{v}_u(t)^\top \nabla_{\mathbf{w}_u} \mathcal{L}(\mathbf{w}(t))}{\|\mathbf{v}_u(t)\|}}{e^{\alpha_s(t)} \frac{\mathbf{v}_s(t)^\top \nabla_{\mathbf{w}_s} \mathcal{L}(\mathbf{w}(t))}{\|\mathbf{v}_s(t)\|}}$ exists. Therefore, using Lemma 6, we can say

$$\lim_{t \to \infty} \frac{e^{\alpha_u(t)} \|\nabla_{\mathbf{w}_u} \mathcal{L}(\mathbf{w}(t))\|}{e^{\alpha_s(t)} \|\nabla_{\mathbf{w}_s} \mathcal{L}(\mathbf{w}(t))\|} = 1$$

□

## D.2 Standard Weight Normalization

### D.2.1 Gradient Flow

**Theorem.** *Under assumptions (A1)-(A4) for gradient flow, the following holds for SWN:*

*(i)* $\|\widetilde{\mathbf{w}}_u\| > 0, \|\widetilde{\mathbf{w}}_v\| > 0 \implies \lim_{t\to\infty} \frac{\|\mathbf{w}_u(t)\|\|\nabla_{\mathbf{w}_v}\mathcal{L}(\mathbf{w}(t))\|}{\|\mathbf{w}_v(t)\|\|\nabla_{\mathbf{w}_u}\mathcal{L}(\mathbf{w}(t))\|} = 1$

*Proof.* Consider $u$ and $v$ such that $\|\widetilde{\mathbf{w}}_u\| > 0$ and $\|\widetilde{\mathbf{w}}_v\| > 0$. Using Proposition 2, we can say $\|\widetilde{\mathbf{g}}_u\| > 0$ and $\|\widetilde{\mathbf{g}}_v\| > 0$. Also, from Proposition 2, we can say, for both $u$ and $v$, weights and gradients converge in opposite directions.

Consider a time $t_2$, such that for any $t > t_2$,

- $-\nabla_{\mathbf{w}_u}\mathcal{L}(\mathbf{w}(t))$ and $\mathbf{w}_u(t)$ atmost make an angle $\epsilon$ with each other
- $-\nabla_{\mathbf{w}_v}\mathcal{L}(\mathbf{w}(t))$ and $\mathbf{w}_v(t)$ atmost make an angle $\epsilon$ with each other

Then, using Equation (16), we can say for any time $t > t_2$,

$$\|\mathbf{w}_u(t)\| \geq \|\mathbf{w}_u(t_2)\| + \cos(\epsilon)\int_{k=t_2}^{t} \eta(k)\|\nabla_{\mathbf{w}_u}\mathcal{L}(\mathbf{w}(k))\|dk$$

$$\|\mathbf{w}_u(t)\| \leq \|\mathbf{w}_u(t_2)\| + \int_{k=t_2}^{t} \eta(k)\|\nabla_{\mathbf{w}_u}\mathcal{L}(\mathbf{w}(k))\|dk$$

$$\|\mathbf{w}_v(t)\| \geq \|\mathbf{w}_v(t_2)\| + \cos(\epsilon)\int_{k=t_2}^{t} \eta(k)\|\nabla_{\mathbf{w}_v}\mathcal{L}(\mathbf{w}(k))\|dk$$

$$\|\mathbf{w}_v(t)\| \leq \|\mathbf{w}_v(t_2)\| + \int_{k=t_2}^{t} \eta(k)\|\nabla_{\mathbf{w}_v}\mathcal{L}(\mathbf{w}(k))\|dk$$

Using the above equations, we can say, for time $t > t_2$,

$$\frac{\|\mathbf{w}_u(t)\|}{\|\mathbf{w}_v(t)\|} \geq \frac{\|\mathbf{w}_u(t_2)\| + \cos(\epsilon)\int_{k=t_2}^{t} \eta(k)\|\nabla_{\mathbf{w}_u}\mathcal{L}(\mathbf{w}(k))\|dk}{\|\mathbf{w}_v(t_2)\| + \int_{k=t_2}^{t} \eta(k)\|\nabla_{\mathbf{w}_v}\mathcal{L}(\mathbf{w}(k))\|dk}$$

$$\frac{\|\mathbf{w}_u(t)\|}{\|\mathbf{w}_v(t)\|} \leq \frac{\|\mathbf{w}_u(t_2)\| + \int_{k=t_2}^{t} \eta(k)\|\nabla_{\mathbf{w}_u}\mathcal{L}(\mathbf{w}(k))\|dk}{\|\mathbf{w}_v(t_2)\| + \cos(\epsilon)\int_{k=t_2}^{t} \eta(k)\|\nabla_{\mathbf{w}_v}\mathcal{L}(\mathbf{w}(k))\|dk}$$

We know that both integrals diverge as $\|\mathbf{w}_u(t)\|$ and $\|\mathbf{w}_v(t)\| \to \infty$, $\lim_{t\to\infty}\frac{\|w_u(t)\|}{\|w_v(t)\|}$ and $\lim_{t\to\infty}\frac{\|\nabla_{w_u(t)}\mathcal{L}(\mathbf{w}(t))\|}{\|\nabla_{w_v(t)}\mathcal{L}(\mathbf{w}(t))\|}$ exist. Taking limit $t \to \infty$ on both the equations and using the Integral form of Stolz-Cesaro theorem, we get

$$\lim_{t\to\infty}\frac{\cos(\epsilon)\|\nabla_{\mathbf{w}_u(t)}\mathcal{L}(\mathbf{w}(t))\|}{\|\nabla_{\mathbf{w}_v(t)}\mathcal{L}(\mathbf{w}(t))\|} \leq \lim_{t\to\infty}\frac{\|\mathbf{w}_u(t)\|}{\|\mathbf{w}_v(t)\|} \leq \lim_{t\to\infty}\frac{\|\nabla_{\mathbf{w}_u(t)}\mathcal{L}(\mathbf{w}(t))\|}{\cos(\epsilon)\|\nabla_{\mathbf{w}_v(t)}\mathcal{L}(\mathbf{w}(t))\|}$$

As we know these limits exist and this holds for any $\epsilon > 0$, therefore

$$\lim_{t\to\infty}\frac{\|\mathbf{w}_u(t)\|\|\nabla_{\mathbf{w}_v(t)}\mathcal{L}\|}{\|\mathbf{w}_v(t)\|\|\nabla_{\mathbf{w}_u(t)}\mathcal{L}\|} = 1$$

$\square$

### D.2.2 GRADIENT DESCENT

**Theorem.** *Under assumptions (A1)-(A5) for gradient descent, the following holds for SWN:*

*(i)* $\|\widetilde{\mathbf{w}}_u\| > 0, \|\widetilde{\mathbf{w}}_v\| > 0 \implies \lim_{t \to \infty} \frac{\|\mathbf{w}_u(t)\|\|\nabla_{\mathbf{w}_v}\mathcal{L}(\mathbf{w}(t))\|}{\|\mathbf{w}_v(t)\|\|\nabla_{\mathbf{w}_u}\mathcal{L}(\mathbf{w}(t))\|} = 1$

*Proof.* Consider $u$ and $v$ such that $\|\widetilde{\mathbf{w}}_u\| > 0$ and $\|\widetilde{\mathbf{w}}_s\| > 0$. Using Proposition 2, we can say $\|\widetilde{\mathbf{g}}_u\| > 0$ and $\|\widetilde{\mathbf{g}}_s\| > 0$. Also, from Proposition 2, we can say, for both $u$ and $s$, weights and gradients converge in opposite directions. Now from Equation (19), we can say

$$\gamma_u(t) = \gamma_u(0) - \sum_{k=0}^{k=t-1} \eta(k) \frac{\mathbf{v}_u(k)^\top \nabla_{\mathbf{w}_u}\mathcal{L}(\mathbf{w}(k))}{\|\mathbf{v}_u(k)\|}$$

$$\gamma_s(t) = \gamma_s(0) - \sum_{k=0}^{k=t-1} \eta(k) \frac{\mathbf{v}_s(k)^\top \nabla_{\mathbf{w}_s}\mathcal{L}(\mathbf{w}(k))}{\|\mathbf{v}_s(k)\|}$$

Now, $\gamma_s(t)$ either diverges to $\infty$ or $-\infty$. In both the cases, it is a strictly monotonic sequence for large enough $t$. Also $\lim_{t \to \infty} \frac{\gamma_u(t+1)-\gamma_u(t)}{\gamma_s(t+1)-\gamma_s(t)}$ exists. Therefore, using Stolz-Cesaro Theorem, we can say

$$\lim_{t \to \infty} \frac{\|\mathbf{w}_u(t)\|\|\nabla_{\mathbf{w}_s(t)}\mathcal{L}\|}{\|\mathbf{w}_s(t)\|\|\nabla_{\mathbf{w}_u(t)}\mathcal{L}\|} = 1$$

$\square$

## E    PROOF OF PROPOSITION 3

**Proposition.** *Let assumptions (A1)-(A5) be satisfied. Consider two nodes $u$ and $v$ in the network such that $\|\widetilde{\mathbf{g}}_v\| \geq \|\widetilde{\mathbf{g}}_u\| > 0$ and $\|\mathbf{w}_u(t)\|, \|\mathbf{w}_v(t)\| \to \infty$. Let $\frac{\|\widetilde{\mathbf{g}}_u\|}{\|\widetilde{\mathbf{g}}_v\|}$ be denoted by $c$. Let $\epsilon, \delta$ be such that $0 < \epsilon < c$ and $0 < \delta < 2\pi$. Then, the following holds:*

1. *There exists a time $t_1$, such that for all $t > t_1$ both SWN and EWN trajectories have the following properties:*

   (a) $\frac{\|\nabla_{\mathbf{w}_u}\mathcal{L}(\mathbf{w}(t))\|}{\|\nabla_{\mathbf{w}_v}\mathcal{L}(\mathbf{w}(t))\|} \in [c - \epsilon, c + \epsilon]$

   (b) $\left(\frac{\mathbf{w}_u(t)}{\|\mathbf{w}_u(t)\|}\right)^\top \left(\frac{-\nabla_{\mathbf{w}_u}\mathcal{L}(\mathbf{w}(t))}{\|\nabla_{\mathbf{w}_u}\mathcal{L}(\mathbf{w}(t))\|}\right) \geq \cos(\delta)$

   (c) $\left(\frac{\mathbf{w}_v(t)}{\|\mathbf{w}_v(t)\|}\right)^\top \left(\frac{-\nabla_{\mathbf{w}_v}\mathcal{L}(\mathbf{w}(t))}{\|\nabla_{\mathbf{w}_v}\mathcal{L}(\mathbf{w}(t))\|}\right) \geq \cos(\delta).$

2. *for SWN, $\lim_{t \to \infty} \frac{\|\mathbf{w}_u(t)\|}{\|\mathbf{w}_v(t)\|} = c$*

3. *for EWN, if at some time $t_2 > t_1$,*

   (a) $\frac{\|\mathbf{w}_u(t_2)\|}{\|\mathbf{w}_v(t_2)\|} > \frac{1}{(c-\epsilon)\cos(\delta)} \implies \lim_{t \to \infty} \frac{\|\mathbf{w}_u(t)\|}{\|\mathbf{w}_v(t)\|} = \infty$

   (b) $\frac{\|\mathbf{w}_u(t_2)\|}{\|\mathbf{w}_v(t_2)\|} < \frac{\cos(\delta)}{c+\epsilon} \implies \lim_{t \to \infty} \frac{\|\mathbf{w}_u(t)\|}{\|\mathbf{w}_v(t)\|} = 0$

The proof of different cases will be split into multiple subsections and corresponding proposition will be stated there for ease of the reader.

### E.1    STANDARD WEIGHT NORMALIZATION

### E.1.1    GRADIENT FLOW

**Proposition.** *Let assumptions (A1)-(A4) be satisfied. Consider two nodes $u$ and $v$ in the network such that $\|\widetilde{\mathbf{g}}_v\| \geq \|\widetilde{\mathbf{g}}_u\| > 0$ and $\|\mathbf{w}_u(t)\|, \|\mathbf{w}_v(t)\| \to \infty$. Let $\frac{\|\widetilde{\mathbf{g}}_u\|}{\|\widetilde{\mathbf{g}}_v\|}$ be denoted by $c$. Let $\epsilon, \delta$ be such that $0 < \epsilon < c$ and $0 < \delta < 2\pi$. Then, the following holds:*

1. *There exists a time $t_1$, such that for all $t > t_1$, SWN trajectory has the following properties:*

   (a) $\frac{\|\nabla_{\mathbf{w}_u}\mathcal{L}(\mathbf{w}(t))\|}{\|\nabla_{\mathbf{w}_v}\mathcal{L}(\mathbf{w}(t))\|} \in [c - \epsilon, c + \epsilon]$

   (b) $\left(\frac{\mathbf{w}_u(t)}{\|\mathbf{w}_u(t)\|}\right)^\top \left(\frac{-\nabla_{\mathbf{w}_u}\mathcal{L}(\mathbf{w}(t))}{\|\nabla_{\mathbf{w}_u}\mathcal{L}(\mathbf{w}(t))\|}\right) \geq \cos(\delta)$

   (c) $\left(\frac{\mathbf{w}_v(t)}{\|\mathbf{w}_v(t)\|}\right)^\top \left(\frac{-\nabla_{\mathbf{w}_v}\mathcal{L}(\mathbf{w}(t))}{\|\nabla_{\mathbf{w}_v}\mathcal{L}(\mathbf{w}(t))\|}\right) \geq \cos(\delta).$

2. $\lim_{t\to\infty} \frac{\|\mathbf{w}_u(t)\|}{\|\mathbf{w}_v(t)\|} = c$

*Proof.* The proof of part 1a, i.e. $\frac{\|\nabla_{\mathbf{w}_u}\mathcal{L}(\mathbf{w}(t))\|}{\|\nabla_{\mathbf{w}_v}\mathcal{L}(\mathbf{w}(t))\|} \in [c - \epsilon, c + \epsilon]$ follows from the definition of limit as $\frac{\|\nabla_{\mathbf{w}_u}\mathcal{L}(\mathbf{w}(t))\|}{\|\nabla_{\mathbf{w}_v}\mathcal{L}(\mathbf{w}(t))\|}$ tends to $c$.

Now, we will move to the proof of part 1b, i.e., $\left(\frac{\mathbf{w}_u(t)}{\|\mathbf{w}_u(t)\|}\right)^\top \left(\frac{-\nabla_{\mathbf{w}_u}\mathcal{L}(\mathbf{w}(t))}{\|\nabla_{\mathbf{w}_u}\mathcal{L}(\mathbf{w}(t))\|}\right) \geq \cos(\delta)$. The assumptions in this Proposition differ slightly from Proposition 2 and thus the proof is slightly more involved as we also need to show that $\mathbf{w}_u(t)$ converges in direction. The proof will be given for $\gamma_u \to \infty$. The one for $\gamma_u \to -\infty$ can be handled similarly.

As $\|\widetilde{\mathbf{g}}_u\| > 0$, therefore $\nabla_{\mathbf{w}_u}\mathcal{L}(\mathbf{w}(t))$ converges in direction. Therefore, for every $\tau$ satisfying $0 < \tau < 2\pi$, there exists a time $t_3$, such that for $t > t_3$, $\left(\frac{-\nabla_{\mathbf{w}_u}\mathcal{L}(\mathbf{w}(t))}{\|\nabla_{\mathbf{w}_u}\mathcal{L}(\mathbf{w}(t))\|}\right)^\top \left(\frac{\widetilde{\mathbf{g}}_u}{\|\widetilde{\mathbf{g}}_u\|}\right) \geq \cos(\tau)$. Now, Let's assume that $\mathbf{w}_u(t)$ does not converge in the direction of $\widetilde{\mathbf{g}}_u$. Then, there must exist a $\tau$ satisfying $0 < \tau < 2\pi$, such that for this $\tau$, there exists a time $t_4 > t_3$ satisfying $\mathbf{v}_u(t_4)^\top \left(\frac{\widetilde{\mathbf{g}}_u}{\|\widetilde{\mathbf{g}}_u\|}\right) = \cos(\Delta)$, where $\Delta > \tau$.

Now, we are going to show that for any $\kappa$ satisfying $\tau < \kappa < \Delta$, there exists a time $t_5 > t_4$ such that $\mathbf{v}_u(t_5)^\top \left(\frac{\widetilde{\mathbf{g}}_u}{\|\widetilde{\mathbf{g}}_u\|}\right) > \cos(\kappa)$. Let's say for a given $\kappa$, no such $t_5$ exists. Then, taking dot product with $\frac{\widetilde{\mathbf{g}}_u}{\|\widetilde{\mathbf{g}}_u\|}$ on both sides of Equation (17), we can say

$$\left(\frac{\widetilde{\mathbf{g}}_u}{\|\widetilde{\mathbf{g}}_u\|}\right)^\top \frac{d\mathbf{v}_u(t)}{dt} = \eta(t)\gamma_u(t)\|\nabla_{\mathbf{w}_u}\mathcal{L}(\mathbf{w}(t))\| \left(\frac{\widetilde{\mathbf{g}}_u}{\|\widetilde{\mathbf{g}}_u\|}\right)^\top (I - \mathbf{v}_u(t)\mathbf{v}_u(t)^\top)\left(\frac{-\nabla_{\mathbf{w}_u}\mathcal{L}(\mathbf{w}(t))}{\|\nabla_{\mathbf{w}_u}\mathcal{L}(\mathbf{w}(t))\|}\right)$$

Now, as $\left(\frac{\widetilde{\mathbf{g}}_u}{\|\widetilde{\mathbf{g}}_u\|}\right)^\top \left(\frac{-\nabla_{\mathbf{w}_u}\mathcal{L}(\mathbf{w}(t))}{\|\nabla_{\mathbf{w}_u}\mathcal{L}(\mathbf{w}(t))\|}\right) \geq \cos(\tau)$ and $\left(\frac{\widetilde{\mathbf{g}}_u}{\|\widetilde{\mathbf{g}}_u\|}\right)^\top \mathbf{v}_u \leq \cos(\kappa)$, we can say

$$\left(\frac{\widetilde{\mathbf{g}}_u}{\|\widetilde{\mathbf{g}}_u\|}\right)^\top \frac{d\mathbf{v}_u(t)}{dt} \geq \eta(t)\gamma_u(t)\|\nabla_{\mathbf{w}_u}\mathcal{L}(\mathbf{w}(t))\|(\cos(\tau) - \cos(\kappa)) \tag{24}$$

Now, using the fact that $\gamma_u \to \infty$ and using Equation (16), we can say

$$\int_{t=t_4}^\infty \eta(t)\|\nabla_{\mathbf{w}_u}\mathcal{L}(\mathbf{w}(t))\|dt = \infty$$

Using this fact and integrating the Equation (24) on both the sides from $t_4$ to $\infty$, we get a contradiction as vectors on LHS have a finite norm while RHS tends to $\infty$. Thus, for every $\kappa$ between $\tau$ and $\Delta$, there must exist a $t_5$, such that $\mathbf{v}_u(t_5)^\top \left(\frac{\widetilde{\mathbf{g}}_u}{\|\widetilde{\mathbf{g}}_u\|}\right) > \cos(\kappa)$.

Now, we are going to show for all $t \geq t_5$, $\mathbf{v}_u(t)^\top \left(\frac{\widetilde{\mathbf{g}}_u}{\|\widetilde{\mathbf{g}}_u\|}\right) > \cos(\kappa)$. Now, consider any $\beta$ such that $\tau < \beta < \kappa$. Using similar argument as in Equation (24), we can say, if for any $t_6 > t_5$, $\mathbf{v}_u(t_6)^\top \left(\frac{\widetilde{\mathbf{g}}_u}{\|\widetilde{\mathbf{g}}_u\|}\right) < \cos(\beta)$, then

$$\left(\frac{\widetilde{\mathbf{g}}_u}{\|\widetilde{\mathbf{g}}_u\|}\right)^\top \frac{d\mathbf{v}_u(t_6)}{dt} \geq \eta(t_6)\gamma_u(t_6)\|\nabla_{\mathbf{w}_u}\mathcal{L}(\mathbf{w}(t_6))\|(\cos(\tau) - \cos(\beta)) \tag{25}$$

This means that the dot product between $\left(\frac{\widetilde{\mathbf{g}}_u}{\|\widetilde{\mathbf{g}}_u\|}\right)$ and $\mathbf{v}_u(t)$ goes up, whenever $\left(\frac{\widetilde{\mathbf{g}}_u}{\|\widetilde{\mathbf{g}}_u\|}\right)^\top \mathbf{v}_u(t) < \cos(\tau)$. Therefore, its not possible that $\mathbf{v}_u(t)^\top \left(\frac{\widetilde{\mathbf{g}}_u}{\|\widetilde{\mathbf{g}}_u\|}\right) \leq \cos(\kappa)$ for any $t > t_5$. As $\kappa$ can be

arbitrarily chosen between $\tau$ and $\Delta$, and the argument holds for any $\epsilon > 0$, $\mathbf{w}_u(t)$ converges in the direction of $\widetilde{\mathbf{g}}_u$

The proof of part 1c, i.e, $\left(\frac{\mathbf{w}_v(t)}{\|\mathbf{w}_v(t)\|}\right)^{\top}\left(\frac{-\nabla_{\mathbf{w}_v}\mathcal{L}(\mathbf{w}(t))}{\|\nabla_{\mathbf{w}_v}\mathcal{L}(\mathbf{w}(t))\|}\right) \geq \cos(\delta)$ can be shown in the same way as 1b.

The proof of part 2, i.e, $\lim_{t\to\infty}\frac{\|\mathbf{w}_u(t)\|}{\|\mathbf{w}_v(t)\|} = c$ can be shown in the same way as Theorem 2 for SWN gradient flow from Appendix D.2.1. $\qquad\square$

### E.1.2 GRADIENT DESCENT

**Proposition.** *Let assumptions (A1)-(A5) be satisfied. Consider two nodes $u$ and $v$ in the network such that $\|\widetilde{\mathbf{g}}_v\| \geq \|\widetilde{\mathbf{g}}_u\| > 0$ and $\|\mathbf{w}_u(t)\|, \|\mathbf{w}_v(t)\| \to \infty$. Let $\frac{\|\widetilde{\mathbf{g}}_u\|}{\|\widetilde{\mathbf{g}}_v\|}$ be denoted by c. Let $\epsilon, \delta$ be such that $0 < \epsilon < c$ and $0 < \delta < 2\pi$. Then, the following holds:*

*1. There exists a time $t_1$, such that for all $t > t_1$, SWN trajectory has the following properties:*

(a) $\frac{\|\nabla_{\mathbf{w}_u}\mathcal{L}(\mathbf{w}(t))\|}{\|\nabla_{\mathbf{w}_v}\mathcal{L}(\mathbf{w}(t))\|} \in [c - \epsilon, c + \epsilon]$

(b) $\left(\frac{\mathbf{w}_u(t)}{\|\mathbf{w}_u(t)\|}\right)^{\top}\left(\frac{-\nabla_{\mathbf{w}_u}\mathcal{L}(\mathbf{w}(t))}{\|\nabla_{\mathbf{w}_u}\mathcal{L}(\mathbf{w}(t))\|}\right) \geq \cos(\delta)$

(c) $\left(\frac{\mathbf{w}_v(t)}{\|\mathbf{w}_v(t)\|}\right)^{\top}\left(\frac{-\nabla_{\mathbf{w}_v}\mathcal{L}(\mathbf{w}(t))}{\|\nabla_{\mathbf{w}_v}\mathcal{L}(\mathbf{w}(t))\|}\right) \geq \cos(\delta).$

*2. $\lim_{t\to\infty}\frac{\|\mathbf{w}_u(t)\|}{\|\mathbf{w}_v(t)\|} = c$*

*Proof.* The proof of part 1a, i.e, $\frac{\|\nabla_{\mathbf{w}_u}\mathcal{L}(\mathbf{w}(t))\|}{\|\nabla_{\mathbf{w}_v}\mathcal{L}(\mathbf{w}(t))\|} \in [c - \epsilon, c + \epsilon]$ follows from the definition of limit as $\frac{\|\nabla_{\mathbf{w}_u}\mathcal{L}(\mathbf{w}(t))\|}{\|\nabla_{\mathbf{w}_v}\mathcal{L}(\mathbf{w}(t))\|}$ tends to $c$.

Now, we will move to the proof of part 1b, i.e, $\left(\frac{\mathbf{w}_u(t)}{\|\mathbf{w}_u(t)\|}\right)^{\top}\left(\frac{-\nabla_{\mathbf{w}_u}\mathcal{L}(\mathbf{w}(t))}{\|\nabla_{\mathbf{w}_u}\mathcal{L}(\mathbf{w}(t))\|}\right) \geq \cos(\delta)$. The assumptions in this Proposition differ slightly from Proposition 2 and thus the proof is slightly more involved as we also need to show that $\mathbf{w}_u(t)$ converges in direction. The proof will be given for $\gamma_u \to \infty$. The one for $\gamma_u \to -\infty$ can be handled similarly.

As $\|\widetilde{\mathbf{g}}_u\| > 0$, therefore $\nabla_{\mathbf{w}_u}\mathcal{L}(\mathbf{w}(t))$ converges in direction. Therefore, for every $\tau$ satisfying $0 < \tau < 2\pi$, there exists a time $t_3$, such that for $t > t_3$, $\left(\frac{-\nabla_{\mathbf{w}_u}\mathcal{L}(\mathbf{w}(t))}{\|\nabla_{\mathbf{w}_u}\mathcal{L}(\mathbf{w}(t))\|}\right)^{\top}\left(\frac{\widetilde{\mathbf{g}}_u}{\|\widetilde{\mathbf{g}}_u\|}\right) \geq \cos(\tau)$. Now, Let's assume that $\mathbf{w}_u(t)$ does not converge in the direction of $\widetilde{\mathbf{g}}_u$. Then, there must exist a $\tau$ satisfying $0 < \tau < 2\pi$, such that for this $\tau$, there exists a time $t_4 > t_3$ satisfying $\mathbf{v}_u(t_4)^{\top}\left(\frac{\widetilde{\mathbf{g}}_u}{\|\widetilde{\mathbf{g}}_u\|}\right) = \cos(\Delta)$, where $\Delta > \tau$.

Now, we are going to show that for any $\kappa$ satisfying $\tau < \kappa < \Delta$, there exists a time $t_5 > t_4$ such that $\left(\frac{\mathbf{v}_u(t_5)}{\|\mathbf{v}_u(t_5)\|}\right)^{\top}\left(\frac{\widetilde{\mathbf{g}}_u}{\|\widetilde{\mathbf{g}}_u\|}\right) > \cos(\kappa)$. Let's say for a given $\kappa$, no such $t_5$ exists. Then, taking dot product with $\frac{\widetilde{\mathbf{g}}_u}{\|\widetilde{\mathbf{g}}_u\|}$ on both sides of Equation (20), we can say

$$\frac{\mathbf{v}_u(t+1)^{\top}\widetilde{\mathbf{g}}_u}{\|\widetilde{\mathbf{g}}_u\|} = \frac{\mathbf{v}_u(t)^{\top}\widetilde{\mathbf{g}}_u}{\|\widetilde{\mathbf{g}}_u\|} +$$

$$\eta(t)\frac{\gamma_u(t)}{\|\mathbf{v}_u(t)\|}\|\nabla_{\mathbf{w}_u}\mathcal{L}(\mathbf{w}(t))\|\left(\frac{\widetilde{\mathbf{g}}_u}{\|\widetilde{\mathbf{g}}_u\|}\right)^{\top}(I - \frac{\mathbf{v}_u(t)\mathbf{v}_u(t)^{\top}}{\|\mathbf{v}_u(t)\|^2})\left(\frac{-\nabla_{\mathbf{w}_u}\mathcal{L}(\mathbf{w}(t))}{\|\nabla_{\mathbf{w}_u}\mathcal{L}(\mathbf{w}(t))\|}\right)$$

Now, as $\left(\frac{\widetilde{\mathbf{g}}_u}{\|\widetilde{\mathbf{g}}_u\|}\right)^{\top}\left(\frac{-\nabla_{\mathbf{w}_u}\mathcal{L}(\mathbf{w}(t))}{\|\nabla_{\mathbf{w}_u}\mathcal{L}(\mathbf{w}(t))\|}\right) \geq \cos(\tau)$ and $\left(\frac{\widetilde{\mathbf{g}}_u}{\|\widetilde{\mathbf{g}}_u\|}\right)^{\top}\left(\frac{\mathbf{v}_u(t)}{\|\mathbf{v}_u(t)\|}\right) \leq \cos(\kappa)$, we can say

$$\frac{\mathbf{v}_u(t+1)^{\top}\widetilde{\mathbf{g}}_u}{\|\widetilde{\mathbf{g}}_u\|} \geq \frac{\mathbf{v}_u(t)^{\top}\widetilde{\mathbf{g}}_u}{\|\widetilde{\mathbf{g}}_u\|} + (\cos(\tau) - \cos(\kappa))(\eta(t)\frac{\gamma_u(t)}{\|\mathbf{v}_u(t)\|}\|\nabla_{\mathbf{w}_u}\mathcal{L}(\mathbf{w}(t))\|) \qquad (26)$$

Now, using arguments similar to the proof of Proposition 2 for SWN gradient descent in Appendix C.2.2, we can show that the above statement leads to a contradiction and thus there must exist a $t_5$ such that $\left(\frac{\mathbf{v}_u(t_5)}{\|\mathbf{v}_u(t_5)\|}\right)^\top \left(\frac{\widetilde{\mathbf{g}}_u}{\|\widetilde{\mathbf{g}}_u\|}\right) > \cos(\kappa)$.

Now, we are going to show that there exists a $t_6 > t_5$, such that for all $t > t_6$, $\left(\frac{\mathbf{v}_u(t)}{\|\mathbf{v}_u(t)\|}\right)^\top \left(\frac{\widetilde{\mathbf{g}}_u}{\|\widetilde{\mathbf{g}}_u\|}\right) > \cos(\kappa)$. Consider a $\beta$ such that $\tau < \beta < \kappa$. Now, if at any time t, $\left(\frac{\mathbf{v}_u(t)}{\|\mathbf{v}_u(t)\|}\right)^\top \left(\frac{\widetilde{\mathbf{g}}_u}{\|\widetilde{\mathbf{g}}_u\|}\right) < \cos(\beta)$, then, similar to Equation (26), we can say

$$\frac{\mathbf{v}_u(t+1)^\top \widetilde{\mathbf{g}}_u}{\|\widetilde{\mathbf{g}}_u\|} \geq \frac{\mathbf{v}_u(t)^\top \widetilde{\mathbf{g}}_u}{\|\widetilde{\mathbf{g}}_u\|} + (\cos(\tau) - \cos(\beta))(\eta(t)\frac{\gamma_u(t)}{\|\mathbf{v}_u(t)\|}\|\nabla_{\mathbf{w}_u}\mathcal{L}(\mathbf{w}(t))\|)$$

Using the upper bound on $\|\mathbf{v}_u(t+1)\|$ from Equation (21), we can say

$$\frac{\mathbf{v}_u(t+1)^\top \widetilde{\mathbf{g}}_u}{\|\mathbf{v}_u(t+1)\|\|\widetilde{\mathbf{g}}_u\|} \geq \frac{\frac{\mathbf{v}_u(t)^\top \widetilde{\mathbf{g}}_u}{\|\widetilde{\mathbf{g}}_u\|} + (\cos(\tau) - \cos(\beta))(\eta(t)\frac{\gamma_u(t)}{\|\mathbf{v}_u(t)\|}\|\nabla_{\mathbf{w}_u}\mathcal{L}(\mathbf{w}(t))\|)}{\sqrt{\|\mathbf{v}_u(t)\|^2 + (\eta(t)\frac{\gamma_u(t)}{\|\mathbf{v}_u(t)\|}\|\nabla_{\mathbf{w}_u}\mathcal{L}(\mathbf{w}(t))\|)^2}} \quad (27)$$

Let $\eta(t)\frac{\gamma_u(t)}{\|\mathbf{v}_u(t)\|}\|\nabla_{\mathbf{w}_u}\mathcal{L}(\mathbf{w}(t))\|$ be denoted by $\chi(t)$. Then, the above equation can be rewritten as

$$\frac{\mathbf{v}_u(t+1)^\top \widetilde{\mathbf{g}}_u}{\|\mathbf{v}_u(t+1)\|\|\widetilde{\mathbf{g}}_u\|} \geq \frac{\mathbf{v}_u(t)^\top \widetilde{\mathbf{g}}_u}{\|\mathbf{v}_u(t)\|\|\widetilde{\mathbf{g}}_u\|}\frac{\|\mathbf{v}_u(t)\|}{\sqrt{\|\mathbf{v}_u(t)\|^2 + \chi(t)^2}} + (\cos(\tau) - \cos(\beta))\frac{\chi(t)}{\sqrt{\|\mathbf{v}_u(t)\|^2 + \chi(t)^2}}$$

Now, we are going to show that for a small enough $\chi(t)$, RHS is greater than $\frac{\mathbf{v}_u(t)^\top \widetilde{\mathbf{g}}_u}{\|\mathbf{v}_u(t)\|\|\widetilde{\mathbf{g}}_u\|}$.

$$\frac{\mathbf{v}_u(t)^\top \widetilde{\mathbf{g}}_u}{\|\mathbf{v}_u(t)\|\|\widetilde{\mathbf{g}}_u\|}\frac{\|\mathbf{v}_u(t)\|}{\sqrt{\|\mathbf{v}_u(t)\|^2 + \chi(t)^2}} + (\cos(\tau) - \cos(\beta))\frac{\chi(t)}{\sqrt{\|\mathbf{v}_u(t)\|^2 + \chi(t)^2}} > \frac{\mathbf{v}_u(t)^\top \widetilde{\mathbf{g}}_u}{\|\mathbf{v}_u(t)\|\|\widetilde{\mathbf{g}}_u\|}$$

$$\implies (\cos(\tau) - \cos(\beta))\frac{\chi(t)}{\sqrt{\|\mathbf{v}_u(t)\|^2 + \chi(t)^2}} > \frac{\mathbf{v}_u(t)^\top \widetilde{\mathbf{g}}_u}{\|\mathbf{v}_u(t)\|\|\widetilde{\mathbf{g}}_u\|}(1 - \frac{\|\mathbf{v}_u(t)\|}{\sqrt{\|\mathbf{v}_u(t)\|^2 + \chi(t)^2}})$$

$$\implies (\cos(\tau) - \cos(\beta)) > \frac{\mathbf{v}_u(t)^\top \widetilde{\mathbf{g}}_u}{\|\mathbf{v}_u(t)\|\|\widetilde{\mathbf{g}}_u\|}(\frac{\sqrt{\|\mathbf{v}_u(t)\|^2 + \chi(t)^2} - \|\mathbf{v}_u(t)\|}{\chi(t)})$$

Clearly as $\chi(t) \to 0$, the RHS tends to 0, therefore the equation is satisfied. Thus for a small enough $\chi(t)$, RHS of Equation (27) is greater than $\frac{\mathbf{v}_u(t)^\top \widetilde{\mathbf{g}}_u}{\|\mathbf{v}_u(t)\|\|\widetilde{\mathbf{g}}_u\|}$. As $\|\mathbf{v}_u(t)\|$ keeps on increasing and by Assumption (A5), $\lim_{t\to\infty} \eta(t)\gamma_u(t)\|\nabla_{\mathbf{w}_u}\mathcal{L}(\mathbf{w}(t))\| = 0$, we can say there exists a time $t_7$, such that for any $t > t_7$, $\frac{\mathbf{v}_u(t)^\top \widetilde{\mathbf{g}}_u}{\|\mathbf{v}_u(t)\|\|\widetilde{\mathbf{g}}_u\|}$ goes up whenever $\left(\frac{\mathbf{v}_u(t)}{\|\mathbf{v}_u(t)\|}\right)^\top \left(\frac{\widetilde{\mathbf{g}}_u}{\|\widetilde{\mathbf{g}}_u\|}\right) < \cos(\beta)$.

Also, by using Equation (20) and Assumption (A5), we can say, that there exists a time $t_8$, such that for $t > t_8$, $\left(\frac{\mathbf{v}_u(t)}{\|\mathbf{v}_u(t)\|}\right)^\top \left(\frac{\widetilde{\mathbf{g}}_u}{\|\widetilde{\mathbf{g}}_u\|}\right) > \cos(\beta) \implies \left(\frac{\mathbf{v}_u(t+1)}{\|\mathbf{v}_u(t+1)\|}\right)^\top \left(\frac{\widetilde{\mathbf{g}}_u}{\|\widetilde{\mathbf{g}}_u\|}\right) > \cos(\kappa)$, as the RHS of Equation (20) goes to 0 norm in limit. Now, define $t_6 > max(t_7, t_8)$ such that $\left(\frac{\mathbf{v}_u(t_6)}{\|\mathbf{v}_u(t_6)\|}\right)^\top \left(\frac{\widetilde{\mathbf{g}}_u}{\|\widetilde{\mathbf{g}}_u\|}\right) > \cos(\kappa)$ (must exist from previous arguments). Then, as the dot product always goes up when between $\cos(\beta)$ and $\cos(\kappa)$, and can't go in a single step from being greater than $\cos(\beta)$ to less than $\cos(\kappa)$, therefore, for every $t > t_6$, $\left(\frac{\mathbf{v}_u(t)}{\|\mathbf{v}_u(t)\|}\right)^\top \left(\frac{\widetilde{\mathbf{g}}_u}{\|\widetilde{\mathbf{g}}_u\|}\right) > \cos(\kappa)$.

Now as the above argument holds for any $\kappa$ between $\tau$ and $\Delta$, and for any $\tau > 0$, we can say that $\mathbf{w}_u(t)$ converges in direction of $\widetilde{\mathbf{g}}_u$.

The proof of part 1c, i.e, $\left(\frac{\mathbf{w}_v(t)}{\|\mathbf{w}_v(t)\|}\right)^\top \left(\frac{-\nabla_{\mathbf{w}_v}\mathcal{L}(\mathbf{w}(t))}{\|\nabla_{\mathbf{w}_v}\mathcal{L}(\mathbf{w}(t))\|}\right) \geq \cos(\delta)$, follows exactly the same steps as part 1b.

The proof of part 2, i.e, $\lim_{t\to\infty}\frac{\|\mathbf{w}_u(t)\|}{\|\mathbf{w}_v(t)\|} = c$ can be shown in the same way as the proof of Theorem 2 for SWN gradient descent from Appendix D.2.2. $\qquad\square$

### E.2 EXPONENTIAL WEIGHT NORMALIZATION

#### E.2.1 GRADIENT FLOW

**Proposition.** *Let assumptions (A1)-(A4) be satisfied. Consider two nodes $u$ and $v$ in the network such that $\|\widetilde{\mathbf{g}}_v\| \geq \|\widetilde{\mathbf{g}}_u\| > 0$ and $\|\mathbf{w}_u(t)\|, \|\mathbf{w}_v(t)\| \to \infty$. Let $\frac{\|\widetilde{\mathbf{g}}_u\|}{\|\widetilde{\mathbf{g}}_v\|}$ be denoted by $c$. Let $\epsilon, \delta$ be such that $0 < \epsilon < c$ and $0 < \delta < 2\pi$. Then, the following holds:*

1. *There exists a time $t_1$, such that for all $t > t_1$, EWN trajectory has the following properties:*

   (a) $\frac{\|\nabla_{\mathbf{w}_u}\mathcal{L}(\mathbf{w}(t))\|}{\|\nabla_{\mathbf{w}_v}\mathcal{L}(\mathbf{w}(t))\|} \in [c - \epsilon, c + \epsilon]$

   (b) $\left( \frac{\mathbf{w}_u(t)}{\|\mathbf{w}_u(t)\|} \right)^\top \left( \frac{-\nabla_{\mathbf{w}_u}\mathcal{L}(\mathbf{w}(t))}{\|\nabla_{\mathbf{w}_u}\mathcal{L}(\mathbf{w}(t))\|} \right) \geq \cos(\delta)$

   (c) $\left( \frac{\mathbf{w}_v(t)}{\|\mathbf{w}_v(t)\|} \right)^\top \left( \frac{-\nabla_{\mathbf{w}_v}\mathcal{L}(\mathbf{w}(t))}{\|\nabla_{\mathbf{w}_v}\mathcal{L}(\mathbf{w}(t))\|} \right) \geq \cos(\delta).$

2. *If at some time $t_2 > t_1$,*

   (a) $\frac{\|\mathbf{w}_u(t_2)\|}{\|\mathbf{w}_v(t_2)\|} > \frac{1}{(c-\epsilon)\cos(\delta)} \implies \lim_{t\to\infty} \frac{\|\mathbf{w}_u(t)\|}{\|\mathbf{w}_v(t)\|} = \infty$

   (b) $\frac{\|\mathbf{w}_u(t_2)\|}{\|\mathbf{w}_v(t_2)\|} < \frac{\cos(\delta)}{c+\epsilon} \implies \lim_{t\to\infty} \frac{\|\mathbf{w}_u(t)\|}{\|\mathbf{w}_v(t)\|} = 0$

*Proof.* The proof of part 1a, i.e, $\frac{\|\nabla_{\mathbf{w}_u}\mathcal{L}(\mathbf{w}(t))\|}{\|\nabla_{\mathbf{w}_v}\mathcal{L}(\mathbf{w}(t))\|} \in [c - \epsilon, c + \epsilon]$ follows from the definition of limit as $\frac{\|\nabla_{\mathbf{w}_u}\mathcal{L}(\mathbf{w}(t))\|}{\|\nabla_{\mathbf{w}_v}\mathcal{L}(\mathbf{w}(t))\|}$ tends to $c$.

Now, we will move to the proof of part 1b, i.e., $\left( \frac{\mathbf{w}_u(t)}{\|\mathbf{w}_u(t)\|} \right)^\top \left( \frac{-\nabla_{\mathbf{w}_u}\mathcal{L}(\mathbf{w}(t))}{\|\nabla_{\mathbf{w}_u}\mathcal{L}(\mathbf{w}(t))\|} \right) \geq \cos(\delta)$. The assumptions in this Proposition differ slightly from Proposition 2 and thus the proof is slightly more involved as we also need to show that $\mathbf{w}_u(t)$ converges in direction.

As $\|\widetilde{\mathbf{g}}_u\| > 0$, therefore $\nabla_{\mathbf{w}_u}\mathcal{L}(\mathbf{w}(t))$ converges in direction. Therefore, for every $\tau$ satisfying $0 < \tau < 2\pi$, there exists a time $t_3$, such that for $t > t_3$, $\left( \frac{-\nabla_{\mathbf{w}_u}\mathcal{L}(\mathbf{w}(t))}{\|\nabla_{\mathbf{w}_u}\mathcal{L}(\mathbf{w}(t))\|} \right)^\top \left( \frac{\widetilde{\mathbf{g}}_u}{\|\widetilde{\mathbf{g}}_u\|} \right) \geq \cos(\tau)$. Now, Let's assume that $\mathbf{w}_u(t)$ does not converge in the direction of $\widetilde{\mathbf{g}}_u$. Then, there must exist a $\tau$ satisfying $0 < \tau < 2\pi$, such that for this $\tau$, there exists a time $t_4 > t_3$ satisfying $\mathbf{v}_u(t_4)^\top \left( \frac{\widetilde{\mathbf{g}}_u}{\|\widetilde{\mathbf{g}}_u\|} \right) = \cos(\Delta)$, where $\Delta > \tau$.

Now, we are going to show that for any $\kappa$ satisfying $\tau < \kappa < \Delta$, there exists a time $t_5 > t_4$ such that $\mathbf{v}_u(t_5)^\top \left( \frac{\widetilde{\mathbf{g}}_u}{\|\widetilde{\mathbf{g}}_u\|} \right) > \cos(\kappa)$. Let's say for a given $\kappa$, no such $t_5$ exists. Then, taking dot product with $\frac{\widetilde{\mathbf{g}}_u}{\|\widetilde{\mathbf{g}}_u\|}$ on both sides of Equation (8), we can say

$$\left( \frac{\widetilde{\mathbf{g}}_u}{\|\widetilde{\mathbf{g}}_u\|} \right)^\top \frac{d\mathbf{v}_u(t)}{dt} = \eta(t)e^{\alpha_u(t)}\|\nabla_{\mathbf{w}_u}\mathcal{L}(\mathbf{w}(t))\| \left( \frac{\widetilde{\mathbf{g}}_u}{\|\widetilde{\mathbf{g}}_u\|} \right)^\top (I - \mathbf{v}_u(t)\mathbf{v}_u(t)^\top) \left( \frac{-\nabla_{\mathbf{w}_u}\mathcal{L}(\mathbf{w}(t))}{\|\nabla_{\mathbf{w}_u}\mathcal{L}(\mathbf{w}(t))\|} \right)$$

Now, as $\left( \frac{\widetilde{\mathbf{g}}_u}{\|\widetilde{\mathbf{g}}_u\|} \right)^\top \left( \frac{-\nabla_{\mathbf{w}_u}\mathcal{L}(\mathbf{w}(t))}{\|\nabla_{\mathbf{w}_u}\mathcal{L}(\mathbf{w}(t))\|} \right) \geq \cos(\tau)$ and $\left( \frac{\widetilde{\mathbf{g}}_u}{\|\widetilde{\mathbf{g}}_u\|} \right)^\top \mathbf{v}_u \leq \cos(\kappa)$, we can say

$$\left( \frac{\widetilde{\mathbf{g}}_u}{\|\widetilde{\mathbf{g}}_u\|} \right)^\top \frac{d\mathbf{v}_u(t)}{dt} \geq \eta(t)e^{\alpha_u(t)}\|\nabla_{\mathbf{w}_u}\mathcal{L}(\mathbf{w}(t))\|(\cos(\tau) - \cos(\kappa)) \tag{28}$$

Now, using the fact that $\alpha_u \to \infty$ and using Equation (7), we can say

$$\int_{t=t_4}^{\infty} \eta(t)e^{\alpha_u(t)}\|\nabla_{\mathbf{w}_u}\mathcal{L}(\mathbf{w}(t))\|dt = \infty$$

Using this fact and integrating the Equation (28) on both the sides from $t_4$ to $\infty$, we get a contradiction as vectors on LHS have a finite norm while RHS tends to $\infty$. Thus, for every $\kappa$ between $\tau$ and $\Delta$, there must exist a $t_5$, such that $\mathbf{v}_u(t_5)^\top \left( \frac{\widetilde{\mathbf{g}}_u}{\|\widetilde{\mathbf{g}}_u\|} \right) > \cos(\kappa)$.

Now, we are going to show for all $t \geq t_5$, $\mathbf{v}_u(t)^\top \left( \frac{\widetilde{\mathbf{g}}_u}{\|\widetilde{\mathbf{g}}_u\|} \right) > \cos(\kappa)$. Now, consider any $\beta$ such that $\tau < \beta < \kappa$. Using similar argument as in Equation (28), we can say, if for any $t_6 > t_5$, $\mathbf{v}_u(t_6)^\top \left( \frac{\widetilde{\mathbf{g}}_u}{\|\widetilde{\mathbf{g}}_u\|} \right) < \cos(\beta)$, then

$$\left( \frac{\widetilde{\mathbf{g}}_u}{\|\widetilde{\mathbf{g}}_u\|} \right)^\top \frac{d\mathbf{v}_u(t_6)}{dt} \geq \eta(t_6) e^{\alpha_u(t_6)} \|\nabla_{\mathbf{w}_u} \mathcal{L}(\mathbf{w}(t_6))\|(\cos(\tau) - \cos(\beta)) \tag{29}$$

This means that the dot product between $\left( \frac{\widetilde{\mathbf{g}}_u}{\|\widetilde{\mathbf{g}}_u\|} \right)$ and $\mathbf{v}_u(t)$ goes up, whenever $\left( \frac{\widetilde{\mathbf{g}}_u}{\|\widetilde{\mathbf{g}}_u\|} \right)^\top \mathbf{v}_u(t) < \cos(\tau)$. Therefore, its not possible that $\mathbf{v}_u(t)^\top \left( \frac{\widetilde{\mathbf{g}}_u}{\|\widetilde{\mathbf{g}}_u\|} \right) \leq \cos(\kappa)$ for any $t > t_5$. As $\kappa$ can be arbitrarily chosen between $\tau$ and $\Delta$, and the argument holds for any $\epsilon > 0$, $\mathbf{w}_u(t)$ converges in the direction of $\widetilde{\mathbf{g}}_u$

The proof of part 1c, i.e, $\left( \frac{\mathbf{w}_v(t)}{\|\mathbf{w}_v(t)\|} \right)^\top \left( \frac{-\nabla_{\mathbf{w}_v} \mathcal{L}(\mathbf{w}(t))}{\|\nabla_{\mathbf{w}_v} \mathcal{L}(\mathbf{w}(t))\|} \right) \geq \cos(\delta)$ follows exactly the same steps as part 1b.

Now, we will move to the proof of part 2a, i.e., $\frac{\|\mathbf{w}_u(t_2)\|}{\|\mathbf{w}_v(t_2)\|} > \frac{1}{(c-\epsilon)\cos(\delta)} \implies \lim_{t \to \infty} \frac{\|\mathbf{w}_u(t)\|}{\|\mathbf{w}_v(t)\|} = \infty$

Using Equation (7),

$$\frac{d\|\mathbf{w}_u(t)\|}{dt} = \frac{de^{\alpha_u(t)}}{dt} = -\eta(t)\|\mathbf{w}_u(t)\|^2 (\mathbf{v}_u(t)^\top \nabla_{\mathbf{w}_u} \mathcal{L}(\mathbf{w}(t)) \tag{30}$$

Using the equation above and part 1 of the Proposition, we can say for $t > t_1$,

$$\frac{d\frac{\|\mathbf{w}_u(t)\|}{\|\mathbf{w}_v(t)\|}}{dt} = \frac{\|\mathbf{w}_v(t)\| \frac{d\|\mathbf{w}_u(t)\|}{dt} - \|\mathbf{w}_u(t)\| \frac{d\|\mathbf{w}_v(t)\|}{dt}}{\|\mathbf{w}_v(t)\|^2}$$
$$\geq \eta(t) \frac{\|\mathbf{w}_u(t)\|}{\|\mathbf{w}_v(t)\|} (\|\mathbf{w}_u(t)\|\|\nabla_{\mathbf{w}_u} \mathcal{L}(\mathbf{w}(t))\| \cos(\delta) - \|\mathbf{w}_v(t)\|\|\nabla_{\mathbf{w}_v} \mathcal{L}(\mathbf{w}(t))\|) \tag{31}$$

In this case, using Equation (31), we can see $\frac{d\frac{\|\mathbf{w}_u(t)\|}{\|\mathbf{w}_v(t)\|}}{dt} > 0$ at $t_2$. Thus, $\frac{\|\mathbf{w}_u(t)\|}{\|\mathbf{w}_v(t)\|}$ always remains greater than $\frac{1}{(c-\epsilon)\cos(\delta)}$ and keeps on increasing. Let's denote $\frac{\|\mathbf{w}_u(t_2)\|}{\|\mathbf{w}_v(t_2)\|}$ by $\Delta$. Then we can say

$$\frac{d\frac{\|\mathbf{w}_u(t)\|}{\|\mathbf{w}_v(t)\|}}{dt} \geq \Delta (\cos(\delta) - \frac{1}{\Delta(c-\epsilon)}) \eta(t)\|\mathbf{w}_u(t)\|\|\nabla_{\mathbf{w}_u} \mathcal{L}(\mathbf{w}(t))\|$$

As $\alpha_u \to \infty$, therefore using Equation (7), we can say $\int_{t_2}^\infty \eta(t)\|\mathbf{w}_u(t)\|\|\nabla_{\mathbf{w}_u} \mathcal{L}(\mathbf{w}(t))\|dt \to \infty$. Thus, integrating both the sides of the equation above from $t_2$ to $\infty$, we get

$$\int_{t_2}^\infty \frac{d\frac{\|\mathbf{w}_u(t)\|}{\|\mathbf{w}_v(t)\|}}{dt} dt \geq \infty$$

Thus $\lim_{t \to \infty} \frac{\|\mathbf{w}_u(t)\|}{\|\mathbf{w}_v(t)\|} = \infty$.

Now, we will move to the proof of part 2b, i.e, $\frac{\|\mathbf{w}_u(t_2)\|}{\|\mathbf{w}_v(t_2)\|} < \frac{\cos(\delta)}{c+\epsilon} \implies \lim_{t \to \infty} \frac{\|\mathbf{w}_u(t)\|}{\|\mathbf{w}_v(t)\|} = 0$.

Using Equation (30) and part 1 of the Proposition, we can say for $t > t_1$,

$$\frac{d\frac{\|\mathbf{w}_u(t)\|}{\|\mathbf{w}_v(t)\|}}{dt} = \frac{\|\mathbf{w}_v(t)\| \frac{d\|\mathbf{w}_u(t)\|}{dt} - \|\mathbf{w}_u(t)\| \frac{d\|\mathbf{w}_v(t)\|}{dt}}{\|\mathbf{w}_v(t)\|^2}$$
$$\leq \eta(t) \frac{\|\mathbf{w}_u(t)\|}{\|\mathbf{w}_v(t)\|} (\|\mathbf{w}_u(t)\|\|\nabla_{\mathbf{w}_u} \mathcal{L}(\mathbf{w}(t))\| - \|\mathbf{w}_v(t)\|\|\nabla_{\mathbf{w}_v} \mathcal{L}(\mathbf{w}(t))\| \cos(\delta)) \tag{32}$$

In this case, using Equation (32), we can see $\frac{d\frac{\|\mathbf{w}_u(t)\|}{\|\mathbf{w}_v(t)\|}}{dt} < 0$ at $t_2$. Thus, $\frac{\|\mathbf{w}_u(t)\|}{\|\mathbf{w}_v(t)\|}$ always remains smaller than $\frac{\cos(\delta)}{c+\epsilon}$ and keeps on decreasing. Now, lets say $\lim_{t \to \infty} \frac{\|\mathbf{w}_u(t)\|}{\|\mathbf{w}_v(t)\|} > 0$. This means that

$\frac{\|\mathbf{w}_u(t)\|}{\|\mathbf{w}_v(t)\|} > \Delta$, for some $\Delta > 0$. Also, let's denote $\frac{\|\mathbf{w}_u(t_2)\|}{\|\mathbf{w}_v(t_2)\|}$ by $\beta$. Then we can say

$$\frac{d\frac{\|\mathbf{w}_u(t)\|}{\|\mathbf{w}_v(t)\|}}{dt} \leq -\Delta(\cos(\delta) - \beta(c + \epsilon))\eta\|\mathbf{w}_v(t)\|\|\nabla_{\mathbf{w}_v}\mathcal{L}(\mathbf{w}(t))\|$$

As $\alpha_v \to \infty$, therefore using Equation (7), we can say $\int_{t_2}^{\infty} \eta\|\mathbf{w}_v(t)\|\|\nabla_{\mathbf{w}_v}\mathcal{L}(\mathbf{w}(t))\|dt \to \infty$. Thus, integrating both the sides of the equation above from $t_2$ to $\infty$, we get

$$\int_{t_2}^{\infty} \frac{d\frac{\|\mathbf{w}_u(t)\|}{\|\mathbf{w}_v(t)\|}}{dt}dt \leq -\infty$$

This is not possible as $\frac{\|\mathbf{w}_u(t)\|}{\|\mathbf{w}_v(t)\|}$ is lower bounded by 0. Thus $\lim_{t\to\infty} \frac{\|\mathbf{w}_u(t)\|}{\|\mathbf{w}_v(t)\|} = 0$. □

### E.2.2 GRADIENT DESCENT

**Proposition.** *Let assumptions (A1)-(A5) be satisfied. Consider two nodes $u$ and $v$ in the network such that $\|\widetilde{\mathbf{g}}_v\| \geq \|\widetilde{\mathbf{g}}_u\| > 0$ and $\|\mathbf{w}_u(t)\|, \|\mathbf{w}_v(t)\| \to \infty$. Let $\frac{\|\widetilde{\mathbf{g}}_u\|}{\|\widetilde{\mathbf{g}}_v\|}$ be denoted by $c$. Let $\epsilon, \delta$ be such that $0 < \epsilon < c$ and $0 < \delta < 2\pi$. Then, the following holds:*

1. *There exists a time $t_1$, such that for all $t > t_1$, EWN trajectory has the following properties:*

    (a) $\frac{\|\nabla_{\mathbf{w}_u}\mathcal{L}(\mathbf{w}(t))\|}{\|\nabla_{\mathbf{w}_v}\mathcal{L}(\mathbf{w}(t))\|} \in [c - \epsilon, c + \epsilon]$

    (b) $\left(\frac{\mathbf{w}_u(t)}{\|\mathbf{w}_u(t)\|}\right)^{\top} \left(\frac{-\nabla_{\mathbf{w}_u}\mathcal{L}(\mathbf{w}(t))}{\|\nabla_{\mathbf{w}_u}\mathcal{L}(\mathbf{w}(t))\|}\right) \geq \cos(\delta)$

    (c) $\left(\frac{\mathbf{w}_v(t)}{\|\mathbf{w}_v(t)\|}\right)^{\top} \left(\frac{-\nabla_{\mathbf{w}_v}\mathcal{L}(\mathbf{w}(t))}{\|\nabla_{\mathbf{w}_v}\mathcal{L}(\mathbf{w}(t))\|}\right) \geq \cos(\delta).$

2. *If at some time $t_2 > t_1$,*

    (a) $\frac{\|\mathbf{w}_u(t_2)\|}{\|\mathbf{w}_v(t_2)\|} > \frac{1}{(c-\epsilon)\cos(\delta)} \implies \lim_{t\to\infty} \frac{\|\mathbf{w}_u(t)\|}{\|\mathbf{w}_v(t)\|} = \infty$

    (b) $\frac{\|\mathbf{w}_u(t_2)\|}{\|\mathbf{w}_v(t_2)\|} < \frac{\cos(\delta)}{c+\epsilon} \implies \lim_{t\to\infty} \frac{\|\mathbf{w}_u(t)\|}{\|\mathbf{w}_v(t)\|} = 0$

*Proof.* The proof of part 1a, i.e, $\frac{\|\nabla_{\mathbf{w}_u}\mathcal{L}(\mathbf{w}(t))\|}{\|\nabla_{\mathbf{w}_v}\mathcal{L}(\mathbf{w}(t))\|} \in [c - \epsilon, c + \epsilon]$ follows from the definition of limit as $\frac{\|\nabla_{\mathbf{w}_u}\mathcal{L}(\mathbf{w}(t))\|}{\|\nabla_{\mathbf{w}_v}\mathcal{L}(\mathbf{w}(t))\|}$ tends to $c$.

Now, we will move to the proof of part 1b, i.e, $\left(\frac{\mathbf{w}_u(t)}{\|\mathbf{w}_u(t)\|}\right)^{\top} \left(\frac{-\nabla_{\mathbf{w}_u}\mathcal{L}(\mathbf{w}(t))}{\|\nabla_{\mathbf{w}_u}\mathcal{L}(\mathbf{w}(t))\|}\right) \geq \cos(\delta)$. The assumptions in this Proposition differ slightly from Proposition 2 and thus the proof is slightly more involved as we also need to show that $\mathbf{w}_u(t)$ converges in direction.

As $\|\widetilde{\mathbf{g}}_u\| > 0$, therefore $\nabla_{\mathbf{w}_u}\mathcal{L}(\mathbf{w}(t))$ converges in direction. Therefore, for every $\tau$ satisfying $0 < \tau < 2\pi$, there exists a time $t_3$, such that for $t > t_3$, $\left(\frac{-\nabla_{\mathbf{w}_u}\mathcal{L}(\mathbf{w}(t))}{\|\nabla_{\mathbf{w}_u}\mathcal{L}(\mathbf{w}(t))\|}\right)^{\top} \left(\frac{\widetilde{\mathbf{g}}_u}{\|\widetilde{\mathbf{g}}_u\|}\right) \geq \cos(\tau)$. Now, Let's assume that $\mathbf{w}_u(t)$ does not converge in the direction of $\widetilde{\mathbf{g}}_u$. Then, there must exist a $\tau$ satisfying $0 < \tau < 2\pi$, such that for this $\tau$, there exists a time $t_4 > t_3$ satisfying $\mathbf{v}_u(t_4)^{\top}\left(\frac{\widetilde{\mathbf{g}}_u}{\|\widetilde{\mathbf{g}}_u\|}\right) = \cos(\Delta)$, where $\Delta > \tau$.

Now, we are going to show that for any $\kappa$ satisfying $\tau < \kappa < \Delta$, there exists a time $t_5 > t_4$ such that $\left(\frac{\mathbf{v}_u(t_5)}{\|\mathbf{v}_u(t_5)\|}\right)^{\top}\left(\frac{\widetilde{\mathbf{g}}_u}{\|\widetilde{\mathbf{g}}_u\|}\right) > \cos(\kappa)$. Let's say for a given $\kappa$, no such $t_5$ exists. Then, taking dot product with $\frac{\widetilde{\mathbf{g}}_u}{\|\widetilde{\mathbf{g}}_u\|}$ on both sides of Equation (12), we can say

$$\frac{\mathbf{v}_u(t+1)^{\top}\widetilde{\mathbf{g}}_u}{\|\widetilde{\mathbf{g}}_u\|} = \frac{\mathbf{v}_u(t)^{\top}\widetilde{\mathbf{g}}_u}{\|\widetilde{\mathbf{g}}_u\|} +$$
$$\eta(t)\frac{e^{\alpha_u(t)}}{\|\mathbf{v}_u(t)\|}\|\nabla_{\mathbf{w}_u}\mathcal{L}(\mathbf{w}(t))\|\left(\frac{\widetilde{\mathbf{g}}_u}{\|\widetilde{\mathbf{g}}_u\|}\right)^{\top}(I - \frac{\mathbf{v}_u(t)\mathbf{v}_u(t)^{\top}}{\|\mathbf{v}_u(t)\|^2})\left(\frac{-\nabla_{\mathbf{w}_u}\mathcal{L}(\mathbf{w}(t))}{\|\nabla_{\mathbf{w}_u}\mathcal{L}(\mathbf{w}(t))\|}\right)$$

Now, as $\left(\frac{\widetilde{\mathbf{g}}_u}{\|\widetilde{\mathbf{g}}_u\|}\right)^\top \left(\frac{-\nabla_{\mathbf{w}_u}\mathcal{L}(\mathbf{w}(t))}{\|\nabla_{\mathbf{w}_u}\mathcal{L}(\mathbf{w}(t))\|}\right) \geq \cos(\tau)$ and $\left(\frac{\widetilde{\mathbf{g}}_u}{\|\widetilde{\mathbf{g}}_u\|}\right)^\top \left(\frac{\mathbf{v}_u(t)}{\|\mathbf{v}_u(t)\|}\right) \leq \cos(\kappa)$, we can say

$$\frac{\mathbf{v}_u(t+1)^\top \widetilde{\mathbf{g}}_u}{\|\widetilde{\mathbf{g}}_u\|} \geq \frac{\mathbf{v}_u(t)^\top \widetilde{\mathbf{g}}_u}{\|\widetilde{\mathbf{g}}_u\|} + (\cos(\tau) - \cos(\kappa))(\eta(t)\frac{e^{\alpha_u(t)}}{\|\mathbf{v}_u(t)\|}\|\nabla_{\mathbf{w}_u}\mathcal{L}(\mathbf{w}(t))\|) \qquad (33)$$

Now, using arguments similar to the proof of Proposition 2 for EWN gradient descent in Appendix C.1.2, we can show that the above statement leads to a contradiction and thus there must exist a $t_5$ such that $\left(\frac{\mathbf{v}_u(t_5)}{\|\mathbf{v}_u(t_5)\|}\right)^\top \left(\frac{\widetilde{\mathbf{g}}_u}{\|\widetilde{\mathbf{g}}_u\|}\right) > \cos(\kappa)$.

Now, we are going to show that there exists a $t_6 > t_5$, such that for all $t > t_6$, $\left(\frac{\mathbf{v}_u(t)}{\|\mathbf{v}_u(t)\|}\right)^\top \left(\frac{\widetilde{\mathbf{g}}_u}{\|\widetilde{\mathbf{g}}_u\|}\right) > \cos(\kappa)$. Consider a $\beta$ such that $\tau < \beta < \kappa$. Now, if at any time t, $\left(\frac{\mathbf{v}_u(t)}{\|\mathbf{v}_u(t)\|}\right)^\top \left(\frac{\widetilde{\mathbf{g}}_u}{\|\widetilde{\mathbf{g}}_u\|}\right) < \cos(\beta)$, then, similar to Equation (33), we can say

$$\frac{\mathbf{v}_u(t+1)^\top \widetilde{\mathbf{g}}_u}{\|\widetilde{\mathbf{g}}_u\|} \geq \frac{\mathbf{v}_u(t)^\top \widetilde{\mathbf{g}}_u}{\|\widetilde{\mathbf{g}}_u\|} + (\cos(\tau) - \cos(\beta))(\eta(t)\frac{e^{\alpha_u(t)}}{\|\mathbf{v}_u(t)\|}\|\nabla_{\mathbf{w}_u}\mathcal{L}(\mathbf{w}(t))\|)$$

Using the upper bound on $\|\mathbf{v}_u(t+1)\|$ from Equation (13), we can say

$$\frac{\mathbf{v}_u(t+1)^\top \widetilde{\mathbf{g}}_u}{\|\mathbf{v}_u(t+1)\|\|\widetilde{\mathbf{g}}_u\|} \geq \frac{\frac{\mathbf{v}_u(t)^\top \widetilde{\mathbf{g}}_u}{\|\widetilde{\mathbf{g}}_u\|} + (\cos(\tau) - \cos(\beta))(\eta(t)\frac{e^{\alpha_u(t)}}{\|\mathbf{v}_u(t)\|}\|\nabla_{\mathbf{w}_u}\mathcal{L}(\mathbf{w}(t))\|)}{\sqrt{\|\mathbf{v}_u(t)\|^2 + (\eta(t)\frac{e^{\alpha_u(t)}}{\|\mathbf{v}_u(t)\|}\|\nabla_{\mathbf{w}_u}\mathcal{L}(\mathbf{w}(t))\|)^2}} \qquad (34)$$

Let $\eta(t)\frac{e^{\alpha_u(t)}}{\|\mathbf{v}_u(t)\|}\|\nabla_{\mathbf{w}_u}\mathcal{L}(\mathbf{w}(t))\|$ be denoted by $\chi(t)$. Then, the above equation can be rewritten as

$$\frac{\mathbf{v}_u(t+1)^\top \widetilde{\mathbf{g}}_u}{\|\mathbf{v}_u(t+1)\|\|\widetilde{\mathbf{g}}_u\|} \geq \frac{\mathbf{v}_u(t)^\top \widetilde{\mathbf{g}}_u}{\|\mathbf{v}_u(t)\|\|\widetilde{\mathbf{g}}_u\|}\frac{\|\mathbf{v}_u(t)\|}{\sqrt{\|\mathbf{v}_u(t)\|^2 + \chi(t)^2}} + (\cos(\tau) - \cos(\beta))\frac{\chi(t)}{\sqrt{\|\mathbf{v}_u(t)\|^2 + \chi(t)^2}}$$

Now, we are going to show that for a small enough $\chi(t)$, RHS is greater than $\frac{\mathbf{v}_u(t)^\top \widetilde{\mathbf{g}}_u}{\|\mathbf{v}_u(t)\|\|\widetilde{\mathbf{g}}_u\|}$.

$$\frac{\mathbf{v}_u(t)^\top \widetilde{\mathbf{g}}_u}{\|\mathbf{v}_u(t)\|\|\widetilde{\mathbf{g}}_u\|}\frac{\|\mathbf{v}_u(t)\|}{\sqrt{\|\mathbf{v}_u(t)\|^2 + \chi(t)^2}} + (\cos(\tau) - \cos(\beta))\frac{\chi(t)}{\sqrt{\|\mathbf{v}_u(t)\|^2 + \chi(t)^2}} > \frac{\mathbf{v}_u(t)^\top \widetilde{\mathbf{g}}_u}{\|\mathbf{v}_u(t)\|\|\widetilde{\mathbf{g}}_u\|}$$

$$\implies (\cos(\tau) - \cos(\beta))\frac{\chi(t)}{\sqrt{\|\mathbf{v}_u(t)\|^2 + \chi(t)^2}} > \frac{\mathbf{v}_u(t)^\top \widetilde{\mathbf{g}}_u}{\|\mathbf{v}_u(t)\|\|\widetilde{\mathbf{g}}_u\|}(1 - \frac{\|\mathbf{v}_u(t)\|}{\sqrt{\|\mathbf{v}_u(t)\|^2 + \chi(t)^2}})$$

$$\implies (\cos(\tau) - \cos(\beta)) > \frac{\mathbf{v}_u(t)^\top \widetilde{\mathbf{g}}_u}{\|\mathbf{v}_u(t)\|\|\widetilde{\mathbf{g}}_u\|}(\frac{\sqrt{\|\mathbf{v}_u(t)\|^2 + \chi(t)^2} - \|\mathbf{v}_u(t)\|}{\chi(t)})$$

Clearly as $\chi(t) \to 0$, the RHS tends to 0, therefore the equation is satisfied. Thus for a small enough $\chi(t)$, RHS of Equation (34) is greater than $\frac{\mathbf{v}_u(t)^\top \widetilde{\mathbf{g}}_u}{\|\mathbf{v}_u(t)\|\|\widetilde{\mathbf{g}}_u\|}$. As $\|\mathbf{v}_u(t)\|$ keeps on increasing and by Assumption (A5), $\lim_{t\to\infty} \eta(t)\gamma_u(t)\|\nabla_{\mathbf{w}_u}\mathcal{L}(\mathbf{w}(t))\| = 0$, we can say there exists a time $t_7$, such that for any $t > t_7$, $\frac{\mathbf{v}_u(t)^\top \widetilde{\mathbf{g}}_u}{\|\mathbf{v}_u(t)\|\|\widetilde{\mathbf{g}}_u\|}$ goes up whenever $\left(\frac{\mathbf{v}_u(t)}{\|\mathbf{v}_u(t)\|}\right)^\top \left(\frac{\widetilde{\mathbf{g}}_u}{\|\widetilde{\mathbf{g}}_u\|}\right) < \cos(\beta)$.

Also, by using Equation (12) and Assumption (A5), we can say, that there exists a time $t_8$, such that for $t > t_8$, $\left(\frac{\mathbf{v}_u(t)}{\|\mathbf{v}_u(t)\|}\right)^\top \left(\frac{\widetilde{\mathbf{g}}_u}{\|\widetilde{\mathbf{g}}_u\|}\right) > \cos(\beta) \implies \left(\frac{\mathbf{v}_u(t+1)}{\|\mathbf{v}_u(t+1)\|}\right)^\top \left(\frac{\widetilde{\mathbf{g}}_u}{\|\widetilde{\mathbf{g}}_u\|}\right) > \cos(\kappa)$, as the RHS of Equation (12) goes to 0 norm in limit. Now, define $t_6 > max(t_7, t_8)$ such that $\left(\frac{\mathbf{v}_u(t_6)}{\|\mathbf{v}_u(t_6)\|}\right)^\top \left(\frac{\widetilde{\mathbf{g}}_u}{\|\widetilde{\mathbf{g}}_u\|}\right) > \cos(\kappa)$ (must exist from previous arguments). Then, as the dot product always goes up when between $\cos(\beta)$ and $\cos(\kappa)$, and can't go in a single step from being greater than $\cos(\beta)$ to less than $\cos(\kappa)$, therefore, for every $t > t_6$, $\left(\frac{\mathbf{v}_u(t)}{\|\mathbf{v}_u(t)\|}\right)^\top \left(\frac{\widetilde{\mathbf{g}}_u}{\|\widetilde{\mathbf{g}}_u\|}\right) > \cos(\kappa)$.

Now as the above argument holds for any $\kappa$ between $\tau$ and $\Delta$, and for any $\tau > 0$, we can say that $\mathbf{w}_u(t)$ converges in direction of $\widetilde{\mathbf{g}}_u$.

The proof of part 1c, i.e, $\left(\frac{\mathbf{w}_v(t)}{\|\mathbf{w}_v(t)\|}\right)^\top \left(\frac{-\nabla_{\mathbf{w}_v}\mathcal{L}(\mathbf{w}(t))}{\|\nabla_{\mathbf{w}_v}\mathcal{L}(\mathbf{w}(t))\|}\right) \geq \cos(\delta)$ follows exactly the same steps as part 1b.

Now, we will move to the proof of part 2a, i.e, $\frac{\|\mathbf{w}_u(t_2)\|}{\|\mathbf{w}_v(t_2)\|} > \frac{1}{(c-\epsilon)\cos(\delta)} \implies \lim_{t\to\infty} \frac{\|\mathbf{w}_u(t)\|}{\|\mathbf{w}_v(t)\|} = \infty$

Using Equation (11) and part 1 of the Proposition, we can say

$$
\begin{aligned}
\frac{\|\mathbf{w}_u(t_2+1)\|}{\|\mathbf{w}_v(t_2+1)\|} &\geq \frac{\|\mathbf{w}_u(t_2)\| + \eta(t_2)\cos(\delta)\|\mathbf{w}_u(t_2)\|^2\|\nabla_{\mathbf{w}_u}\mathcal{L}(\mathbf{w}(t_2))\|}{\|\mathbf{w}_v(t_2)\| + \eta(t_2)\|\mathbf{w}_v(t_2)\|^2\|\nabla_{\mathbf{w}_v}\mathcal{L}(\mathbf{w}(t_2))\|} \\
&= \frac{\|\mathbf{w}_u(t_2)\|}{\|\mathbf{w}_v(t_2)\|}\left(\frac{1+\cos(\delta)\eta(t_2)\|\mathbf{w}_u(t_2)\|\|\nabla_{\mathbf{w}_u}\mathcal{L}(\mathbf{w}(t_2))\|}{1+\eta(t_2)\|\mathbf{w}_v(t_2)\|\|\nabla_{\mathbf{w}_v}\mathcal{L}(\mathbf{w}(t_2))\|}\right) \\
&\geq \frac{\|\mathbf{w}_u(t_2)\|}{\|\mathbf{w}_v(t_2)\|}
\end{aligned}
$$

Thus, $\frac{\|\mathbf{w}_u(t)\|}{\|\mathbf{w}_v(t)\|}$ keeps on increasing for $t > t_2$. It can either diverge to infinity or converge to a finite value. If it converges to a finite value, then by Stolz Cesaro theorem,

$$
\lim_{t\to\infty} \frac{\|\mathbf{w}_u(t)\|}{\|\mathbf{w}_v(t)\|} = \lim_{t\to\infty} \frac{\|\mathbf{w}_u(t)\|^2\|\nabla_{\mathbf{w}_u}\mathcal{L}(\mathbf{w}(t))\|}{\|\mathbf{w}_v(t)\|^2\|\nabla_{\mathbf{w}_v}\mathcal{L}(\mathbf{w}(t))\|}
$$

However, this is not possible as $\frac{\|\mathbf{w}_u(t)\|}{\|\mathbf{w}_v(t)\|} > \frac{1}{c}$ for every $t > t_2$. Thus, $\frac{\|\mathbf{w}_u(t)\|}{\|\mathbf{w}_v(t)\|}$ diverges to infinity.

Now, we will move to the proof of part 2b, i.e, $\frac{\|\mathbf{w}_u(t_2)\|}{\|\mathbf{w}_v(t_2)\|} < \frac{\cos(\delta)}{c+\epsilon} \implies \lim_{t\to\infty} \frac{\|\mathbf{w}_u(t)\|}{\|\mathbf{w}_v(t)\|} = 0$

Using Equation (11) and part 1 of the Proposition, we can say

$$
\begin{aligned}
\frac{\|\mathbf{w}_u(t_2+1)\|}{\|\mathbf{w}_v(t_2+1)\|} &\leq \frac{\|\mathbf{w}_u(t_2)\| + \eta(t_2)\|\mathbf{w}_u(t_2)\|^2\|\nabla_{\mathbf{w}_u}\mathcal{L}(\mathbf{w}(t_2))\|}{\|\mathbf{w}_v(t_2)\| + \eta(t_2)\cos(\delta)\|\mathbf{w}_v(t_2)\|^2\|\nabla_{\mathbf{w}_v}\mathcal{L}(\mathbf{w}(t_2))\|} \\
&= \frac{\|\mathbf{w}_u(t_2)\|}{\|\mathbf{w}_v(t_2)\|}\left(\frac{1+\eta(t_2)\|\mathbf{w}_u(t_2)\|\|\nabla_{\mathbf{w}_u}\mathcal{L}(\mathbf{w}(t_2))\|}{1+\eta(t_2)\cos(\delta)\|\mathbf{w}_v(t_2)\|\|\nabla_{\mathbf{w}_v}\mathcal{L}(\mathbf{w}(t_2))\|}\right) \\
&\leq \frac{\|\mathbf{w}_u(t_2)\|}{\|\mathbf{w}_v(t_2)\|}
\end{aligned}
$$

Thus, $\frac{\|\mathbf{w}_u(t)\|}{\|\mathbf{w}_v(t)\|}$ keeps on decreasing for $t > t_2$. As it is always greater than zero, it must converge. Therefore, by Stolz Cesaro Theorem,

$$
\lim_{t\to\infty} \frac{\|\mathbf{w}_u(t)\|}{\|\mathbf{w}_v(t)\|} = \lim_{t\to\infty} \frac{\|\mathbf{w}_u(t)\|^2\|\nabla_{\mathbf{w}_u}\mathcal{L}(\mathbf{w}(t))\|}{\|\mathbf{w}_v(t)\|^2\|\nabla_{\mathbf{w}_v}\mathcal{L}(\mathbf{w}(t))\|}
$$

For $\frac{\|\mathbf{w}_u(t)\|}{\|\mathbf{w}_v(t)\|} < \frac{1}{c}$, this can only be satisfied when $\lim_{t\to\infty}\frac{\|\mathbf{w}_u(t)\|}{\|\mathbf{w}_v(t)\|} = 0$. $\qquad\square$

# F    PROOF OF COROLLARY 1

**Corollary.** *Consider a weight normalized(SWN or EWN) multilayer linear net, represented by $y = W_n W_{n-1}...W_1 x$. Under assumptions (A1)-(A4) for gradient flow and (A1)-(A5) for gradient descent, if the dataset is sampled from a continuous distribution w.r.t $\mathbb{R}^d$, then, with probability 1, $\boldsymbol{\theta} = W_1^\top W_2^\top ....W_n^\top$ converges in direction to the maximum margin separator for all linearly separable datasets.*

*Proof.* Consider a linear net given by $f = W_n W_{n-1}...W_1 x$, where $f$ is a scalar as we are considering a binary classification problem with exponential loss. Then

$$
\frac{\partial \mathcal{L}(\mathbf{w}(t))}{\partial W_1(t)} = -\sum_{i=0}^m \ell_i(t)y_i(W_n(t)W_{n-1}(t)...W_2(t))^\top \mathbf{x}_i^\top \tag{35}
$$

Now, atleast one of the neurons in every layer would have a non-zero component on $\widetilde{\mathbf{w}}$, otherwise $\Phi(\widetilde{\mathbf{w}}, \mathbf{x}_i) = 0$ for all $i$, that implies $\rho = 0$.

Let $u$ be one of the nodes of $W_1$ that has a non-zero component in $\widetilde{\mathbf{w}}_u$ and let $\mathbf{w}_u$ be the $k^{th}$ row of the matrix $W_1$. Now, from Proposition 2, we know $\widetilde{\mathbf{w}}_u = \lambda \widetilde{\mathbf{g}}_u$. Denoting the component of $\widetilde{\mathbf{w}}$ along matrix $W_k$ by $\widetilde{W}_k$ and using Equation (35), we get, for some $\lambda_k > 0$,

$$\widetilde{\mathbf{w}}_u = \lambda_k((\widetilde{W}_n \widetilde{W}_{n-1}.....\widetilde{W}_2)^\top)[k] \sum_{i=1}^{m} \widetilde{\ell}_i y_i \mathbf{x}_i^\top$$

where $((\widetilde{W}_n \widetilde{W}_{n-1}.....\widetilde{W}_2)^\top)[k]$ represents the $k^{th}$ component of the product column vector. Let $S$ represent the set of rows in $W_1$ that have a non-zero component in $\widetilde{\mathbf{w}}$. Also, let $\widetilde{\theta}$ denote the final convergent direction of $\theta$. Then, for some $\mu > 0$, we can say

$$\widetilde{\theta} = \mu \widetilde{W}_1^T \widetilde{W}_2^T ....\widetilde{W}_n^T = \mu \sum_{j \in S} \lambda_j (((\widetilde{W}_n \widetilde{W}_{n-1}.....\widetilde{W}_2)^\top)[j])^2 \sum_{i=1}^{m} \widetilde{\ell}_i y_i \mathbf{x}_i$$

Thus, $\theta$ satisfies the KKT conditions of the maximum margin problem and if data is sampled from a continuous distribution w.r.t $\mathbb{R}^d$, then with probability 1(Soudry et al., 2018), would converge to the maximum margin separator. □

## G  PROOF OF THEOREM 3

**Theorem.** *For EWN, under Assumptions (A1)-(A5) and $\lim_{t \to \infty} \frac{\|\mathbf{r}(t+1) - \mathbf{r}(t)\|}{g(t+1) - g(t)} = 0$, the following hold*

1. *$\|\mathbf{w}(t)\|$ asymptotically grows at $\Theta\left((\log(d(t))^{\frac{1}{L}}\right)$*

2. *$\mathcal{L}(\mathbf{w}(t))$ asymptotically goes down at the rate of $\Theta\left(\frac{1}{d(t)(\log d(t))^2}\right)$.*

First, we will establish rates for gradient flow and then go to the case of gradient descent.

### G.1  GRADIENT FLOW

Although the asymptotic convergence rates for smooth homogeneous neural nets have been established in Lyu & Li (2020), the proof technique becomes easier to understand for smooth homogeneous nets, without weight normalization.

#### G.1.1  UNNORMALIZED NETWORK

**Theorem.** *For Unnorm, under Assumptions (A1)-(A4) for gradient flow and $\lim_{t \to \infty} \frac{\|\frac{d\mathbf{r}(t)}{dt}\|}{g'(t)} = 0$, the following hold*

1. *$\|\mathbf{w}(t)\|$ asymptotically grows at $\Theta\left((\log(t)^{\frac{1}{L}}\right)$*

2. *$\mathcal{L}(\mathbf{w}(t))$ asymptotically goes down at the rate of $\Theta\left(\frac{1}{t(\log t)^{2-\frac{2}{L}}}\right)$.*

*Proof.* Consider $\mathbf{w} = g(t)\widetilde{\mathbf{w}} + \mathbf{r}(t)$, where $\lim_{t \to \infty} \frac{\|\mathbf{r}(t)\|}{g(t)} = 0$ and $\mathbf{r}(t)^\top \widetilde{\mathbf{w}} = 0$. Now, we make an additional assumption that $\lim_{t \to \infty} \frac{\|\frac{d\mathbf{r}(t)}{d(t)}\|}{g'(t)} = 0$. This basically avoids any oscillations in $\mathbf{r}(t)$ for large $t$, where it can have a higher derivative, but the value may be bounded. Now, we know

$$\frac{d\mathbf{w}(t)}{dt} = \sum_{i=1}^{m} e^{-y_i \Phi(\mathbf{w}(t), \mathbf{x}_i)} y_i \nabla_{\mathbf{w}} \Phi(\mathbf{w}(t), \mathbf{x}_i)$$

Now, we know $\|\frac{d\mathbf{w}(t)}{dt}\| \neq 0$ for any finite $t$, otherwise $\mathbf{w}$ won't change and $\mathcal{L}$ can't converge to 0. Thus, for all $t$, we can say

$$\frac{\|\frac{d\mathbf{w}(t)}{dt}\|}{\|\sum_{i=1}^m e^{-y_i\Phi(\mathbf{w}(t),\mathbf{x}_i)}y_i\nabla_\mathbf{w}\Phi(\mathbf{w},\mathbf{x}_i)\|} = 1$$

Taking limit $t \to \infty$ on both the sides, we get

$$\lim_{t\to\infty}\frac{\|\frac{d\mathbf{w}(t)}{dt}\|}{\|\sum_{i=1}^m e^{-y_i\Phi(\mathbf{w}(t),\mathbf{x}_i)}y_i\nabla_\mathbf{w}\Phi(\mathbf{w},\mathbf{x}_i)\|} = 1 \tag{36}$$

Now, we know

$$\|\frac{d\mathbf{w}(t)}{dt}\| = \|g'(t)\widetilde{\mathbf{w}} + \frac{d\mathbf{r}(t)}{dt}\|$$

$$\|\sum_{i=1}^m e^{-y_i\Phi(\mathbf{w}(t),\mathbf{x}_i)}y_i\nabla_\mathbf{w}\Phi(\mathbf{w},\mathbf{x}_i)\| = \|\sum_{i=1}^m e^{-y_i g(t)^L\Phi(\widetilde{\mathbf{w}}+\frac{\mathbf{r}(t)}{g(t)},\mathbf{x}_i)}(y_i g(t)^{L-1}\nabla_\mathbf{w}\Phi(\widetilde{\mathbf{w}}+\frac{\mathbf{r}(t)}{g(t)},\mathbf{x}_i)\|$$

Let $S = \{i : \Phi(\widetilde{\mathbf{w}},\mathbf{x}_i) = \min_j \Phi(\widetilde{\mathbf{w}},\mathbf{x}_j)\}$. Then as $\rho > 0$, we can say

$$\lim_{t\to\infty}\frac{\|\sum_{i=1}^m e^{-y_i g(t)^L\Phi(\widetilde{\mathbf{w}}+\frac{\mathbf{r}(t)}{g(t)},\mathbf{x}_i)}(y_i g(t)^{L-1}\nabla_\mathbf{w}\Phi(\widetilde{\mathbf{w}}+\frac{\mathbf{r}(t)}{g(t)},\mathbf{x}_i)\|}{e^{-\rho g(t)^L}g(t)^{L-1}\|\sum_{i\in S}\widetilde{\ell}_i y_i\nabla_\mathbf{w}\Phi(\widetilde{\mathbf{w}},\mathbf{x}_i)\|} = k$$

where $k$ is some constant. Also, by the assumption

$$\lim_{t\to\infty}\frac{\|\frac{d\mathbf{w}(t)}{dt}\|}{g'(t)} = 1$$

Substituting the above two equations in Equation (36), we get

$$\lim_{t\to\infty}\frac{g'(t)}{e^{-\rho g(t)^L}g(t)^{L-1}\|\sum_{i\in S}\widetilde{\ell}_i y_i\nabla_\mathbf{w}\Phi(\widetilde{\mathbf{w}},\mathbf{x}_i)\|} = \frac{1}{k}$$

Now, as loss goes down at the rate of $e^{-\rho g(t)^L}$, multiplying the numerator and denominator by $\rho L g(t)^{L-1}$ and denoting $h(t) = \rho g(t)^L$, we get

$$\lim_{t\to\infty}\frac{\rho^{1-\frac{2}{L}}h'(t)}{Le^{-h(t)}h(t)^{2-\frac{2}{L}}\|\sum_{i\in S}\widetilde{\ell}_i y_i\nabla_\mathbf{w}\Phi(\widetilde{\mathbf{w}},\mathbf{x}_i)\|} = \frac{1}{k}$$

Thus, asymptotically, $h(t)$ grows at $\Theta(\log(t) + (2 - \frac{2}{L})\log\log t)$ and thus loss goes down at $\Theta(\frac{1}{t(\log t)^{2-\frac{2}{L}}})$. $\qquad\square$

### G.1.2 Exponential Weight Normalization

**Theorem.** *For EWN, under Assumptions (A1)-(A4) for gradient flow and $\lim_{t\to\infty}\frac{\|\frac{d\mathbf{r}(t)}{dt}\|}{g'(t)} = 0$, the following hold*

1. *$\|\mathbf{w}(t)\|$ asymptotically grows at $\Theta\left((\log(t)^{\frac{1}{L}}\right)$*

2. *$\mathcal{L}(\mathbf{w}(t))$ asymptotically goes down at the rate of $\Theta\left(\frac{1}{t(\log t)^2}\right)$.*

*Proof.* Consider $\mathbf{w} = g(t)\widetilde{\mathbf{w}} + \mathbf{r}(t)$, where $\lim_{t\to\infty}\frac{\|\mathbf{r}(t)\|}{g(t)} = 0$ and $\mathbf{r}(t)^\top\widetilde{\mathbf{w}} = 0$. Now, we make an additional assumption that $\lim_{t\to\infty}\frac{\|\frac{d\mathbf{r}(t)}{d(t)}\|}{g'(t)} = 0$.

In this case,

$$-\frac{d\mathcal{L}(\mathbf{w}(t))}{d\mathbf{w}} = \sum_{i=1}^m e^{-y_i\Phi(\mathbf{w}(t),\mathbf{x}_i)}y_i\nabla_\mathbf{w}\Phi(\mathbf{w}(t),\mathbf{x}_i)$$

However, in this case, for a node $u$,

$$\frac{d\mathbf{w}_u(t)}{dt} = -\|\mathbf{w}_u(t)\|^2 \frac{d\mathcal{L}(\mathbf{w}(t))}{d\mathbf{w}_u}$$

Consider a vector $\mathbf{a}(t)$ of equal dimension as $\mathbf{w}$, and its components corresponding to a node $u$ is given by $\mathbf{a}_u(t) = -\|\widetilde{\mathbf{w}}_u\|^2 \frac{d\mathcal{L}(\mathbf{w}(t))}{d\mathbf{w}_u}$. Now as we know $\mathbf{w}$ converges in direction to $\widetilde{\mathbf{w}}$, therefore, using the update equation above, we can say

$$\lim_{t\to\infty} \frac{\|\frac{d\mathbf{w}(t)}{dt}\|}{g(t)^2\|\mathbf{a}(t)\|} = 1$$

Using the update equation for $-\frac{d\mathcal{L}(\mathbf{w}(t))}{d\mathbf{w}}$, we can say

$$\lim_{t\to\infty} \frac{\|\frac{d\mathcal{L}(\mathbf{w}(t))}{dw}\|}{e^{-\rho g(t)^L} g(t)^{L-1}} = k_1 \tag{37}$$

where $k_1$ is some constant. Now, using the expression for $\mathbf{a}(t)$, we can say

$$\lim_{t\to\infty} \frac{\|\mathbf{a}(t)\|}{e^{-\rho g(t)^L} g(t)^{L-1}} = k$$

where $k$ is some constant. Using the equations above, we can say

$$\lim_{t\to\infty} \frac{g'(t)}{e^{-\rho g(t)^L} g(t)^{L+1}} = \frac{1}{k}$$

Now, as loss goes down at the rate of $e^{-\rho g(t)^L}$, multiplying the numerator and denominator by $\rho L g(t)^{L-1}$ and denoting $h(t) = \rho g(t)^L$, we get

$$\lim_{t\to\infty} \frac{\rho^2 h'(t)}{e^{-h(t)} h(t)^2} = \frac{1}{k}$$

Thus, asymptotically, $h(t)$ grows at $\Theta(\log(t) + 2\log\log t)$ and thus loss goes down at the rate of $\Theta(\frac{1}{t(\log t)^2})$. $\qquad\square$

## G.2 GRADIENT DESCENT

**Theorem.** *For Exponential Weight Normalization, under Assumptions (A1)-(A5) and* $\lim_{t\to\infty} \frac{\|\mathbf{r}(t+1)-\mathbf{r}(t)\|}{g(t+1)-g(t)} = 0$, *the following hold*

1. $\|\mathbf{w}(t)\|$ *asymptotically grows at* $\Theta\left((\log(d(t))^{\frac{1}{L}})\right)$

2. $\mathcal{L}(\mathbf{w}(t))$ *asymptotically goes down at the rate of* $\Theta\left(\frac{1}{d(t)(\log d(t))^2}\right)$.

*Proof.* Consider $\mathbf{w} = g(t)\widetilde{\mathbf{w}} + \mathbf{r}(t)$, where $\lim_{t\to\infty} \frac{\|\mathbf{r}(t)\|}{g(t)} = 0$ and $\mathbf{r}(t)^\top \widetilde{\mathbf{w}} = 0$. Now, we make additional assumptions that $\lim_{t\to\infty} \frac{\|\mathbf{r}(t+1)-\mathbf{r}(t)\|}{g(t+1)-g(t)} = 0$.

Consider a node $u$ in the network that has $\|\widetilde{\mathbf{w}}_u\| > 0$. The update equations for $\mathbf{v}_u(t)$ and $\alpha_u(t)$ are given by

$$\alpha_u(t+1) = \alpha_u(t) - \eta(t)e^{\alpha_u(t)} \frac{\mathbf{v}_u(t)^\top \nabla_{\mathbf{w}_u}\mathcal{L}(\mathbf{w}(t))}{\|\mathbf{v}_u(t)\|}$$

$$\mathbf{v}_u(t+1) = \mathbf{v}_u(t) - \eta(t)\frac{e^{\alpha_u(t)}}{\|\mathbf{v}_u(t)\|}(I - \frac{\mathbf{v}_u(t)\mathbf{v}_u(t)^\top}{\|\mathbf{v}_u(t)\|^2})\nabla_{\mathbf{w}_u}\mathcal{L}(\mathbf{w}(t))$$

Now, we will first estimate $\|e^{\alpha_u(t+1)}\frac{\mathbf{v}_u(t+1)}{\|\mathbf{v}_u(t+1)\|} - e^{\alpha_u(t)}\frac{\mathbf{v}_u(t)}{\|\mathbf{v}_u(t)\|}\|$. Let $\delta_u(t)$ denote $\eta(t)e^{\alpha_u(t)}\|\nabla_{\mathbf{w}_u}\mathcal{L}(\mathbf{w}(t))\|$ and $\epsilon_u(t)$ denote the angle between $\mathbf{v}_u(t)$ and $-\nabla_{\mathbf{w}_u}\mathcal{L}(\mathbf{w}(t))$. We know

$\lim_{t\to\infty} \delta_u(t) = 0$ and $\lim_{t\to\infty} \epsilon_u(t) = 0$. Now, rewriting update equations in terms of these symbols, we get

$$e^{\alpha_u(t+1)} = e^{\alpha_u(t)} e^{\delta_u(t)\cos(\epsilon_u(t))}$$

$$\mathbf{v}_u(t+1) = \mathbf{v}_u(t) + \delta_u(t)\sin(\epsilon_u(t))\frac{-\nabla_{\mathbf{w}_u}\mathcal{L}(\mathbf{w}(t))_{\mathbf{v}_u(t)_\perp}}{\|\nabla_{\mathbf{w}_u}\mathcal{L}(\mathbf{w}(t))_{\mathbf{v}_u(t)_\perp}\|}$$

where $-\nabla_{\mathbf{w}_u}\mathcal{L}(\mathbf{w}(t))_{\mathbf{v}_u(t)_\perp}$ denotes the component of $-\nabla_{\mathbf{w}_u}\mathcal{L}(\mathbf{w}(t))$ perpendicular to $\mathbf{v}_u(t)$. Now, using these equations we can say

$$e^{\alpha_u(t+1)}\frac{\mathbf{v}_u(t+1)}{\|\mathbf{v}_u(t+1)\|} - e^{\alpha_u(t)}\frac{\mathbf{v}_u(t)}{\|\mathbf{v}_u(t)\|} = e^{\alpha_u(t)}(e^{\delta_u(t)\cos(\epsilon_u(t))}\frac{\|\mathbf{v}_u(t)\|}{\|\mathbf{v}_u(t+1)\|} - 1)\frac{\mathbf{v}_u(t)}{\|\mathbf{v}_u(t)\|}$$
$$+ \left(\frac{e^{\alpha_u(t+1)}\delta_u(t)\sin(\epsilon_u(t))}{\|\mathbf{v}_u(t)\|\|\mathbf{v}_u(t+1)\|}\right)\frac{-\nabla_{\mathbf{w}_u}\mathcal{L}(\mathbf{w}(t))_{\mathbf{v}_u(t)_\perp}}{\|\nabla_{\mathbf{w}_u}\mathcal{L}(\mathbf{w}(t))_{\mathbf{v}_u(t)_\perp}\|}$$
$$\tag{38}$$

Now as $\lim_{t\to\infty}\frac{\|\mathbf{v}_u(t)\|}{\|\mathbf{v}_u(t+1)\|} = 1$, therefore we can say

$$\lim_{t\to\infty}\frac{e^{\delta_u(t)cos(\epsilon_u(t))\frac{\|\mathbf{v}_u(t)\|}{\|\mathbf{v}_u(t+1)\|}} - 1}{\delta_u(t)cos(\epsilon_u(t))} = 1$$

Now, as $\|\mathbf{v}_u(t)\|$ keeps on increasing during the gradient descent trajectory, therefore we can say $\frac{1}{\|\mathbf{v}_u(t)\|\|\mathbf{v}_u(t+1)\|} \le k$, where $k > 0$ is some constant. Now dividing both sides of Equation (38) by $e^{\alpha_u(t)}\delta_u(t)\cos(\epsilon_u(t))$ and analyzing the coefficient of the second term on RHS, we get

$$\lim_{t\to\infty}\frac{e^{\delta_u(t)\cos(\epsilon_u(t))}\sin(\epsilon_u(t))}{\|\mathbf{v}_u(t)\|\|\mathbf{v}_u(t+1)\|\cos(\epsilon_u(t))} \le 0$$

Taking norm on both sides of Equation (38), using Pythagoras theorem and the limits established above, we can say

$$\lim_{t\to\infty}\frac{\|e^{\alpha_u(t+1)}\frac{\mathbf{v}_u(t+1)}{\|\mathbf{v}_u(t+1)\|} - e^{\alpha_u(t)}\frac{\mathbf{v}_u(t)}{\|\mathbf{v}_u(t)\|}\|}{e^{\alpha_u(t)}\delta_u(t)} = 1$$

Now, we also know

$$\lim_{t\to\infty}\frac{\|e^{\alpha_u(t+1)}\frac{\mathbf{v}_u(t+1)}{\|\mathbf{v}_u(t+1)\|} - e^{\alpha_u(t)}\frac{\mathbf{v}_u(t)}{\|\mathbf{v}_u(t)\|}\|}{g(t+1) - g(t)} = \|\widetilde{\mathbf{w}}_u\|$$

Now, using equations above and Equation (37), we can say

$$\lim_{t\to\infty}\frac{g(t+1) - g(t)}{\eta(t)e^{-\rho g(t)^L}g(t)^{L+1}} = c$$

where $c$ is some constant. This determines the asymptotic rate of $g(t)$. To get a better closed form, define define a map $d : \mathbb{N} \to \mathbb{R}$, given by $d(t) = \sum_{\tau=0}^{t}\eta(\tau)$ and a real analytic function $f(t)$ satisfying $f(d(t)) = g(t)$ for all $t \in \mathbb{N}$ and $\lim_{t\to\infty}\frac{\eta(t)f''(d(t))}{f'(d(t))} = 0$. Substituting this $f$ in the equation above, we can say

$$\lim_{t\to\infty}\frac{f'(d(t))}{e^{-\rho f(d(t))^L}f(d(t))^{L+1}} = c$$

Thus $f(d(t))$ grows at $\Theta(log(d(t))^{\frac{1}{L}})$. Now, to get convergence rate for loss, multiply and divide the equation by $\rho f(d(t))^{L-1}$ and denoting $h(d(t)) = \rho f(d(t))^L$, we get

$$\lim_{t\to\infty}\frac{\rho^2 h'(d(t))}{e^{-h(d(t))}h(d(t))^2} = c$$

Thus, $h(d(t))$ grows at the $\Theta(\log(d(t)) + 2\log\log d(t))$. Now as $g(t) = f(d(t))$ and $\rho g(t)^L = h(d(t))$, therefore $g(t)$ asymptotically grows at $\Theta(\log d(t)^{\frac{1}{L}})$ and loss goes down asymptotically at $\Theta(\frac{1}{d(t)\log d(t)^2})$. $\qquad\square$

# H    CROSS-ENTROPY LOSS

In this section, we will provide the corresponding assumptions and theorems, along with their proofs, for cross-entropy loss.

## H.1    NOTATIONS

Let $k$ denote the total number of classes. As $\Phi(\mathbf{w}, \mathbf{x}_i)$ is a multidimensional function for multi-class classification, let's denote the $j^{th}$ component of the output by $\Phi_j(\mathbf{w}, \mathbf{x}_i)$. Also, denote the margin for $j^{th}$ class corresponding to $i^{th}$ data point($j \neq y_i$) by $\rho_{i,j}$, i.e, $\rho_{i,j} = \Phi_{y_i}(\widetilde{\mathbf{w}}, \mathbf{x}_i) - \Phi_j(\widetilde{\mathbf{w}}, \mathbf{x}_i)$. Margin for a data point $i$ is defined as $\rho_i = \min_{j \neq y_i} \rho_{i,j}$. The margin for the entire network is defined as $\rho = \min_i \rho_i$. Also, define a matrix $M(\mathbf{w})$ of dimensions $(m, k)$, that is given by

$$M(\mathbf{w})[i, j] = \begin{cases} 0 & y_i = j \\ e^{-(\Phi_{y_i}(\mathbf{w}, \mathbf{x}_i) - \Phi_j(\mathbf{w}, \mathbf{x}_i))} & y_i \neq j \end{cases}$$

Also, for a matrix $A$, $vec(A)$ represents the matrix vectorized column-wise.

## H.2    ASSUMPTIONS

The assumptions can be broadly divided into loss function/architecture based assumptions and trajectory based assumptions. The loss functions/architecture based assumptions are shared across both gradient flow and gradient descent.

**Loss function/Architecture based assumptions**

1. $\ell(y_i, \Phi(\mathbf{w}, \mathbf{x}_i)) = \log(1 + \sum_{j \neq y_i} e^{-(\Phi_{y_i}(\mathbf{w}, \mathbf{x}_i) - \Phi_j(\mathbf{w}, \mathbf{x}_i))})$

2. $\Phi(., \mathbf{x})$ is a $\mathcal{C}^2$ function, for a fixed $\mathbf{x}$

3. $\Phi(\lambda \mathbf{w}, \mathbf{x}) = \lambda^L \Phi(\mathbf{w}, \mathbf{x})$, for some $\lambda > 0$ and $L > 0$

**Gradient flow**. For gradient flow, we make the following trajectory based assumptions

(A1) $\lim_{t \to \infty} \mathcal{L}(\mathbf{w}(t)) = 0$

(A2) $\lim_{t \to \infty} \frac{\mathbf{w}(t)}{\|\mathbf{w}(t)\|} := \widetilde{\mathbf{w}}$

(A3) $\lim_{t \to \infty} \frac{vec(M(\mathbf{w}(t)))}{\|vec(M(\mathbf{w}(t)))\|} := vec(\widetilde{M})$

(A4) $\rho > 0$.

All the assumptions above are exactly the same as for exponential loss, except for (A3). Using assumption (A1), we can say

$$\lim_{t \to \infty} \sum_{j \neq y_i} e^{-(\Phi_{y_i}(\mathbf{w}(t), \mathbf{x}_i) - \Phi_j(\mathbf{w}(t), \mathbf{x}_i))} = 0 \tag{39}$$

$$\lim_{t \to \infty} \frac{\log(1 + \sum_{j \neq y_i} e^{-(\Phi_{y_i}(\mathbf{w}(t), \mathbf{x}_i) - \Phi_j(\mathbf{w}(t), \mathbf{x}_i))})}{\sum_{j \neq y_i} e^{-(\Phi_{y_i}(\mathbf{w}(t), \mathbf{x}_i) - \Phi_j(\mathbf{w}(t), \mathbf{x}_i))}} = 1 \tag{40}$$

Thus, we can say, for large enough $t$, $\ell_i \approx \sum_{j \neq y_i} e^{-(\Phi_{y_i}(\mathbf{w}(t), \mathbf{x}_i) - \Phi_j(\mathbf{w}(t), \mathbf{x}_i))}$. Thus, assumption (A3) basically states that, not just the loss vector converges in direction, but its components corresponding to various classes also converge in direction. This is required to show that gradients converge in direction in case of multi-class classification.

**Gradient Descent**. For gradient descent, we also require the learning rate $\eta(t)$ to not grow too fast.

(A5) $\lim_{t \to \infty} \eta(t) \|\mathbf{w}_u(t)\| \|\nabla_{\mathbf{w}_u} \mathcal{L}(\mathbf{w}(t))\| = 0$ for all $u$ in the network

**Proposition 4.** *Under assumptions (A1)-(A4), $\lim_{t \to \infty} \eta(t) \|\mathbf{w}_u(t)\| \|\nabla_{\mathbf{w}_u} \mathcal{L}(\mathbf{w}(t))\| = 0$ holds for every $u$ in the network with $\eta(t) = O(\frac{1}{\mathcal{L}^c})$, where $c < 1$.*

*Proof.* The gradient of the loss function is given by

$$\nabla_{\mathbf{w}} \mathcal{L} = \sum_{i=1}^{m} \frac{1}{1 + \sum_{j \neq y_i} e^{-(\Phi_{y_i}(\mathbf{w}, \mathbf{x}_i) - \Phi_j(\mathbf{w}, \mathbf{x}_i))}} \sum_{j \neq y_i} e^{-(\Phi_{y_i}(\mathbf{w}, \mathbf{x}_i) - \Phi_j(\mathbf{w}, \mathbf{x}_i))} (\nabla_{\mathbf{w}} \Phi_j(\mathbf{w}, \mathbf{x}_i) - \nabla_{\mathbf{w}} \Phi_{y_i}(\mathbf{w}, \mathbf{x}_i)) \tag{41}$$

Under Assumption (A1), $\mathbf{w}(t)$ can be represented as $\mathbf{w}(t) = g(t)\widetilde{\mathbf{w}} + \mathbf{r}(t)$, where $\lim_{t\to\infty} \frac{\|\mathbf{r}(t)\|}{g(t)} = 0$ and $\mathbf{r}(t)^\top \widetilde{\mathbf{w}} = 0$. Using the equation above, we can say

$$\lim_{t\to\infty} \frac{\|\nabla_\mathbf{w}\mathcal{L}(\mathbf{w}(t))\|}{e^{-\rho g(t)^L} g(t)^{L-1}} = k$$

As the order remains the same as in the proof for exponential loss, the proof follows from Appendix B. $\qquad\square$

This proposition establishes that the Assumption (A5) is mild and holds for constant $\eta(t)$, that is generally used in practice.

## H.3 EFFECT OF NORMALISATION ON WEIGHT AND GRADIENT NORMS

This section contains the main theorems and the difference between EWN and SWN that makes EWN asymptotically relatively sparse as compared to SWN. First, we will state a common proposition for both SWN and EWN.

**Proposition 5.** *Under assumptions (A1)-(A4) for gradient flow and (A1)-(A5) for gradient descent, for both SWN and EWN, the following hold:*

(i) $\lim_{t\to\infty} \frac{-\nabla_\mathbf{w}\mathcal{L}(\mathbf{w}(t))}{\|\nabla_\mathbf{w}\mathcal{L}(\mathbf{w}(t))\|} = \mu \sum_{i=1}^m \sum_{j\neq y_i} \widetilde{M}(i,j)(\nabla_\mathbf{w}\Phi_{y_i}(\widetilde{\mathbf{w}}, \mathbf{x}_i) - \nabla_\mathbf{w}\Phi_j(\widetilde{\mathbf{w}}, \mathbf{x}_i)) = \widetilde{\mathbf{g}}$, *where* $\mu > 0$.

(ii) $\|\widetilde{\mathbf{w}}_u\| > 0 \implies \widetilde{\mathbf{w}}_u = \lambda\widetilde{\mathbf{g}}_u$ *for some* $\lambda > 0$

**(i)** $\lim_{t\to\infty} \frac{-\nabla_\mathbf{w}\mathcal{L}(\mathbf{w}(t))}{\|\nabla_\mathbf{w}\mathcal{L}(\mathbf{w}(t))\|} = \mu \sum_{i=1}^m \sum_{j\neq y_i} \widetilde{M}[i,j](\nabla_\mathbf{w}\Phi_{y_i}(\widetilde{\mathbf{w}}, \mathbf{x}_i) - \nabla_\mathbf{w}\Phi_j(\widetilde{\mathbf{w}}, \mathbf{x}_i)) = \widetilde{\mathbf{g}}$, where $\mu > 0$

*Proof.* using Assumption (A1), $\mathbf{w}(t)$ can be represented as $\mathbf{w}(t) = g(t)\widetilde{\mathbf{w}} + \mathbf{r}(t)$, where $\lim_{t\to\infty} \frac{\|\mathbf{r}(t)\|}{g(t)} = 0$. Then,

$$e^{-(\Phi_{y_i}(\mathbf{w}, \mathbf{x}_i) - \Phi_j(\mathbf{w}, \mathbf{x}_i))} = e^{-g(t)^L((\Phi_{y_i}(\widetilde{\mathbf{w}} + \frac{\mathbf{r}(t)}{g(t)}, \mathbf{x}_i) - \Phi_j(\widetilde{\mathbf{w}} + \frac{\mathbf{r}(t)}{g(t)}, \mathbf{x}_i))}$$

$$\nabla_\mathbf{w}\Phi_j(\mathbf{w}, \mathbf{x}_i) - \nabla_\mathbf{w}\Phi_{y_i}(\mathbf{w}, \mathbf{x}_i) = g(t)^{L-1}(\nabla_\mathbf{w}\Phi_j(\widetilde{\mathbf{w}} + \frac{\mathbf{r}(t)}{g(t)}, \mathbf{x}_i) - \nabla_\mathbf{w}\Phi_{y_i}(\widetilde{\mathbf{w}} + \frac{\mathbf{r}(t)}{g(t)}, \mathbf{x}_i))$$

Now, using $\rho > 0$ and Euler's homogeneity theorem, we can say

$$\widetilde{\mathbf{w}}^\top(\nabla_\mathbf{w}\Phi_j(\mathbf{w}, \mathbf{x}_i) - \nabla_\mathbf{w}\Phi_{y_i}(\mathbf{w}, \mathbf{x}_i)) = L((\Phi_{y_i}(\mathbf{w}, \mathbf{x}_i) - \Phi_j(\mathbf{w}, \mathbf{x}_i)) > 0$$

Thus, $\|\nabla_\mathbf{w}\Phi_j(\mathbf{w}, \mathbf{x}_i) - \nabla_\mathbf{w}\Phi_{y_i}(\mathbf{w}, \mathbf{x}_i)\| > 0$ for all $i, j$. Using these facts, Equation (39), Equation (40) and Equation (41), we can say

$$\lim_{t\to\infty} \frac{-\nabla_\mathbf{w}\mathcal{L}(\mathbf{w}(t))}{\|\nabla_\mathbf{w}\mathcal{L}(\mathbf{w}(t))\|} = \mu \sum_{i=1}^m \sum_{j\neq y_i} \widetilde{M}[i,j](\nabla_\mathbf{w}\Phi_{y_i}(\widetilde{\mathbf{w}}, \mathbf{x}_i) - \nabla_\mathbf{w}\Phi_j(\widetilde{\mathbf{w}}, \mathbf{x}_i))$$

$\qquad\square$

**(ii)** $\|\widetilde{\mathbf{w}}_u\| > 0 \implies \widetilde{\mathbf{w}}_u = \lambda\widetilde{\mathbf{g}}_u$ for some $\lambda > 0$

*Proof.* The proof follows from Appendix C. $\qquad\square$

The first and second part state that under the given assumptions, for both SWN and EWN, gradients converge in direction and the weights that contribute to the final direction of $\mathbf{w}$, converge in opposite direction of the gradients. Now, we provide the main theorem that distinguishes SWN and EWN.

**Theorem 4.** *Under assumptions (A1)-(A4) for gradient flow and (A1)-(A5) for gradient descent, the following hold*

(i) *for EWN,* $\|\widetilde{\mathbf{w}}_u\| > 0, \|\widetilde{\mathbf{w}}_v\| > 0 \implies \lim_{t\to\infty} \frac{\|\mathbf{w}_u(t)\|\|\nabla_{\mathbf{w}_u}\mathcal{L}(\mathbf{w}(t))\|}{\|\mathbf{w}_v(t)\|\|\nabla_{\mathbf{w}_v}\mathcal{L}(\mathbf{w}(t))\|} = 1$

(ii) for SWN, $\|\widetilde{\mathbf{w}}_u\| > 0, \|\widetilde{\mathbf{w}}_v\| > 0 \implies \lim_{t\to\infty} \frac{\|\mathbf{w}_u(t)\|\|\nabla_{\mathbf{w}_v}\mathcal{L}(\mathbf{w}(t))\|}{\|\mathbf{w}_v(t)\|\|\nabla_{\mathbf{w}_u}\mathcal{L}(\mathbf{w}(t))\|} = 1$

*Proof.* The proof follows from Appendix D. □

### H.4 SPARSITY INDUCTIVE BIAS FOR EXPONENTIAL WEIGHT NORMALISATION

The following proposition shows why EWN is likely to converge to relatively sparse points.

**Proposition 6.** *Let assumptions (A1)-(A5) be satisfied. Consider two nodes $u$ and $v$ in the network such that $\|\widetilde{\mathbf{g}}_v\| > 0, \|\widetilde{\mathbf{g}}_u\| > 0$, $\|\mathbf{w}_u(t)\| \to \infty$ and $\|\mathbf{w}_v(t)\| \to \infty$. Let $\frac{\|\widetilde{\mathbf{g}}_u\|}{\|\widetilde{\mathbf{g}}_v\|}$ be denoted by $c$. Let $\epsilon, \delta$ be such that $0 < \epsilon < c$ and $0 < \delta < 2\pi$. Then, the following holds:*

1. *There exists a time $t_1$, such that for all $t > t_1$ both SWN and EWN trajectories have the following properties:*

   (a) $\frac{\|\nabla_{\mathbf{w}_u}\mathcal{L}(\mathbf{w}(t))\|}{\|\nabla_{\mathbf{w}_v}\mathcal{L}(\mathbf{w}(t))\|} \in [c - \epsilon, c + \epsilon]$

   (b) $\left(\frac{\mathbf{w}_u(t)}{\|\mathbf{w}_u(t)\|}\right)^\top \left(\frac{-\nabla_{\mathbf{w}_u}\mathcal{L}(\mathbf{w}(t))}{\|\nabla_{\mathbf{w}_u}\mathcal{L}(\mathbf{w}(t))\|}\right) \geq \cos(\delta)$

   (c) $\left(\frac{\mathbf{w}_v(t)}{\|\mathbf{w}_v(t)\|}\right)^\top \left(\frac{-\nabla_{\mathbf{w}_v}\mathcal{L}(\mathbf{w}(t))}{\|\nabla_{\mathbf{w}_v}\mathcal{L}(\mathbf{w}(t))\|}\right) \geq \cos(\delta).$

2. *for SWN, $\lim_{t\to\infty} \frac{\|\mathbf{w}_u(t)\|}{\|\mathbf{w}_v(t)\|} = c$*

3. *for EWN, if at some time $t_2 > t_1$,*

   (a) $\frac{\|\mathbf{w}_u(t_2)\|}{\|\mathbf{w}_v(t_2)\|} > \frac{1}{(c-\epsilon)\cos(\delta)} \implies \lim_{t\to\infty} \frac{\|\mathbf{w}_u(t)\|}{\|\mathbf{w}_v(t)\|} = \infty$

   (b) $\frac{\|\mathbf{w}_u(t_2)\|}{\|\mathbf{w}_v(t_2)\|} < \frac{\cos(\delta)}{c+\epsilon} \implies \lim_{t\to\infty} \frac{\|\mathbf{w}_u(t)\|}{\|\mathbf{w}_v(t)\|} = 0$

The above proposition shows that the limit property of the weights in Theorem 4 makes non-sparse $\mathbf{w}$ an unstable convergent direction for EWN, while that is not the case for SWN.

*Proof.* The proof follows from Appendix E. □

### H.5 CONVERGENCE RATES

**Theorem 5.** *For EWN, under Assumptions (A1)-(A5) and $\lim_{t\to\infty} \frac{\|\mathbf{r}(t+1)-\mathbf{r}(t)\|}{g(t+1)-g(t)} = 0$, the following hold*

1. *$\|\mathbf{w}\|$ asymptotically grows at $\Theta\left((\log(d(t))^{\frac{1}{L}}\right)$*

2. *$\mathcal{L}(\mathbf{w}(t))$ asymptotically goes down at the rate of $\Theta\left(\frac{1}{d(t)(\log d(t))^2}\right)$.*

*Proof.* The proof follows Appendix G.2, the only difference is in the gradient update. Let $\mathbf{w}$ be represented as $\mathbf{w} = g(t)\widetilde{\mathbf{w}} + \mathbf{r}(t)$, where $\lim_{t\to\infty} \frac{\|\mathbf{r}(t)\|}{g(t)} = 0$. Using Equation (41), we can say

$$\lim_{t\to\infty} \frac{\|\nabla_{\mathbf{w}}\mathcal{L}(\mathbf{w}(t))\|}{e^{-\rho g(t)^L} g(t)^{L-1}} = k$$

As the order remains the same as in the proof for exponential loss, the proof follows from Appendix G.2. □

## I   LEMMA PROOFS

**Lemma 1.** *Under assumptions (A1)-(A4) for gradient flow and (A1)-(A5) for gradient descent, for both SWN and EWN, $\widetilde{\mathbf{w}}_u^\top \widetilde{\mathbf{g}}_u \geq 0$ for all nodes $u$ in the network.*

*Proof.* We will show the proof just for exponential parameterization and gradient descent, but other cases can be handled similarly.

We only need to consider nodes having $\|\widetilde{\mathbf{w}}_u\| > 0$ and $\|\widetilde{\mathbf{g}}_u\| > 0$, as for other nodes $\widetilde{\mathbf{w}}_u^\top \widetilde{\mathbf{g}}_u = 0$.

Consider a node $u$ having $\|\widetilde{\mathbf{w}}_u\| > 0$ and $\|\widetilde{\mathbf{g}}_u\| > 0$. Let's say $\widetilde{\mathbf{w}}_u^\top \widetilde{\mathbf{g}}_u < 0$. This means, there exists a time $t_1$, such that for any $t > t_1$, $\mathbf{w}_u(t)^\top (-\nabla_{\mathbf{w}_u} \mathcal{L}(\mathbf{w}(t))) < 0$. Then using Equation (11), we can say that, for $t > t_1$, $\alpha_u(t+1) < \alpha_u(t)$. But, this contradicts the assumption that $\|\mathbf{w}_u\| \to \infty$. Thus, $\widetilde{\mathbf{w}}_u^\top \widetilde{\mathbf{g}}_u \geq 0$. $\qquad\square$

**Lemma 2.** *Under assumptions (A1)-(A4) for gradient flow and (A1)-(A5) for gradient descent, for both SWN and EWN, there exists atleast one node $u$ in the network satisfying $\|\widetilde{\mathbf{w}}_u\| > 0$ and $\|\widetilde{\mathbf{g}}_u\| > 0$.*

*Proof.* Under the assumption that $\rho > 0$ and using Euler's homogeneous theorem, using Proposition 2, we can say

$$\widetilde{\mathbf{w}}^\top \widetilde{\mathbf{g}} = \mu \sum_{i=1}^m \widetilde{\ell}_i y_i \widetilde{\mathbf{w}}^\top \nabla_{\mathbf{w}} \Phi(\widetilde{\mathbf{w}}, \mathbf{x}_i) = L\mu \sum_{i=1}^m \widetilde{\ell}_i y_i \Phi(\widetilde{\mathbf{w}}, \mathbf{x}_i) > 0$$

Thus, there must be atleast one node $s$ satisfying $\|\widetilde{\mathbf{w}}_s\| > 0$ and $\|\widetilde{\mathbf{g}}_s\| > 0$. Similarly, it can be shown for cross-entropy loss as well. $\qquad\square$

**Lemma 3.** *Consider two unit vectors $\mathbf{a}$ and $\mathbf{b}$ satisfying $\mathbf{a}^\top \mathbf{b} \geq 0$ and $\mathbf{a}^\top \mathbf{b} < 1$. Then, there exists a small enough $\epsilon > 0$, such that for any unit vector $\mathbf{c}$ satisfying $\mathbf{c}^\top \mathbf{a} \geq \cos(\epsilon)$ and any unit vector $\mathbf{d}$ satisfying $\mathbf{d}^\top \mathbf{b} \geq \cos(\epsilon)$, $\mathbf{b}^\top (I - \mathbf{c}\mathbf{c}^\top)\mathbf{d} \geq \epsilon$.*

*Proof.* First we will try to find bounds on $\mathbf{b}^\top \mathbf{c}$ and $\mathbf{c}^\top \mathbf{d}$.

$$\begin{aligned}
\mathbf{b}^\top \mathbf{c} &= \mathbf{b}^\top (\mathbf{a} + \mathbf{c} - \mathbf{a}) \\
&= \mathbf{b}^\top \mathbf{a} + \mathbf{b}^\top (\mathbf{c} - \mathbf{a}) \\
\mathbf{c}^\top \mathbf{d} &= (\mathbf{a} + \mathbf{c} - \mathbf{a})^\top (\mathbf{b} + \mathbf{d} - \mathbf{b}) \\
&= \mathbf{a}^\top \mathbf{b} + \mathbf{a}^\top (\mathbf{d} - \mathbf{b}) + \mathbf{b}^\top (\mathbf{c} - \mathbf{a}) + (\mathbf{c} - \mathbf{a})^\top (\mathbf{d} - \mathbf{b})
\end{aligned}$$

Now, using the fact that $\mathbf{c}^\top \mathbf{a} \geq \cos(\epsilon)$ and $\mathbf{d}^\top \mathbf{b} \geq \cos(\epsilon)$, we can say $\|\mathbf{c} - \mathbf{a}\| \leq \sqrt{2 - 2\cos(\epsilon)}$ and $\|\mathbf{d} - \mathbf{b}\| \leq \sqrt{2 - 2\cos(\epsilon)}$. Using these bounds and the equation above, we can say

$$\begin{aligned}
\mathbf{b}^\top \mathbf{c} &\leq \mathbf{a}^\top \mathbf{b} + \sqrt{2 - 2\cos(\epsilon)} \\
\mathbf{c}^\top \mathbf{d} &\leq \mathbf{a}^\top \mathbf{b} + 2\sqrt{2 - 2\cos(\epsilon)} + (2 - 2\cos(\epsilon))
\end{aligned}$$

Using these, we can say

$$\mathbf{b}^\top (I - \mathbf{c}\mathbf{c}^\top)\mathbf{d} \geq \cos(\epsilon) - (\mathbf{a}^\top \mathbf{b} + \sqrt{2 - 2\cos(\epsilon)})(\mathbf{a}^\top \mathbf{b} + 2\sqrt{2 - 2\cos(\epsilon)} + (2 - 2\cos(\epsilon)))$$

Now, we need to show that there exists an $\epsilon > 0$ such that $\cos(\epsilon) - (\mathbf{a}^\top \mathbf{b} + \sqrt{2 - 2\cos(\epsilon)})(\mathbf{a}^\top \mathbf{b} + 2\sqrt{2 - 2\cos(\epsilon)} + (2 - 2\cos(\epsilon))) > \epsilon$. At $\epsilon = 0$, LHS takes the value $1 - (\mathbf{a}^\top \mathbf{b})^2$, while RHS takes the value 0. Thus, by continuity with respect to $\epsilon$ of the functions involved, we can say that there exists an $\epsilon > 0$ for which the condition is satisfied. $\qquad\square$

**Lemma 4.** *Consider sequence $a$ satisfying the following properties*

    *1. $a_k > 0$*

    *2. $\sum_{k=0}^\infty a_k = \infty$*

    *3. $\lim_{k\to\infty} a_k = 0$*

*Then $\sum_{k=0}^\infty \frac{a_k}{\sqrt{\sum_{j=0}^k a_j^2}} = \infty$*

*Proof.* If $\sum_{k=0}^{\infty} a_k^2$ is bounded, then the statement is obvious. Let's consider the case when $\sum_{k=0}^{\infty} a_k^2$ diverges. As $\lim_{k\to\infty} a_k = 0$, therefore there must be an index $k_1$, such that for $k \geq k_1$, $a_k \leq \epsilon$. Now, as $a_k \leq \epsilon$, therefore $a_k^2 \leq \epsilon a_k$. Now, as $\sum_{k=0}^{\infty} a_k^2$ diverges, therefore, there must be an index $k_2 > k_1$, such that for any $k > k_2$, $\sum_{j=k_1}^{k} a_j^2 \geq \sum_{j=0}^{k_1-1} a_j^2$. Now, for $k > k_2$, we can say

$$\sum_{j=k_1}^{k} \frac{a_j}{\sqrt{\sum_{l=0}^{j} a_l^2}} \geq \frac{1}{\sqrt{2}} \sum_{j=k_1}^{k} \frac{a_j}{\sqrt{\sum_{l=k_1}^{j} a_l^2}}$$

$$\geq \frac{1}{\sqrt{2}} \sum_{j=k_1}^{k} \frac{a_j}{\sqrt{\sum_{l=k_1}^{j} \epsilon a_l}}$$

$$\geq \frac{1}{\sqrt{2\epsilon}} \sum_{j=k_1}^{k} \frac{a_j}{\sqrt{\sum_{l=k_1}^{k} a_l}}$$

$$= \frac{1}{\sqrt{2\epsilon}} \sqrt{\sum_{j=k_1}^{k} a_j}$$

As $\sum_{k=0}^{\infty} a_k$ diverges, therefore $\sum_{k=0}^{\infty} \frac{a_k}{\sqrt{\sum_{j=0}^{k} a_j^2}}$ diverges as well. □

**Lemma 5.** *Consider two sequences a and b satisfying the following properties*

1. $a_k > 0, \sum_{k=0}^{\infty} a_k = \infty$ *and* $\lim_{k\to\infty} a_k = 0$

2. $b_0 > 0$, *b is increasing and* $b_{k+1}^2 \leq b_k^2 + (\frac{a_k}{b_k})^2$

*Then* $\sum_{k=0}^{\infty} \frac{a_k}{b_k} = \infty$.

*Proof.* As we know $b$ is increasing and $b_{k+1}^2 \leq b_k^2 + (\frac{a_k}{b_k})^2$, we get

$$b_k \leq \sqrt{b_0^2 + \sum_{j=0}^{k-1} (\frac{a_j}{b_j})^2} \leq \sqrt{b_0^2 + \frac{1}{b_0^2} \sum_{j=0}^{k-1} a_j^2}$$

Using this, we can say

$$\sum_{j=0}^{k} \frac{a_j}{b_j} \geq \sum_{j=0}^{k} \frac{a_j}{\sqrt{b_0^2 + \frac{1}{b_0^2} \sum_{l=0}^{j-1} a_l^2}} \geq \sum_{j=0}^{k} \frac{a_j}{\sqrt{b_0^2 + \frac{1}{b_0^2} \sum_{l=0}^{k-1} a_l^2}}$$

Now, if $\sum_{k=0}^{\infty} a_k^2$ does not diverge to infinity, then $b$ remains bounded using the bound above and then its trivial to establish that $\sum_{k=0}^{\infty} \frac{a_k}{b_k}$ diverges. In case, $\sum_{k=0}^{\infty} a_k^2$ diverges to infinity, then there must be an index $k_1$ such that for any $k > k_1$, we can say $\sum_{j=0}^{k-1} a_j^2 \geq b_0^4$. So, for $k > k_1$, we can say

$$\sum_{j=0}^{k} \frac{a_j}{b_j} \geq \sum_{j=0}^{k} \frac{b_0}{\sqrt{2}} \frac{a_j}{\sqrt{\sum_{l=0}^{k-1} a_l^2}}$$

Now, as we have assumed $a$ tends to zero, so there must be an index $k_2$ such that for any $k > k_2$, $a_k \leq \epsilon$. Also, as we have assumed $\sum_{j=0}^{\infty} a_j^2$ diverges, therefore there must be an index $k_3 > k_2$, such that for $k > k_3$, $\sum_{j=k_2}^{k} a_j^2 \geq \sum_{j=0}^{k_2} a_j^2$. Using these things and that if $a_j \leq \epsilon$, then $a_j^2 \leq \epsilon a_j$, we can say for $k > k_3$,

$$\sum_{j=k_3}^{k} \frac{a_j}{b_j} \geq \sum_{j=k_3}^{k} \frac{b_0}{2} \frac{a_j}{\sqrt{\sum_{l=k_3}^{k-1} \epsilon a_l}} \geq \frac{b_0}{2\sqrt{\epsilon}} \sqrt{\sum_{j=k_3}^{k-1} a_j}$$

Now, as $\sum_{k=0}^{\infty} a_k$ diverges, thus $\sum_{k=0}^{\infty} \frac{a_k}{b_k}$ diverges as well. □

**Lemma 6.** *Consider two sequences a and b satisfying the following properties*

1. $a_k > 0$ and $\sum_{k=0}^{\infty} a_k = \infty$

2. $b_k > 0$ and $\sum_{k=0}^{\infty} b_k = \infty$

3. $\sum_{k=0}^{\infty}(a_k - b_k)$ *converges to a finite value*

4. $lim_{k\to\infty} \frac{a_k}{b_k}$ *exists*

*Then* $\lim_{k\to\infty} \frac{a_k}{b_k} = 1$.

*Proof.* Let's say $\lim_{k\to\infty} \frac{a_k}{b_k} = c > 1$. The other case can be handled similarly. Choose an $\epsilon > 0$ such that $c - \epsilon > 1$. Then, there exists an index $k_1$, such that for $k > k_1$, we can say

$$c - \epsilon \le \frac{a_k}{b_k} \le c + \epsilon$$

Using this, we can say, for $k > k_1$,

$$b_k(c - \epsilon - 1) \le a_k - b_k \le b_k(c + \epsilon - 1)$$

Summing the equation above from $k_1$ to $\infty$ and recognizing that $\sum_{k=0}^{\infty} b$ diverges, we get $\sum_{k=k_1}^{\infty}(a_k - b_k) = \infty$. This contradicts. Therefore $\lim_{k\to\infty} \frac{a_k}{b_k} = 1$. $\square$

## J  INTEGRAL FORM OF STOLZ-CESARO THEOREM

We first state the Stolz-Cesaro Theorem.

**Theorem.** *(Muresan, 2015) Assume that $\{a\}_{k=1}^{\infty}$ and $\{b\}_{k=1}^{\infty}$ are two sequences of real numbers such that $\{b\}_{k=1}^{\infty}$ is strictly monotonic and diverging. Additionally, if $\lim_{k\to\infty} \frac{a_{k+1}-a_k}{b_{k+1}-b_k} = L$ exists, then $\lim_{k\to\infty} \frac{a_k}{b_k}$ exists and is equal to L.*

Now, we state and prove the Integral Form of Stolz-Cesaro Theorem.

**Theorem.** *Consider two functions $f(t)$ and $g(t)$ greater than zero satisfying $\int_a^b f(t)dt < \infty$ and $\int_a^b g(t)dt < \infty$ for every finite $a, b$. For any time t, its known that $\int_t^{\infty} f(t)dt = \infty$ and $\int_t^{\infty} g(t)dt = \infty$. If $\lim_{t\to\infty} \frac{f(t)}{g(t)}$ exist and is equal to L, then $\lim_{t\to\infty} \frac{\int_c^t f(t)dt}{\int_c^t g(t)dt}$ exists for any c and is equal to L.*

*Proof.* **Case 1:** L = 0 or $\infty$:

We will prove for L = $\infty$. The case for 0 can be handled similarly. For any $M > 0$, there must exist a time $t_1 > c$, such that $\frac{f(t)}{g(t)} > M$, for $t > t_1$. Thus we can say for $t > t_1$,

$$\int_c^t f(t)dt > \int_c^{t_1} f(t)dt + M \int_{t_1}^t g(t)dt$$

Adding $M \int_c^{t_1} g(t)dt$ on both the sides, we get

$$\int_c^t f(t)dt + M \int_c^{t_1} g(t)dt > \int_c^{t_1} f(t)dt + M \int_c^t g(t)dt$$

Dividing both sides by $\int_c^t g(t)dt$ and taking limsup $t \to \infty$(using also the fact that $\int_a^b f(t)dt < \infty$ and $\int_a^b g(t)dt < \infty$ for every finite $a, b$), we get

$$\limsup_{t\to\infty} \frac{\int_c^t f(t)dt}{\int_c^t g(t)dt} > M$$

Similarly the equation holds for liminf as well. Thus, both liminf and limsup are greater than $M$ for any $M$. Hence $\lim_{t\to\infty} \frac{\int_c^t f(t)dt}{\int_c^t g(t)dt} = \infty$.

**Case2:** L is finite

In this case, there must exist some time $t_1 > c$, such that $L - \epsilon < \frac{f(t)}{g(t)} < L + \epsilon$.Thus, we can say for $t > t_1$,

$$\int_c^{t_1} f(t)dt + (L - \epsilon) \int_{t_1}^t g(t)dt \leq \int_c^t f(t)dt \leq \int_c^{t_1} f(t)dt + (L + \epsilon) \int_{t_1}^t g(t)dt$$

Taking the left inequality, adding $(L - \epsilon) \int_c^{t_1} g(t)dt$ on both the sides, dividing both the sides by $\int_c^t g(t)dt$ and taking $\liminf_{t \to \infty}$, we get

$$L - \epsilon \leq \liminf_{t \to \infty} \frac{\int_c^t f(t)dt}{\int_c^t g(t)dt}$$

Similarly, taking the right inequality, adding $(L + \epsilon) \int_c^{t_1} g(t)dt$ on both the sides, dividing both the sides by $\int_c^t g(t)dt$ and taking $\limsup_{t \to \infty}$, we get

$$\limsup_{t \to \infty} \frac{\int_c^t f(t)dt}{\int_c^t g(t)dt} \leq L + \epsilon$$

Using the two inequalities, we get, for any $\epsilon > 0$,

$$\limsup_{t \to \infty} \frac{\int_c^t f(t)dt}{\int_c^t g(t)dt} - \liminf_{t \to \infty} \frac{\int_c^t f(t)dt}{\int_c^t g(t)dt} \leq 2\epsilon$$

Thus, $\lim_{t \to \infty} \frac{\int_c^t f(t)dt}{\int_c^t g(t)dt}$ exists and is equal to $L$. $\qquad\square$

## K  STANDARD WEIGHT NORMALIZATION IS NOT LOCALLY LIPSCHITZ IN ITS PARAMETERS

In this section, we will denote $\mathbf{w}$ by $\boldsymbol{\theta}$ so as to be consistent with the notaion in Lyu & Li (2020). SWN(in its parameters $\gamma$ and $\mathbf{v}$) is also a homogeneous network. Therefore, results from Lyu & Li (2020) should directly apply to the case of SWN as well. However, a crucial point to be noted is that it is not even locally Lipschitz around $\|\mathbf{v}_u\| = 0$. Therefore, the assumptions from Lyu & Li (2020) do not hold.

However, during gradient descent or gradient flow, if started from a finite $\|\mathbf{v}_u\| > 0$, for all $u$, then during the entire trajectory, $\|\mathbf{v}_u\|$ cannot go down. Therefore, the network is still locally Lipschitz along the trajectory it takes. Examining the proofs from Lyu & Li (2020), its clear that the proof regarding monotonicity of margin and convergence rates are just dependent on the path that gradient descent/flow takes and thus the proofs hold.

However, the result regarding the limit points of $\frac{\boldsymbol{\theta}}{\|\boldsymbol{\theta}\|}$ do not hold. One of the crucial theorems the proof relies on is stated below

**Theorem.** *Let $\{x_k \in \mathcal{R}^d : k \in \mathbb{N}\}$ be a sequence of feasible points of an optimization problem (P), $\{\epsilon_k > 0 : k \in \mathbb{N}\}$ and $\{\delta_k > 0 : k \in \mathbb{N}\}$ be two sequences. $x_k$ is an $(\epsilon_k, \delta_k)$-KKT point for every $k$ and $\epsilon_k \to 0, \delta_k \to 0$. If $x_k \to x$ as $k \to \infty$ and MFCQ holds at $x$, then $x$ is a KKT point of (P)*

The above statement requires MFCQ to be satisfied at $x$, that was shown in Lyu & Li (2020) assuming local lipschitzness/smoothness at $x$. However, in this case, for gradient flow, as $\|\mathbf{v}_u\|$ does not grow, while $|\gamma_u| \to \infty$, therefore the convergent point of $\frac{\boldsymbol{\theta}}{\|\boldsymbol{\theta}\|}$ will always have the component corresponding to $\mathbf{v}_u$ as 0. Thus, the network is not locally lipschitz at $x$ and the proof that MFCQ holds is violated. Similarly, for gradient descent as well, it can't be said that $\mathbf{v}_u$ has a non-zero component in $\frac{\boldsymbol{\theta}}{\|\boldsymbol{\theta}\|}$. Thus, the proof does not hold.

## L  EXPERIMENT DETAILS

In all the experiments, techniques for handling numerical underflow were used as described in Lyu & Li (2020). However, the learning rate they used was of $O(\frac{1}{L})$, but in our case, we generally modify it to be $O(\frac{1}{L^c})$, where $c < 1$.

### L.1 LIN−SEP

The learning rate used was $\frac{k(t)}{\mathcal{L}^{0.97}}$, so that it speeds up at the beginning of training, but slows down as loss approaches $e^{-300}$. The constant $k(t)$ was initialized at $0.01$, and was increased by a factor of $1.1$ every time loss went down and decreased by a factor of $1.1$ every time loss went up after a gradient step. Its value was capped at $0.01$ for EWN and SWN.

### L.2 SIMPLE−TRAJ

The learning rate used was $\frac{k(t)}{\mathcal{L}^{0.9}}$, so that it speeds up at the beginning of training, but slows down as loss approaches $e^{-50}$. The constant $k(t)$ was initialized at $0.01$, and was increased by a factor of $1.1$ every time loss went down and decreased by a factor of $1.1$ every time loss went up after a gradient step. Its value was capped at $0.1$ for EWN and Unnorm.

### L.3 XOR

The learning rate used was $\frac{k(t)}{\mathcal{L}^{0.93}}$, so that it speeds up at the beginning of training, but slows down as loss approaches $e^{-50}$. The constant $k(t)$ was initialized at $0.01$, and was increased by a factor of $1.1$ every time loss went down and decreased by a factor of $1.1$ every time loss went up after a gradient step. Its value was capped at $0.01$ for EWN and $0.1$ for other cases.

### L.4 CONVERGENCE RATE EXPERIMENT

For all SWN, EWN and Unnorm, the learning rate was constant $\eta = 0.01$ and they were trained for $5000$ steps. All the networks were explicitly initialized to the same point in function space.

### L.5 MNIST PRUNING EXPERIMENT

The learning rate used was $\frac{k(t)}{\mathcal{L}}$. The constant $k(t)$ was initialized at $0.01$, and was increased by a factor of $1.1$ every time loss went down and decreased by a factor of $1.1$ every time loss went up after an epoch. Its value was capped at $0.01$ for all the cases.

## M RESULTS FOR STANDARD WEIGHT NORMALIZATION

## N MNIST PRUNING EXPERIMENTS

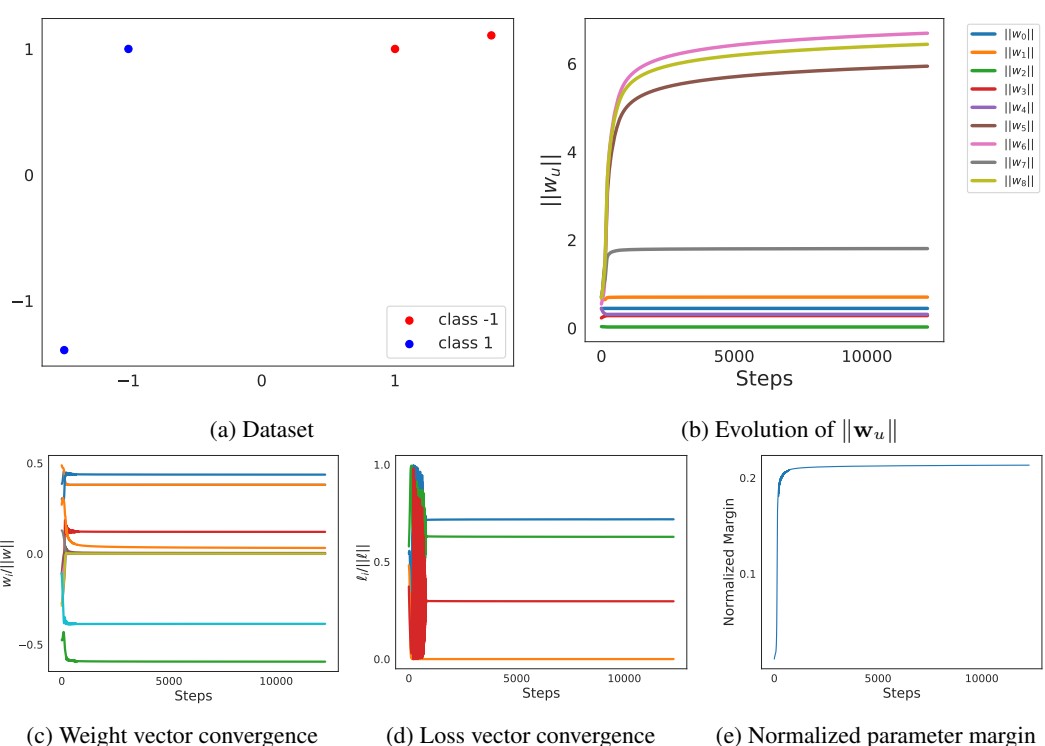

(a) Dataset             (b) Evolution of $\|\mathbf{w}_u\|$

(c) Weight vector convergence     (d) Loss vector convergence     (e) Normalized parameter margin

Figure 8: **Verification of assumptions for SWN in `Lin-Sep` experiment:** (a) shows the dataset. In (b), it can be seen that only weights 5, 6 and 8 keep on growing in norm. So, only for these, $\|\widetilde{\mathbf{w}}_u\| > 0$. (c) shows the components of the unit vector $\frac{\mathbf{w}}{\|\mathbf{w}\|}$, only for the weights 5, 6 and 8 as they keep evolving with time. Eventually their contribution to the unit vector become constant. (d) shows the components of the loss vector and they also become constant eventually. (e) shows the normalized parameter margin converging to a value greater than 0.

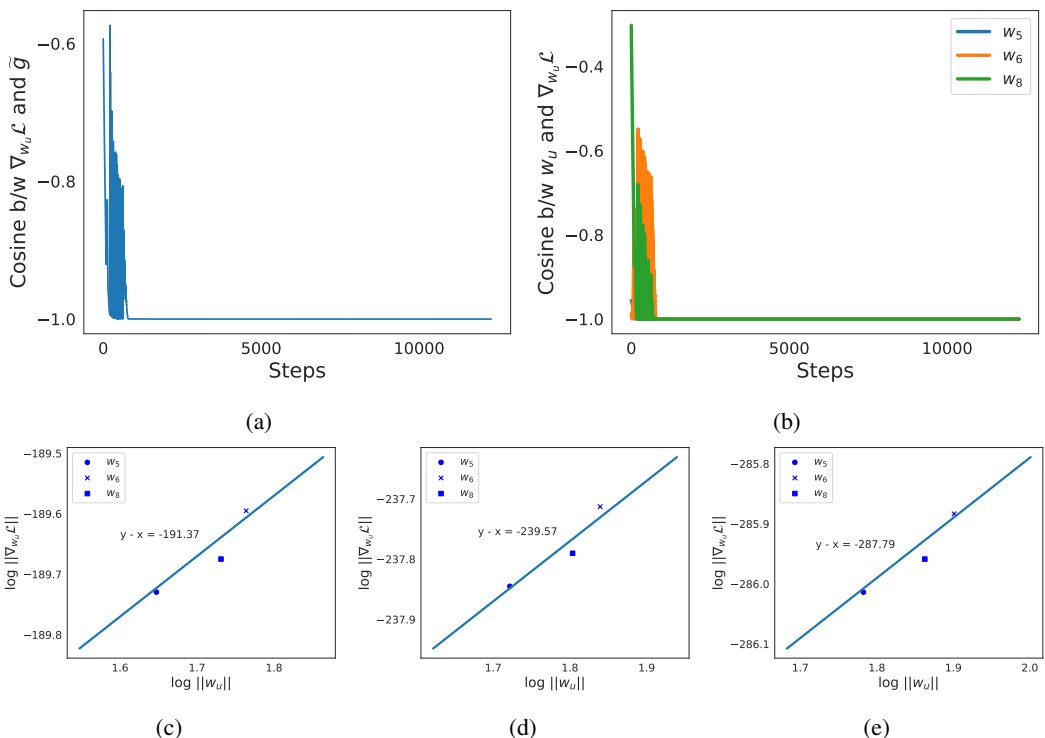

Figure 9: **Demonstration of Results for SWN in `Lin-Sep` experiment:** (a) demonstrates part 1 of Proposition 2, where $\widetilde{\mathbf{g}}$ is approximated by using $\mathbf{w}$ from the last point of the trajectory. Clearly, $\nabla_{\mathbf{w}_u}\mathcal{L}$ stops oscillating and converges to $\widetilde{\mathbf{g}}$. (b) demonstrates part 2 of Proposition 2 and shows that for weight vectors 5,7 and 8, $\mathbf{w}_u(t)$ converges in opposite direction of $\nabla_{\mathbf{w}_u}\mathcal{L}(\mathbf{w}(t))$. (c), (d) and (e) demonstrate Theorem 2 for SWN, where for weight vectors 5,7 and 8. The three graphs are plotted at loss values of $e^{-200}, e^{-250}$ and $e^{-300}$ respectively. At each loss value, for the 3 weights, $\log\|\nabla_{\mathbf{w}_u}\mathcal{L}\| - \log\|\mathbf{w}_u\|$ is approximately same.

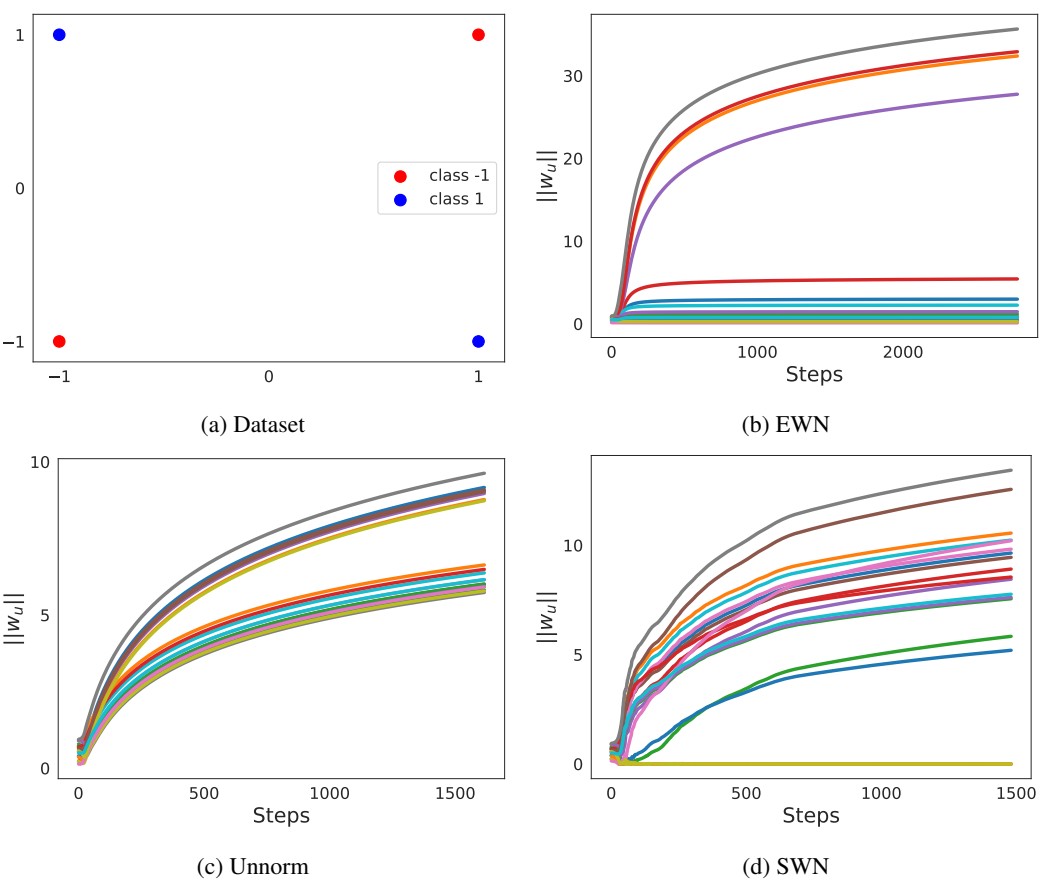

Figure 10: (a) shows the XOR dataset. (b), (c) and (d) demonstrate that EWN weights grow sparsely when compared to Unnorm and SWN

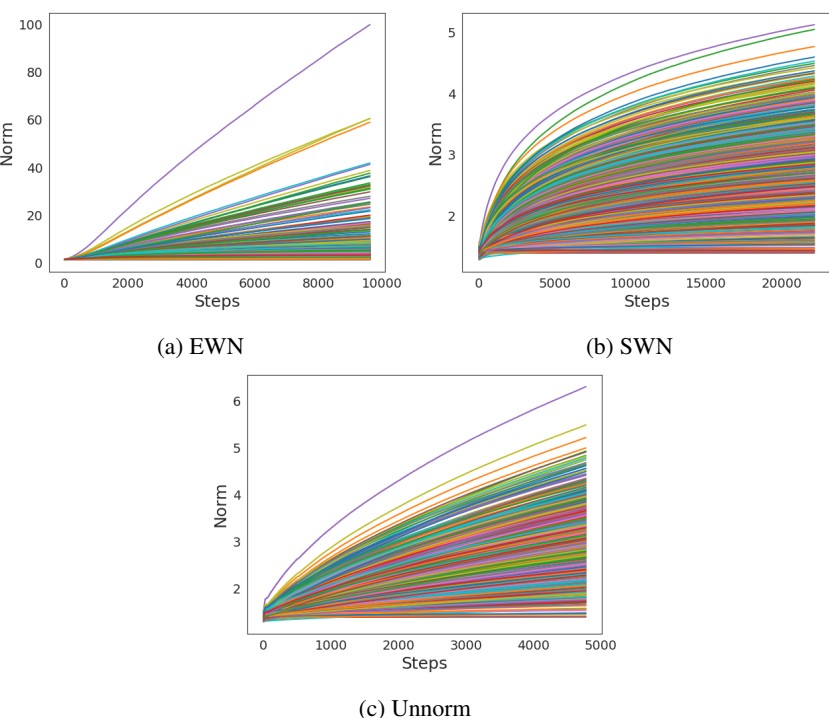

(a) EWN

(b) SWN

(c) Unnorm

Figure 11: Norm of the weight vector vs gradient descent steps, for various nodes in the first layer, when trained on MNIST.

