# OpenReview forum: "Inductive Bias of Gradient Descent for Exponentially Weight Normalized Smooth Homogeneous Neural Nets"
_ICLR.cc/2021/Conference — Reject_

### Official Review · AnonReviewer1 · 2020-10-26
**An interesting work on the implicit bias of gradient descent with exponential weight normalization**

**Rating:** 7
**Confidence:** 3

**Review:**

This paper analyzes the implicit bias of gradient descent with both the standard weight normalization (SWN), which basically uses the gradients with respect to the radial part and spherical part of the weights, and the exponential weight normalization (EWN), which further parameterizes the radial part using an exponential function. Under a few convergence assumptions, it is shown that for SWN, given a node in the network, the norm of the input weight vector is proportional to the norm of the gradient with respect to this weight vector, while for EWN, the norm of the weight vector is inversely proportional to the norm of gradient. It is further shown that such an implicit bias implies that EWN induces sparse limiting directions, and empirical support is provided.

I think SWN and EWN proposed by this paper are interesting, and it is surprising that they introduce opposite implicit biases. It is also interesting that EWN can find sparse or "simple" solutions.

I have the following questions regarding experiments:
1. Can Proposition 3 be verified on MNIST? For example, can you compare the distribution of norms of weight vectors for EWN, SWN, and unnormalized gradient descent?
2. Can EWN also improve generalization or sparsity on more complicated datasets, such as CIFAR?

---

> ### Author Response · Authors · 2020-11-12
> **Response to comments**
>
> We thank the reviewer for the detailed comments. Responses to individual comments below. The numbering of equations and propositions here correspond to the original submission and not the revised version.
> 1. We have added Figure 11 in Appendix N depicting the weight norms attained in case of MNIST experiment in the revised version.
> 2. It is possible that EWN on more complicated datasets will also improve sparsity. The sparsity of the asymptotic solution (under assumptions A1-A5) is likely to enable the final learned network to be more robust to pruning. However, with multiple hidden layers the pruning strategy to choose is not particularly clear. That is the reason why in the MNIST experiment here we have used a single hidden layer network.
> It should be emphasised that we are providing the MNIST experiment as a proof of concept that EWN results in sparsity on real-world datasets and could be beneficial for pruning. However, for deeper nets, even with EWN, we believe better pruning strategies would need to be designed, rather than pruning all the layers on the basis of difference in initial and final norm.

---

### Official Review · AnonReviewer2 · 2020-10-28
**Solid results for analyzing normalization methods, but significance is not exactly clear and several results lack details.**

**Rating:** 7
**Confidence:** 4

**Review:**

### Summary
This paper studies the inductive bias of gradient methods with normalization on smooth homogeneous models. The focus is on two normalization methods, standard weight normalization (SWN) and exponential weight normalization (EWN). The authors show two main results. The first characterizes the trajectory of normalized gradient methods from which they provide theoretical evidence that EWN is biased towards sparse solutions. The second provides convergence rates for the normalized methods which shows the difference between convergence rates of normalized and unnormalized methods. The theoretical results are corroborated with experiments on several toy datasets.

### Reason for score
I am currently inclined towards accepting the paper because the results are novel, solid and should be interesting and useful for researchers working on theory of deep learning. However, the score is only marginally above the acceptance threshold, because I have several concerns regarding the clarity and significance of the results. I am willing to raise my score if the authors address my concerns in the rebuttal.

### Pros
1.	The theoretical results are solid, novel and the proof techniques might be useful in other inductive bias analyses.
2.	The sparsity result for EWN is interesting and provides novel insights on pruning neural networks as the MNIST experiments show.
3.	Most of the paper is clearly written.

### Cons (roughly ordered from major to minor comments)

1.	It is not clear in which cases SWN and EWN are used in practice. The authors do not explicitly cite papers that use them. Therefore, it is not clear how to assess the significance of the results.
2.	After Theorem 2 it is claimed that ||w_u(t)|| is inversely proportional to ||grad_u L(t)||. I am not sure why this is correct. If ||w_u(t)|| = t and ||grad_u L(t)|| = 1/t^2 for all u, then the theorem result holds, but the claim after the theorem (mentioned above) does not hold. Am I missing something?
3.	In Proposition 3, the assumption that the ratio of gradient norms is exactly c from some t onwards is very strong. The authors should comment on this. Does it hold in practice? Does the Proposition hold under weaker assumptions?
4.	Most of the experiments are performed on very simple datasets with few points in the training sets. I think that experiments on other datasets (e.g., with 1000s of points) can strengthen the results.
5.	In Proposition 1, eta(t) is said to be a constant but it seems to depend on the loss which changes with time. What is the L in the denominator of the learning rate equation? Is it the loss?
6.	In Figure 1, the neighborhood of a point for different geometries is not formally defined. The current figures are not clear.
7.	In several experiments, it is claimed that the loss achieved values of order e^(-300). This seems like an unrealistic precision to get empirically. Is there a mistake here?
8.	The presentation of the normalization methods in the equations in page 2 is not very clear. Specifically, why these equations result in a form of normalization. Can the updates be presented in a concise equation where the normalization is showed explicitly?
9.	I think that the authors should provide more context to the pruning results in Section 6. Specifically, say why the insights on EWN in previous sections can be useful for pruning applications.

---

> ### Author Response · Authors · 2020-11-12
> **Response to comments**
>
> We thank the reviewer for the detailed comments. Responses to individual comments below. The numbering of equations and propositions here correspond to the original submission and not the revised version.
> 1. SWN was proposed by Salimans and Kingma (2016) as a substitute for Batch Normalization. They showed that under some assumptions SWN can be shown to be equivalent to Batch normalization, while being free of the disadvantages of batch normalization with small batch size. EWN was mentioned in passing in the same paper as another way to reparameterize the weights, and has not been widely used to the best of our knowledge. We have mentioned some of the papers that use SWN in the revised version of the paper.
> 2. The confusion here is caused by imprecise wording in the paper. The inverse proportion statement is for an asymptotic t, over different neurons. In Theorem 2 and the statement later, we simply mean that $||\mathbf{w}_u(t)|| * || \nabla _{\mathbf{w}_u} \mathcal{L}(\mathbf{w}(t)) ||$ does not depend on $u$. We don’t mean to say anything about a dependence on t. We have slightly reworded the statement in the revised version to remove the confusion.
> 3. It is true that in proposition 3, we make an artificial strong assumption for the purpose of showing a sharp instability in the convergent direction. The assumption states “the ratio of two gradient norms at two nodes stays constant forever after some point in training”. It is possible to omit the strong limit behaviour assumption in Proposition 3, for a slightly less sharp instability result in the convergent direction. This is reflected in the revised version.
> 4. The main reason we have very simple datasets is that it is quite difficult to check the validity of assumptions and correctness of the implications on large networks. For the simple 4 point datasets, we can achieve a zero loss using very simple networks (assumption A1) and the small number of nodes enables us to check the assumptions (See Figure 2, where every node gets a curve). It is indeed possible to check the high level implication of sparsity in the learned network, and we do that using the MNIST dataset. We have also added the distribution of the final norm values learnt in case of MNIST in the revised version to demonstrate sparsity in EWN.
> 5. eta(t) being set to a constant works for all the asymptotic results. For gradient flow, any non-zero scalar learning rate (which does not fall too fast) will give the same trajectory. For gradient descent, it should be ensured that the discretisation of the trajectory does not adversely affect the dynamics. A constant learning rate is too conservative however, and even learning rates like O(1/L) , where L is the current loss value, can be shown to work (this is because the loss landscape at low loss values is smoother than at higher loss values). This essentially allows the network to take longer steps than usual when the loss value is extremely small.
> 6. For EWN, the neighborhood is formally defined as follows:
> $N_\epsilon =$ { $\mathbf{w} = e^{\alpha} \frac{\mathbf{v}}{||\mathbf{v}||} :  [\alpha, \mathbf{v}]  \in B([\alpha_c, \mathbf{v}_c], \epsilon)$ } where B(c,r) is the euclidean ball of radius r centred at $c \in \mathbb{R}^{d+1}$. This neighborhood is around the weight vector given by $\mathbf{w}_c = e^{\alpha_c}  \frac{\mathbf{v}_c}{||\mathbf{v}_c||}$.
> We have added this definition to the figure in the revised version of the paper as well.
> 7. No, we really mean $e^{-300}$. We need these absurdly low values to exactly verify the asymptotic inductive bias. We achieve this using techniques from Lyu and Li (2020). Informally, the log loss at the current step is measured relative to the log loss at the previous step, this avoids numerical underflow.
> 8. We are not sure we fully understand this comment, do inform us if this does not answer your question. The equations in Page 2 before Theorem 1 are a result of chain rule for gradients, and the reparameterization of weight vector, as a product of a scalar and a unit vector, $\gamma \frac{\mathbf{v}}{|| \mathbf{v} ||}$ in case of SWN and $e^\alpha \frac{\mathbf{v}}{|| \mathbf{v} ||}$ in case of EWN.
> 9. We have added a motivating line at the start of section 6 in the revised version.

---

> > ### Comment · AnonReviewer2 · 2020-11-22
> > **Thanks for the clarifications**
> >
> > You addressed most of my comments.
> >
> > 3. In the new version of Proposition 3, the first part is an assumption right ("There exists a time $t_1$...")? It currently looks like one of the claims.
> >
> > 5. After Proposition 1, it is still written that Proposition 1 holds for a constant eta(t), but the eta(t) in the Proposition statement is not constant. This is not clear.
> >
> > I read the other reviews and responses and raised the score from 6 to 7.

---

> > > ### Author Response · Authors · 2020-11-23
> > > **Thanks for the comments**
> > >
> > > Thanks very much for your feedback and comments.
> > >
> > > 3. It is indeed a claim that we have proved in Appendix E. We have also added a revised version, elaborating on the proof of this proposition in Appendix E.
> > > 4. The constant eta(t) is a special case when c is set to 0.

---

### Official Review · AnonReviewer3 · 2020-10-29

**Rating:** 5
**Confidence:** 3

**Review:**

This paper analyzes weight normalization methods, including exponential weight normalization (EWN) and standard weight normalization (SWN), in contrast with unnormalized networks. Under a number of assumptions, the paper characterizes the asymptotic relation between weight norm and gradient norm at the node level (Theorem 2), which shows a distinction between SWN and EWN. Then it's argued that SWN leads to sparser solutions (Proposition 3), which is potentially beneficial for pruning. The paper also shows a convergence rate for SWN which is slightly faster than unnormalized and SWN from previous work, but under stronger assumptions. The paper verifies these results empirically on some toy examples.

pros:
+ The exponential weight normalization method seems new.
+ The paper has some interesting findings regarding the asymptotic behavior of weight normalization methods (if the results can be justified properly).

cons:
The theoretical results are based on very strong asymptotic assumptions, which are not justified properly. The experiments are on very toy settings which are far below the bar. Either the theory or the experiments need to be stronger for this paper to be a solid contribution.

- The assumptions (A1)-(A4) used throughout the paper are much stronger than those in previous work, such as Lyu & Li (2020). In particular, (A3) and (A4) are nonstandard. I'm not sure when these assumptions are expected to hold, and they are only empirically verified on an extremely simple dataset (4 examples).

- In Proposition 3, which is where it is shown that SWN leads to sparsity, there is an extremely strong assumption that the ratio of two gradient norms at two nodes stays constant forever after some point in training. How can this possibly be true?


---------- after rebuttal ----------

Thanks for the response and the updated manuscript. I'm raising my score from 4 to 5. I'm still leaning towards rejection since I still find the results quite subtle and I hope to see more empirical justifications.

In the updated Proposition 3, the sparsity-inducing property 3 assumes the existence of a time $t_2>t_1$ when the ratio between the two weight norms deviate from $1/c$. However, it seems entirely possible that this ratio will have already converged $1/c$ after time $t_1$; in this case the two weight norms grow at the same rate. It would be good to investigate this more carefully to see which cases are more likely to happen. I'm also concerned that the advantage of EWN for pruning only shows up in extremely small loss value (Figure 7), and therefore the practical relevance shown in the current paper is not very convincing.

---

> ### Author Response · Authors · 2020-11-12
> **Strength of assumptions**
>
> We thank the reviewer for the detailed comments. Responses to individual comments below. The numbering of equations and propositions here correspond to the original submission and not the revised version.
> 1. We agree that the assumptions made are much stronger than in Lyu & Li (2020). However, an earlier paper by Nacson et al. (2019a) have assumptions that are close to the assumptions (A3) and (A4).
> Moreover, the assumptions (A3) and (A4) can be replaced by the assumptions that the gradient converges in direction (B3) and there exists at least one node $u$ in the network that satisfies $||\widetilde{\mathbf{w}}_u|| > 0$ and $||\widetilde{\mathbf{g}}_u|| > 0$ (B4). In fact, in our proofs, A3 and A4 are used to show these two statements, and influence the proof only through these.
> The new assumption (B3) regarding gradient convergence has been used in previous papers(Gunasekar et al. 2018b) studying inductive bias as well. The new assumption (B4) is very mild as it needs to be satisfied by a single neuron out of the entire network.
> 2. It is true that in proposition 3, we make an artificial strong assumption for the purpose of showing a sharp instability in the convergent direction. The assumption states “the ratio of two gradient norms at two nodes stays constant forever after some point in training”. It is possible to omit the strong limit behaviour assumption in Proposition 3, for a slightly less sharp instability result in the convergent direction. This is reflected in the revised version.

---

### Official Review · AnonReviewer4 · 2020-10-31
**Technical issues**

**Rating:** 4
**Confidence:** 5

**Review:**

The paper studies the gradient flow dynamics over smooth homogeneous models with two types of weight normalized parameterization - standard weight normalization (SWN) and exponentiated weight normalization (EWN). Thm 1 shows the induced dynamics in the unnormalized parameter space resulting from gradient flow on respective weight normalized parameterization. This result is a good starting point that highlights the different dynamics arising from the two parameterizations.

However, in the remainder of the paper, there are several technical issues/confusions, outlined below (p.s., please number the equations):

1. In the proof of Proposition 3 and also Thm 2 (see e.g.,  last eq. in page 23 and corresponding equations for GD in Appendix E.1.2, and similarly, last eqn in page 20), the following equality is used which is not true in general. Please clarify if I missed something: ||w(t)||=||w(t_2)||+int_{k=t_2}^t ||dw(t)|| -- triangle inequality would show that the RHS is an upper bound but I do not see how we can get exact equality.

2. In the proof of Thm 3 (page 20), why does Proposition 2 imply that w_u and its negative gradient are aligned in opposite directions? Specifically, why should there be a t_2 such that for all t>t_2, cos(-\nabla_{w_u} L,w_u)<=\epsilon?

3. In Appendix D.2 (page 22) while bounding ||w_u(t)|| for SWN, along with the above two concerns, I am also not sure how the two terms in ||dw(t)|| from Thm 1 lead to the simplified bounds on ||w_u(t)|| in the first non-thm equation on page 22.

4. Finally, although not a technical mistake, I believe that the discussion comparing between EWN and unnormalized GF (which I will simply call GF) is conceptually confusing. As the authors themselves note, EWN and GF both follow the *same trajectory*. EWN simply has a scaling factor of ||w(t)||^2 which affects the “speed” along the trajectory but the path itself if the same -- both have dw(t) = -s(t) nabla_w L(w(t)) for different scalar speeds s(t) and it corresponds to the same path in the space of w but with different time warping. Thus, if one solves the differential equations indefinitely both EWN and GF will trace the exact same path albeit at different times and will eventually lead to the same separator. But the plots and the discussion about Fig 5 for example suggest that EWN and GF leads to different asymptotic solutions, which is not correct.

Thus, when comparing EWN and GF, the message could be that EWN when discretized could lead to faster convergence - this is somewhat justified experimentally (from Fig 5) but not theoretically as to truly compare one needs to show analysis for the discretized algorithm. Also experimentally to provide correct comparison of the speed, in Fig 5, the number of iterations of the two methods (EWN and GF) should be matched which is not true in the current plots. On the other hand, it is simply wrong to phrase the message as “EWN and GF lead to different solutions asymptotically”.

---

> ### Author Response · Authors · 2020-11-12
> **Neuron dependent adaptive learning rate and response to other comments**
>
> We thank the reviewer for the detailed comments. Responses to individual comments below. The numbering of equations and propositions here correspond to the original submission and not the revised version.
> 1. In proposition 3, we make an artificial strong assumption for the purpose of showing a sharp instability in the convergent direction. The assumption states “the weight vector and its gradient align exactly in opposite directions after some t>t2”. Because of this assumption, equation 10 gets simplified to $\frac{d\gamma_u(t)}{dt} = \frac{d ||w_u(t)||}{dt} = \eta(t) || \nabla _{\mathbf{w}_u} \mathcal{L}(\mathbf{w}(t))||$.
> It is possible to omit the strong limit behaviour assumption in Proposition 3, for a slightly less sharp instability result in the convergent direction. This is reflected in the revised version. The modified proof for SWN here directly appeals to Theorem 2.
> 2. Consider the unit norm vectors along $\mathbf w_u(t)$ and $- \nabla_{\mathbf w_u} \mathcal L(\mathbf w(t))$. Proposition 2 states these two sequences converge to the same value. Hence, for any $\delta$ there exists a $t_1$ such that for all $t>t_1$ the two unit norm vectors are $\delta$ close in euclidean norm. As these are unit vectors the cosine between these two vectors is greater than $\sqrt{1-\frac{\delta^2}{2}}$.
> 3. For SWN, the norm change can be more easily obtained through equation 10. (As $\gamma_u(t) = ||\mathbf{w}_u(t)||$). There is no need to appeal to Theorem 1 directly.
> 4. Maybe we didn’t make this absolutely clear in the paper. Even though for a given neuron $u$ the trajectory equation for EWN is simply a scaled version of the Unnormalized network(GF). However, different neurons have different scalings, leading to very different trajectories for the network as a whole. Simulating the EWN trajectory using unnormalized GF would require an adaptive neuron dependent learning rate. For example, the trajectory for EWN and unnormalized GF are completely different in the  simple-traj experiment (Figure 4). This is because the two parameters are associated with different neurons. Hence, it is correct to say EWN and unnormalized GF lead to different solutions asymptotically.

---

### Author Response · Authors · 2020-11-12
**Summary of changes in the rebuttal revision (version dated Nov 12)**

1. We have modified proposition 3, removing the extra assumptions, resulting in a slightly less sharp instability result.
2. We have added a figure showing the norm of the weights for SWN, EWN and NWN for the MNIST training procedure. It is in Appendix N, Figure 11.
3. A few more references that use SWN in practice have been added to page 3.
4. We have changed the caption of Figure 1 giving a more detailed explanation of the figure.
5. The statement after Theorem 2, has been modified to remove a potential source of confusion.
6. In Section 6, we add a sentence motivating the usage of EWN, for generating a network that is more robust to pruning.
7. We have clarified a few definitions in Proposition 2.
8. We have also fixed a few typos and grammatical errors.

---

### Author Response · Authors · 2020-11-23
**Summary of changes in the rebuttal revision (version dated Nov 23)**

There is no change in the main paper. The changes in the Appendix are listed below:
1. We have provided more details on the proof of Proposition 3 in Appendix E.

---

### Decision · Program_Chairs · 2021-01-07
**Final Decision**

**Decision:**

Reject

**Comment:**

The main concern is that the results in this paper are based on strong asymptotic assumptions. (At least) more empirical results are needed.